# Private Model Personalization Revisited

**Conor Snedeker** [* 1]  **Xinyu Zhou** [* 1]  **Raef Bassily** [2]

## Abstract

We study model personalization under user-level differential privacy (DP) in the shared representation framework. In this problem, there are $n$ users whose data is statistically heterogeneous, and their optimal parameters share an unknown embedding $U^* \in \mathbb{R}^{d \times k}$ that maps the user parameters in $\mathbb{R}^d$ to low-dimensional representations in $\mathbb{R}^k$, where $k \ll d$. Our goal is to privately recover the shared embedding and the local low-dimensional representations with small excess risk in the federated setting. We propose a private, efficient federated learning algorithm to learn the shared embedding based on the FedRep algorithm in (Collins et al., 2021). Unlike (Collins et al., 2021), our algorithm satisfies differential privacy, and our results hold for the case of noisy labels. In contrast to prior work on private model personalization (Jain et al., 2021), our utility guarantees hold under a larger class of users' distributions (sub-Gaussian instead of Gaussian distributions). Additionally, in natural parameter regimes, we improve the privacy error term in (Jain et al., 2021) by a factor of $\tilde{O}(dk)$. Next, we consider the binary classification setting. We present an information-theoretic construction to privately learn the shared embedding and derive a margin-based accuracy guarantee that is independent of $d$. Our method utilizes the Johnson-Lindenstrauss transform to reduce the effective dimensions of the shared embedding and the users' data. This result shows that dimension-independent risk bounds are possible in this setting under a margin loss.

## 1 Introduction

The rapid advances in machine learning have revolutionized domains such as healthcare, finance, and personalized services. However, this progress relies heavily on the availability of user data, which presents two critical challenges. First, user data is often statistically heterogeneous, that is, individual users have distinct data distributions. This heterogeneity hinders the training of a single global model that performs well for all users. Second, the reliance on sensitive user data raises significant privacy concerns, necessitating robust mechanisms that offer strong privacy protections.

Model personalization has emerged as a key strategy to address the challenge of statistical heterogeneity by adapting models to individual users, rather than relying on a single global model that may perform suboptimally across diverse data distributions. A widely used framework for personalization is shared representation learning, where users collaborate to learn a low-dimensional embedding that captures commonalities in their tasks. This shared embedding allows users to train their local models more efficiently, leveraging the embedding to complement their unique data.

However, despite its effectiveness in handling heterogeneity, model personalization introduces additional privacy concerns, as learning shared representations involves user collaboration, which can inadvertently expose sensitive information. Ensuring rigorous privacy protections in this setting requires formal guarantees, such as user-level differential privacy (DP) (Dwork et al., 2006), which prevents adversaries from inferring an individual's presence in the training dataset based on the trained model. Existing approaches to private model personalization suffer from limitations such as restrictive assumptions about data distributions, centralized processing requirements, and suboptimal privacy-utility trade-offs.

In this work, we study model personalization under user-level differential privacy via shared representation learning. In this problem, there are $n$ users, where user $i \in [n]$ is associated with a data distribution $\mathcal{D}_i$ and a dataset $S_i = (z_1, \ldots z_m) \sim \mathcal{D}_i^m$. The optimal parameter of user $i$ is denoted as $w_i^* \in \mathbb{R}^d$. We aim at learning a shared low-dimensional embedding $U^* \in \mathbb{R}^{d \times k}$, where $k \ll d$, and user-specific parameters $v_1^*, \ldots, v_n^* \in \mathbb{R}^k$ such that $w_i^* = U^* v_i^*$ for each user $i \in [n]$, under user-level DP. We

---

*Equal contribution [1] Department of Computer Science & Engineering, The Ohio State University [2] Department of Computer Science & Engineering and the Translational Data Analytics Institute (TDAI), The Ohio State University. Correspondence to: Conor Snedeker <snedeker.31@buckeyemail.osu.edu>, Xinyu Zhou <zhou.3542@buckeyemail.osu.edu>.

*Proceedings of the 42$^{nd}$ International Conference on Machine Learning*, Vancouver, Canada. PMLR 267, 2025. Copyright 2025 by the author(s).

propose a novel private federated algorithm for model personalization via shared representation that achieves superior privacy-utility trade-off under relaxed assumptions. Our contributions are summarized as follows:

## 1.1 Contributions

**Efficient private federated algorithm for regression:** We extend the FedRep algorithm of (Collins et al., 2021) to ensure user-level DP while addressing the case of noisy labels in statistically heterogeneous user data. The algorithm is iterative, where in each iteration $t$, it alternates between local updates by the users to their local vectors $v_{1,t}, \ldots, v_{n,t} \in \mathbb{R}^k$ and private aggregated gradient update to the shared embedding $U_t \in \mathbb{R}^{d \times k}$. Previous work (Jain et al., 2021) required performing exact minimization with respect to both the local vectors and the shared embedding in a centralized manner. Since the embedding update in (Jain et al., 2021) involves a minimization over data from all the users, transforming their algorithm to a federated algorithm is not directly feasible. Meanwhile, ours is naturally a federated algorithm since the gradient is computed by each user locally, which the server privately aggregates. In natural parameter regimes, we show that our algorithm attains excess population risk of $\widetilde{O}\left(\frac{d^2 k}{n^2 \epsilon^2 \sigma_{\min,*}^4} + \frac{d}{nm\sigma_{\min,*}^4}\right) + \frac{k}{m}$ where $\sigma_{\min,*}$ is $k$-th largest singular value of $\frac{1}{\sqrt{n}}V^*$. Compared to (Jain et al., 2021), under the same parameter regimes, this reduces the privacy error term by a factor of $\widetilde{O}(dk)$.

**Robustness to broader data distributions:** Unlike prior work (Jain et al., 2021) that assumes Gaussian data, the utility guarantees of our private federated algorithm apply to a broader class of sub-Gaussian distributions. Moreover, our results apply to statistically heterogeneous users in the sense that individual users may have distinct sub-Gaussian distributions, while the work of (Jain et al., 2021) assumes that the features of all the users are standard Gaussian. Moreover, we extend the results of (Collins et al., 2021) not only to the private case but also to the non-realizable case of noisy labels.

**Improved initialization under privacy constraints:** We introduce a private initialization algorithm for our federated algorithm. Our initialization algorithm is based on the subspace recovery approach of (Duchi et al., 2022). Our algorithm leverages a top-k singular value decomposition (SVD) with added Gaussian noise. This ensures a feasible starting point for the shared embedding, even under relaxed data assumptions.

**Dimension-independent risk guarantees for classification:** We provide an information-theoretic construction for the binary classification setting under the margin loss and derive an risk bound that benefits from margin guarantees. We incorporate the Johnson-Lindenstrauss transform to re-

duce the effective dimensionality of the embedding space. This leads to a margin-based risk bound independent of the input dimension $d$. This result holds for arbitrary distributions where the feature vectors are bounded in the Euclidean norm.

## 1.2 Related Work

Our work builds on and extends several lines of research in federated learning, model personalization, and differential privacy:

**Federated personalization via shared representations:** Shared representation learning has proven effective in addressing user heterogeneity (Collins et al., 2021; Jain et al., 2021). Additionally, centralized subspace recovery methods (Tripuraneni et al., 2021; Duchi et al., 2022) may be used to provide a good initialization for the shared representation for better federated personalization guarantees. While prior work focuses on non-private or centralized settings, our approach incorporates user-level DP in a federated framework and achieves better utility guarantees.

**Differentially private model personalization:** (Jain et al., 2021) introduced a private algorithm for model personalization with centralized processing and restrictive Gaussian assumptions. Our method improves excess risk guarantees and extends applicability to sub-Gaussian data distributions. There are other prior works studying private model personalization like (Bietti et al., 2022; Hu et al., 2021), but these works are not under the representation learning framework and their results are not comparable to ours.

**Optimization methods for representation learning:** Relevant optimization techniques include alternating exact minimization frameworks (Thekumparampil et al., 2021) and gradient-based optimization, which has also been widely used for personalization (e.g., (Collins et al., 2021; Raghu et al., 2019; Lee et al., 2019; Tian et al., 2020)). Our alternating minimization framework adapts these gradient techniques to a privacy-preserving federated setting with noisy labels.

**Dimensionality reduction with DP guarantees:** The use of the Johnson-Lindenstrauss transform in private learning has been explored in recent work (e.g., (Lê Nguyen et al., 2020; Bassily et al., 2022; Arora et al., 2022)). We adapt this technique for federated personalization, deriving margin-based risk guarantees independent of the data dimension $d$.

## 2 Preliminaries

We consider a setting of $n$ users, where each user $i \in [n]$ is associated with distribution $D_i$ observed by a dataset $S_i = ((x_1, y_1), \ldots (x_m, y_m)) \sim D_i^m$ where $x_i \in \mathbb{R}^d$ and $y_i \in \mathbb{R}$. Given a low dimensional embedding matrix $U \in \mathbb{R}^{d \times k}$

with orthonormal columns and local vector $v \in R^k$, the loss incurred by the parameter $(U, v)$ over data point $(x, y)$ is defined as $\ell(U, v, (x, y)) = \ell'(y, \langle x, Uv \rangle)$. Suppose $k \ll d$. It is useful conceptually to think of the matrix $U$ as a transformation $x \mapsto U^\top x$ that maps $x \in \mathbb{R}^d$ onto the low-dimensional space $\mathbb{R}^k$. In addition, we assume that $k \ll m$. This assumption is necessary to attain useful utility guarantees as in prior work (Jain et al., 2018).

Given a collection of data sets $S = (S_1, \ldots, S_n) \in \mathbb{R}^{nm(d+1)}$, embedding matrix $U \in \mathbb{R}^{d \times k}$ and collection of local vectors $V = [v_1, \ldots v_n]^\top \in \mathbb{R}^{n \times k}$, we write the average empirical risk of $(U, V)$ over $S$ as

$$\widehat{L}(U, V; S) = \frac{1}{nm} \sum_{i=1}^{n} \sum_{(x,y) \in S_i} \ell(U, v_i, (x, y))$$

We also define the average population loss of $(U, V)$ over the data distributions $\mathcal{D} = (\mathcal{D}_1, \ldots, \mathcal{D}_n)$ as

$$L(U, V; \mathcal{D}) = \frac{1}{n} \sum_{i=1}^{n} \mathbf{E}_{(x,y) \sim \mathcal{D}_i} \left[ \ell(U, v_i, (x, y)) \right].$$

Given any $(U^*, V^*) \in \mathbb{R}^{d \times k} \times \mathbb{R}^{n \times k}$ for $U^*$ with orthonormal columns, our goal is to minimize the excess population risk w.r.t $(U^*, V^*)$ defined as

$$\mathcal{E}(U, V) = L(U, V; \mathcal{D}) - L(U^*, V^*; \mathcal{D}).$$

Let $M \in \mathbb{R}^{d \times k}$ be any matrix. Recall our setting in which $k \ll n$. Denote the spectral norm of $M$ with $\|M\|_2$ and its Frobenius norm with $\|M\|_F$. We denote by $\sigma_{\max}(M)$ and $\sigma_{\min}(M)$ the largest and $k$-th largest singular values of $M$, respectively. Further, suppose that $Q \in \mathbb{R}^{d \times k}$ is a matrix with orthonormal columns and $P \in \mathbb{R}^{k \times k}$ an upper triangular matrix where $QP = M$ is the QR decomposition of $M$. We take QR$(\cdot)$ to be the function that maps $M$ to $(Q, P)$. Finally, given a subspace $B$, we denote its orthogonal subspace as $B_\perp$.

**Definition 1.** *Let $X \in \mathbb{R}$ be a random variable. We call $X$ a centered $R$-**sub-Gaussian** if $X$ satisfies $\mathbb{E}\left[e^{\lambda X}\right] \leq e^{\lambda^2 R^2 / 2}$ for all $\lambda \in \mathbb{R}$. Furthermore, we denote the distribution of $X$ as $\mathrm{SG}(R^2)$.*

We now introduce a quantity that is commonly used in matrix completion (Jain et al., 2013) and shared representation learning (Jain et al., 2021; Collins et al., 2021) as a measure of distance between subspaces of $\mathbb{R}^d$. It is conceptually useful to think of this distance as measuring the dissimilarity of span and alignment between two subspaces.

**Definition 2.** *We define the **principal angle** between the column spaces of matrices $M_1, M_2 \in \mathbb{R}^{d \times k}$ via* $\mathrm{dist}(M_1, M_2) = \left\| \hat{M}_{1,\perp}^\top \hat{M}_2 \right\|_2$ *where $\hat{M}_{1,\perp}^\top, \hat{M}_2$ are matrices with orthonormal columns that satisfy* $\mathrm{span}(\hat{M}_{1,\perp}^\top) = \mathrm{span}(M_1)_\perp$ *and* $\mathrm{span}(\hat{M}_2) = \mathrm{span}(M_2)$.

Note that we can instantiate the matrix $\hat{M}_{1,\perp}$ with $I_{d \times d} - \hat{M}_1 \hat{M}_1^\top$. This is used many times in our analysis. We show how to bound the excess risk with the principal angle in Lemma 35 in Appendix B.

**Definition 3** (User-level Differential Privacy (Dwork et al., 2006))**.** *A randomized algorithm $\mathcal{A}$ is $(\epsilon, \delta)$-**user-level differentially private** (**user-level DP**) if for any pair of neighboring collections of datasets $S, S' \in \mathbb{R}^{nm(d+1)}$ differing in a single user dataset and any event $B$ in the output range of $\mathcal{A}$, we have*

$$\mathbb{P}\left[\mathcal{A}(S) \in B\right] \leq e^\epsilon \mathbb{P}\left[\mathcal{A}(S') \in B\right] + \delta.$$

**Billboard Model.** In this paper, we adopt the billboard model of differential privacy (Hsu et al., 2014; Kearns et al., 2014). In this model, there are $n$ users and a central computing server. The server first runs a differentially private algorithm using the sensitive data from the users. The output of the algorithm is then broadcast to all users. Each user will independently train their own model using the broadcast output and the user's own data. The results of these individual computations will be kept to each user and never shared with other users.

**Personalization via Representation Learning.** We denote the the optimal parameters of user $i$ as $w_i^*$, and we assume all user parameters $\{w_1^*, \ldots w_n^*\}$ share a low dimensional embedding matrix $U^* \in \mathbb{R}^{k \times d}$ with $k \ll d$ such that $w_i^* = U^* v_i^*$ with a local vector $v_i^* \in \mathbb{R}^k$ for $i \in [n]$. Given a dataset collection $S = (S_1, \ldots S_n)$, the goal is to privately find a shared low dimensional embedding matrix $U \in \mathbb{R}^{d \times k}$ and local vectors $V = [v_1, \ldots v_n]^\top$ with low excess risk compared with $U^*$ and $V^* = [v_1^*, \ldots v_n^*]^\top$. Throughout this work, we assume $U$ has orthonormal columns, hence the shared matrix $U$ effectively maps $d$-dimensional feature vectors $x \in \mathbb{R}^d$ to low-dimensional representations $U^\top x \in \mathbb{R}^k$ that ease local demands on each user. Given parameter spaces $\mathcal{U}, \mathcal{V}$, the objective can be formulated as a minimization problem written as $\min_{U \in \mathcal{U}, V \in \mathcal{V}} L(U, V; \mathcal{D})$.

## 3 Private FedRep Algorithm

In this section, we propose an efficient federated private model personalization algorithm based on the FedRep algorithm in (Collins et al., 2021) for regression problems. Our algorithm is based on an alternating minimization framework where the algorithm alternates between the updates of user local vectors and the shared embedding by first finding an empirical minimizer for each user, followed by computing a gradient with respect to the embedding. After we compute the private shared representation, each user will independently solve for their user-specific local model, which will be kept to each user. We provide rigorous utility and privacy guarantees for our algorithm and give a detailed comparison with prior work.

We make several modifications to the original FedRep algorithms and its analysis. First, we sample disjoint data batches in each iteration to update the local vectors and shared embedding separately while the original FedRep uses the same data batches for both. This use of disjoint data batches enables us to give a tighter bound on the Frobenius norm of the computed gradients, which in turn reduces the amount of noise required for privacy preservation. Moreover, we consider the setting of noisy labels, as opposed to the original FedRep analysis. Incorporating label noise introduces unique technical challenges into the analysis. Roughly speaking, without label noise, we can show the distance between the computed embedding with the ground truth decreases geometrically with high probability across iterations. However, such monotonic distance decrease is no longer guaranteed in the presence of label noise. Instead, we employ an induction-based argument to upper bound the distance.

Before we introduce the Private FedRep algorithm, we first give some definitions and assumptions that hold for the entire section.

## 3.1 Problem Setting and Assumptions

In this section, we focus on quadratic loss defined $\ell(U, v, (x, y)) = (y - \langle x, Uv \rangle)^2$ for all $U \in \mathbb{R}^{d \times k}, v \in \mathbb{R}^k, (x, y) \in \mathbb{R}^{d+1}$. Define $\Gamma = \max_{i \in [n]} \|v_i^*\|_2$. Denote $\sigma_{\min,*} = \sigma_{\min} \left( \frac{1}{\sqrt{n}} V^* \right)$ and $\sigma_{\max,*} = \sigma_{\max} \left( \frac{1}{\sqrt{n}} V^* \right)$, respectively. In our setting, we assume that the rank of $\frac{1}{\sqrt{n}} V^* \in \mathbb{R}^{n \times k}$ is $k$, which implies $\sigma_{\min,*}$, the $k$-th largest singular value of $\frac{1}{\sqrt{n}} V^*$, is greater than 0. The values $\Gamma, \sigma_{\max,*}$, and $\sigma_{\min,*}$ are important in this setting and appear in our final bounds. We additionally define the condition number to be $\gamma = \frac{\sigma_{\max,*}}{\sigma_{\min,*}}$.

The guarantees of our algorithm will rely on the following assumptions.

**Assumption 4** (Client Diversity). *There are known $\Lambda, \lambda > 0$ such that $\Lambda \geq \sigma_{\max,*} \geq \sigma_{\min,*} \geq \lambda$.*

The assumption that $\sigma_{\min,*} > 0$ is made explicitly or implicitly in prior work (Collins et al., 2021; Jain et al., 2021). Observe that Assumption 4 implies $\gamma \leq \frac{\Lambda}{\lambda}$. In this sense, $\frac{\Lambda}{\lambda}$ is an estimate of the condition number $\gamma$.

To make our analysis tractable, we will introduce some standard data assumptions similar to those invoked in prior works (Collins et al., 2021; Du et al., 2020; Jain et al., 2021; Thekumparampil et al., 2021; Tripuraneni et al., 2021). Although we make these assumptions, our algorithm can be used for a wider range of data sources than what is shown here.

**Assumption 5.** *Let $i \in [n]$ and $x \sim \mathcal{D}_i$. For all $i \in [n]$ the feature distribution $\mathcal{D}_i$ has covariance matrix $I_{d \times d}$ and*

*is sub-Gaussian in the sense of $\mathbb{E}\left[e^{\langle x, u \rangle}\right] \leq e^{\frac{\|u\|_2^2}{2}}$ for all $u \in \mathbb{R}^d$.*

**Assumption 6.** *Let $\mathrm{SG}(R)$ be any centered $R$-sub-Gaussian distribution over $\mathbb{R}$. Sample $\zeta \sim \mathrm{SG}(R)$. For all $i \in [n]$ and any data points $(x, y) \sim \mathcal{D}_i$ we generate the label $y$ for the $i$-th user via $y = x^\top U^* v_i^* + \zeta$ where $\zeta$ is independent for all $x \sim \mathcal{D}_i$.*

Unless stated otherwise, in this section, we assume that Assumptions 4, 5, and 6 hold. However, we note that our algorithms do not require knowledge of Assumptions 5 and 6.

## 3.2 Main Algorithm

Algorithm 1 uses alternating minimization for convergence to an approximate minimizer of the excess population risk in parameters $U \in \mathbb{R}^{d \times k}$ with orthonormal columns and $V \in \mathbb{R}^{n \times k}$, simultaneously. At each iteration of the inner loop each user samples $b$ data points without replacement from their data set $S_i$. The users then split this sample into disjoint batches $B_{i,t}, B'_{i,t}$ for use in the computing parameters in iteration $t$. Our inner loop uses parallel execution at iteration $t$ to locally compute a minimizer $v_{i,t} \in \arg\min_{v \in \mathbb{R}^k} \widehat{L}(U_t, v; B_{i,t})$ and a gradient $\nabla_{i,t} = \nabla_U \widehat{L}(v_{i,t}, U_t; B'_{i,t})$ for each user $i$. The server then aggregates clipped gradients $\{\mathsf{clip}(\nabla_{i,t}, \psi)\}_{t \in [T]}$ for its gradient step and then returns $U_{t+1}$ to the users where $\mathsf{clip}(M, \tau) = \frac{M}{\|M\|_F} \cdot \min\left\{1, \frac{\tau}{\|M\|_F}\right\}$ for any $M \in \mathbb{R}^{d \times k}$ and $\tau \in \mathbb{R}$.

Each iteration Algorithm 1 ensures that $U_{t+1}$ has orthonormal columns. Due to orthonormality, the noise injected during the aggregation of $(\nabla_{i,t})_{i \in [n]}$ does not affect the norm of $\nabla_{i,t+1}$ for each $i, t$. This lack of noise propagation, $\mathcal{D}_i^x$ having a covariance matrix $I_{d \times d}$, and the use of disjoint batches $B_{i,t}, B'_{i,t}$ ensure the following bound on the size of $\nabla_{i,t}$.

**Lemma 7.** *With probability at least $1 - O(T \cdot n^{-10})$, we have*

$$\|\nabla_{i,t}\|_F \leq \widetilde{O}\left((R + \Gamma)\Gamma\sqrt{dk}\right)$$

*for all $i \in [n]$ and all $t \in [T]$ simultaneously.*

We use Lemma 7 to choose the clipping parameter $\psi$ in Algorithm 1 in later results.

**Theorem 8.** *Algorithm 1 is $(\epsilon, \delta)$-user-level DP in the billboard model by setting $\hat{\sigma} = C \frac{\psi \sqrt{T \log(1/\delta)}}{n\epsilon}$ for some absolute constant $C > 0$.*

We define $\Delta_{\epsilon, \delta} := C \frac{\sqrt{\log(1/\delta)}}{\epsilon}$ for any $\epsilon > 0$ and $\delta \in [0, 1]$.

For the following result, recall the definition of $\gamma = \frac{\sigma_{\max,*}}{\sigma_{\min,*}}$ and Assumption 4, where we assume there exists $\lambda > 0$ such that $\sigma_{\min,*} \geq \lambda$.

**Algorithm 1 Private FedRep** for linear regression

**Require:** $S_i = \{(x_{i,1}, y_{i,1}), \ldots, (x_{i,m}, y_{i,m})\}$ data for users $i \in [n]$, learning rate $\eta$, iterations $T$, privacy noise parameters $\hat{\sigma}$, clipping parameters $\psi, \psi_{\text{init}}$, batch size $b \leq \lfloor m/2T \rfloor$, initial embedding $U_{\text{init}} \in \mathbb{R}^{d \times k}$
Let $S_i^0 \leftarrow \{(x_{i,j}, y_{i,j}) : j \in [m/2]\} \quad \forall i \in [n]$
Let $S_i^1 \leftarrow S_i \setminus S_i^0 \quad \forall i \in [n]$

1: **Initialize:** $U_0 \leftarrow U_{\text{init}}$
2: **for** $t = 0, \ldots, T-1$ **do**
3:     Server sends $U_t$ to clients $[n]$
4:     **for** Clients $i \in [n]$ in parallel **do**
5:         Sample two disjoint batches $B_{i,t}$ and $B'_{i,t}$, each of size $b$, without replacement from $S_i^0$
6:         Update $v_i$ as $v_{i,t} \leftarrow \arg\min_{v \in \mathbb{R}^k} \widehat{L}(U_t, v; B_{i,t})$
7:         Compute the gradient w.r.t $U$

$$\nabla_{i,t} \leftarrow \nabla_U \widehat{L}(U_t, v_{i,t}; B'_{i,t})$$

8:         Send $\nabla_{i,t}$ to server
9:     **end for**
10:     Server aggregates the client gradients as

$$\hat{U}_{t+1} \leftarrow U_t - \eta \left( \frac{1}{n} \sum_{i=1}^{n} \mathsf{clip}(\nabla_{i,t}, \psi) + \xi_{t+1} \right)$$

$$U_{t+1}, P_{t+1} \leftarrow \mathsf{QR}(\hat{U}_{t+1})$$

    where $\xi_{t+1} \leftarrow \mathcal{N}^{d \times k}(0, \hat{\sigma}^2)$
11: **end for**
12: Server sends $U^{\text{priv}} \leftarrow U_T$ to all clients
13: **for** clients $i \in [n]$ independently **do**
14:     $v_i^{\text{priv}} \leftarrow \arg\min_{v \in \mathbb{R}^k} \widehat{L}(U^{\text{priv}}, v; S_i^1) \quad \forall i \in [n]$
15: **end for**
16: **Return:** $U^{\text{priv}}, V^{\text{priv}} \leftarrow [v_1^{\text{priv}}, \ldots, v_n^{\text{priv}}]^\top$

---

**Lemma 9.** *Let $\eta \leq \frac{1}{2\sigma_{\max,*}^2}$. Suppose $1 - \text{dist}^2(U_0, U^*) \geq c$ for some constant $c > 0$. Set the clipping parameter $\psi = \widetilde{O}\left((R+\Gamma)\Gamma\sqrt{dk}\right)$. Assume $T = \frac{\log n}{\eta\lambda^2} = \widetilde{O}\left(\min\left(\frac{nm\lambda^2}{\max\{\Delta_{\epsilon,\delta}, 1\}(R+\Gamma)\Gamma d\sqrt{k}m + R^2\Gamma^2 d\sigma_{\min,*}^2}, \frac{m\Gamma^2\sigma_{\max,*}^2}{\max\{R^2, 1\} \cdot \max\{\Gamma^2, 1\}\gamma^4 k\Gamma^2 + R^2 k\sigma_{\max,*}^2}\right)\right)$ and Assumptions 5 and 6 hold for all user data. Set $\hat{\sigma}$ as in Theorem 8 and batch size $b = \lfloor m/2T \rfloor$. Then, $U^{\text{priv}}$, the first output of Algorithm 1, satisfies*

$$\text{dist}(U^{\text{priv}}, U^*) \leq \left(1 - \frac{c\eta\sigma_{\min,*}^2}{4}\right)^{\frac{T}{2}} \text{dist}(U_0, U^*)$$

$$+ \widetilde{O}\left(\frac{(R+\Gamma)\Gamma d\sqrt{kT}}{n\epsilon\sigma_{\min,*}^2} + \sqrt{\frac{(R^2+\Gamma^2)\Gamma^2 dT}{nm\sigma_{\min,*}^4}}\right)$$

*with probability at least $1 - O(T \cdot n^{-10})$.*

Note that $\text{dist}(U_0, U^*)$ in the right-hand side of the bound in Lemma 9 is bounded from above by $\sqrt{1-c}$.

Given Lemma 9, we may obtain our main excess population risk bound. Let $U^{\text{priv}}$ be the final shared embedding returned from Algorithm 1. The intuition behind this result is that a small principal angle between the two matrices $U^{\text{priv}}, U^*$ implies $U^{\text{priv}}$ approximates $U^*$ as a transformation $\mathbb{R}^d \to \mathbb{R}^k$. Our choice of $v_i^{\text{priv}} \in \arg\min_{v \in \mathbb{R}^k} \widehat{L}(U^{\text{priv}}, v; S_i^1)$ for $S_i^1$ independent of $U^{\text{priv}}$ means $w_i = U^{\text{priv}} v_i^{\text{priv}}$ reliably transforms a fresh feature vector $x \sim \mathcal{D}_i^x$ into a scalar $x^\top w_i$ that is close to the noisy label $x^\top U^* v_i^* + \zeta$ when $n, m$ are sufficiently large.

Recall in Assumption 4 we assume there exist $\Lambda, \lambda > 0$ such that $\Lambda \geq \sigma_{\max,*} \geq \sigma_{\min,*} \geq \lambda$.

**Theorem 10.** *Suppose all conditions of Lemma 9 hold with $\eta = \frac{1}{2\Lambda^2}$, $T = \Theta\left(\frac{\Lambda^2 \log(n^3)}{\lambda^2}\right)$, and that Assumption 4 holds as well. Then, $U^{\text{priv}}$ and $V^{\text{priv}}$, the outputs of Algorithm 1, satisfy*

$$L(U^{\text{priv}}, V^{\text{priv}}; \mathcal{D}) - L(U^*, V^*; \mathcal{D})$$

$$\leq \widetilde{O}\left(\frac{(R^2+\Gamma^2)\Gamma^4\Lambda^2 d^2 k}{n^2\epsilon^2\sigma_{\min,*}^4\lambda^2} + \frac{(R^2+\Gamma^2)\Gamma^4\Lambda^2 d}{nm\sigma_{\min,*}^4\lambda^2}\right)$$

$$+ \frac{R^2 k}{m}$$

*with probability at least $1 - O(T \cdot n^{-10})$. Furthermore, Algorithm 1 is $(\epsilon, \delta)$-user-level DP in the billboard model.*

Note our results in Theorem 10 can also be extended to the case where new clients share the same embedding $U^*$. This is formally presented in the following corollary.

**Corollary 11.** *Suppose all conditions of Theorem 10 hold. Consider a new client with a dataset $S_{n+1} \sim \mathcal{D}_{n+1}^{m'}$, where Assumptions 5 and 6 hold with $\|v_{n+1}^*\|_2 \leq \Gamma$. Let $U^{\text{priv}}$ be the output of Algorithm 1 and $v_{n+1}^{\text{priv}} = \arg\min_{v \in \mathbb{R}^k} \widehat{L}(U^{\text{priv}}, v; S_{n+1})$, if $m' \gtrsim k \log m'$. We have*

$$L(U^{\text{priv}}, v_{n+1}^{\text{priv}}; \mathcal{D}_{n+1}) - L(U^*, v_{n+1}^*; \mathcal{D}_{n+1})$$

$$\leq \widetilde{O}\left(\frac{(R^2+\Gamma^2)\Gamma^4\Lambda^2 d^2 k}{n^2\epsilon^2\sigma_{\min,*}^4\lambda^2} + \frac{(R^2+\Gamma^2)\Gamma^4\Lambda^2 d}{nm\sigma_{\min,*}^4\lambda^2}\right)$$

$$+ \frac{R^2 k}{m'}$$

*with probability at least $1 - O(T \cdot n^{-10} + m'^{-100})$.*

**Initialization.** We require $\text{dist}(U_0, U^*)$ in Lemma 9 to be bounded away from 1 otherwise the bound is trivial since the first term becomes 1 when $c = 0$. By definition of the principal angle (2), $\text{dist}(U_0, U^*) = 1$ when the column space of $U_0$ and $U^*$ are orthogonal. For now, we assume

such a $U_0$ whose column space overlaps modestly with that of $U^*$ in the sense of $\text{dist}(U_0, U^*) < \sqrt{1-c}$ can be found with reasonable effort. In the next subsection we give an efficient algorithm that guarantees such an initialization.

### 3.3 Private Initialization Algorithm

In this section, we introduce an initialization algorithm for our private FedRep algorithm (Algorithm 1). Our initialization algorithm is based on the estimator in (Duchi et al., 2022) and the privacy guarantee is achieved by adding Gaussian noise to the estimator before using top-$k$-SVD. Our private initialization algorithm achieves the same convergence rate as the one in (Jain et al., 2021). But unlike (Jain et al., 2021), which is limited to i.i.d. standard Gaussian feature vectors, the guarantee for our algorithm holds in a broader class of sub-Gaussian feature vectors. The detailed steps of the algorithm are provided in Algorithm 2, while its privacy and utility guarantees are established in Lemma 12.

---

**Algorithm 2 Private Initialization** for Private FedRep

---

**Require:** $S_i = \{(x_{i,1}, y_{i,1}), \ldots, (x_{i,m/2}, y_{i,m/2})\}$ data for users $i \in [n]$, privacy parameters $\epsilon, \delta$, clipping bound $\psi_{\text{init}}$, rank $k$

1: Let $\hat{\sigma}_{\text{init}} \leftarrow \frac{\psi_{\text{init}} \sqrt{2 \log\left(\frac{1.25}{\delta}\right)}}{n\epsilon}$

2: Let $\xi_{\text{init}} \leftarrow \mathcal{N}^{d \times d}(\vec{0}, \hat{\sigma}_{\text{init}}^2)$

3: **for** Clients $i \in [n]$ in parallel **do**

4:     Send $Z_i \leftarrow \frac{2}{m(m-1)} \sum_{j_1 \neq j_2} y_{i,j_1} y_{i,j_2} x_{i,j_1} x_{i,j_2}^{\top}$ to server

5: **end for**

6: Server aggregates $Z_i$ and add noise for privatization

$$\hat{Z} = \frac{1}{n} \sum_{i=1}^{n} \text{clip}(Z_i, \psi_{\text{init}}) + \xi_{\text{init}}$$

7: Server computes

$$U_{\text{init}} D U_{\text{init}}^{\top} \leftarrow \text{rank-}k\text{-SVD}(\hat{Z})$$

8: **Return:** $U_{\text{init}}$

---

**Lemma 12.** *Suppose that Assumptions 5 and 6 hold. Let $U_{\text{init}}$ be the output of Algorithm 2. Then, by setting $\psi_{\text{init}} = \widetilde{O}((R^2 + \Gamma^2)d)$, we have*

$$\text{dist}(U_{\text{init}}, U^*) \leq \widetilde{O}\left(\frac{(R^2 + \Gamma^2)d^{3/2}}{n\epsilon\sigma_{\min,*}^2} + \sqrt{\frac{(R^2 + \Gamma^2)\Gamma^2 d}{mn\sigma_{\min,*}^4}}\right)$$

*with probability at least $1 - O(n^{-10})$. Furthermore, Algorithm 2 is $(\epsilon, \delta)$ user-level DP.*

By using Lemma 12 in conjunction with Theorem 10 and an iteration count $T = \Theta\left(\frac{\Lambda^2 \log\left(R^2 d/k\right)}{\lambda^2}\right)$, we are able to

achieve the same utility guarantee as Theorem 10 without any assumptions on $\text{dist}(U_{\text{init}}, U^*)$.

**Comparison with (Jain et al., 2021):** Under conventional assumptions in prior work (Tripuraneni et al., 2021) to normalize $V^*$ and $\text{SG}(R)$ so that both $\Gamma$ and $R$ are $\widetilde{\Theta}(1)$, the Priv-AltMin algorithm in (Jain et al., 2021) achieves an excess risk bound of $\widetilde{O}\left(\frac{d^3 k^2}{n^2 \epsilon^2 \sigma_{\min,*}^4} + \frac{d}{nm\sigma_{\min,*}^4}\right) + \frac{k}{m}$ for Gaussian data. Meanwhile, our Private FedRep algorithm achieves a rate of $\widetilde{O}\left(\frac{d^2 k \Lambda^2}{n^2 \epsilon^2 \sigma_{\min,*}^4 \lambda^2} + \frac{d\Lambda^2}{nm\sigma_{\min,*}^4 \lambda^2}\right) + \frac{k}{m}$. When $\frac{\Lambda}{\lambda}$ (an upper bound on the condition number of $\frac{1}{\sqrt{n}} V^*$) is $\widetilde{O}(1)$, which is a regime considered in (Jain et al., 2021) and assumed in (Collins et al., 2021), we improve the privacy error term in (Jain et al., 2021) by a factor of $\widetilde{O}(dk)$.

More generally, in the regime where $n = \widetilde{O}\left(\min\left(\frac{dkm}{\epsilon^2}, \sqrt{\frac{d^3 km}{\epsilon^2 \sigma_{\min,*}^4}}, \sqrt{\frac{d^2 m \Lambda^2}{\epsilon^2 \sigma_{\min,*}^4 \lambda^2}}\right)\right)$, our bound is tighter than the bound of (Jain et al., 2021) by a factor of $\widetilde{O}\left(\frac{dk\lambda^2}{\Lambda^2}\right)$ when $\frac{\Lambda}{\lambda} = \widetilde{O}(\sqrt{dk})$. Meanwhile, in the regime where $n = \widetilde{O}\left(\min\left(\frac{d^2 k^2 m}{\epsilon^2}, \sqrt{\frac{d^3 km}{\epsilon^2 \sigma_{\min,*}^4}}, \frac{d\Lambda^2}{k\sigma_{\min,*}^4 \lambda^2}\right)\right)$ and $n = \widetilde{\Omega}\left(\frac{dkm}{\epsilon^2}\right)$, we achieve a bound tighter than that of (Jain et al., 2021) by a factor of $\widetilde{O}\left(\frac{d^2 k^2 m \lambda^2}{n\epsilon^2 \Lambda^2}\right)$ when $\frac{\Lambda}{\lambda} = \widetilde{O}\left(\frac{dk}{\epsilon} \sqrt{\frac{m}{n}}\right)$.

Additionally, our work improves over that of (Jain et al., 2021) in two important respects. First, it requires less restrictive data assumptions (sub-Gaussian instead of Gaussian feature vectors). Second, our algorithm can be naturally written as a federated algorithm while the algorithm in (Jain et al., 2021) requires centralized processing and transforming it to a federated algorithm is infeasible.

### 3.4 Synthetic Data Experiment

The results in Figure 1 are obtained via data features from $\mathcal{N}(0, I_d)$ with problem parameters $n = 20,000$, $d = 50$, $k = 2$, and $m = 10$. Our data labels are generated as in Assumption 6 given label noise sampled from $\mathcal{N}(0, R^2)$ with $R = 0.01$. We use local GD and non-private FedRep as baselines for our comparison. See Appendix B.4 for details[1].

## 4 Margin-based Dimension Reduced Construction for Classification

We now introduce our information-theoretic guarantees for private representation learning for classification problem with margin loss. Our approach is based on the observation that learning a shared linear representation can be framed as

---

[1]Note as well this GitHub repository with a copy of our code.

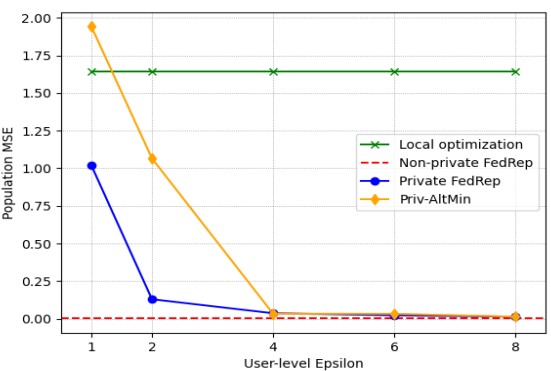

Figure 1: Graph of population MSE over choice of privacy parameter $\epsilon \in [1, 8]$ for synthetic data comparing Algorithm 1 to Priv-AltMin in (Jain et al., 2021).

training a 2-layer linear neural network, enabling the use of the Johnson-Lindenstrauss (JL) transform (Johnson, 1984) for dimensionality reduction.

We demonstrate this utility guarantee as follows. First, we introduce the assumptions and definitions that hold for this section. This includes the definition for the Johnson-Lindenstrauss transform along with a randomized instantiation. Subsequently, we describe the score function and covering argument required to combine dimension reduction with the exponential mechanism. We then introduce Algorithm 3 that exploits the margin loss to attain our main result of the section, Theorem 17. Finally, we discuss the relevance of the main result and its independence from the data dimension $d$.

Let $k' \ll d$. By applying a linear transformation of the data via an appropriately chosen matrix $M \in \mathbb{R}^{k' \times d}$, we show that learning personalized parameters can support further dimension reduction. This is done by first learning $\widetilde{U} \in \mathbb{R}^{k' \times k}$ over the linearly transformed data in $\mathbb{R}^{k'}$ then constructing the final shared embedding $M^\top \widetilde{U} \in \mathbb{R}^{d \times k}$. We obtain utility guarantees independent of the data dimension $d$ via this process of first learning a low-dimensional embedding then mapping it to $\mathbb{R}^{d \times k}$.

In this section, we study the classification setting. Let $\mathcal{X}$ be the $d$-dimensional Euclidean ball with constant radius $r > 0$ centered at the origin and $\mathcal{Y} = \{-1, 1\}$. Define that data space $\mathcal{Z} = \mathcal{X} \times \mathcal{Y}$. Let $\mathcal{U} \subset \mathbb{R}^{d \times k}$ be the space of $d \times k$ matrices with orthonormal columns and $\mathcal{V} \subset \mathbb{R}^{n \times k}$ be the space of matrices with rows bounded in Euclidean norm by a universal constant $\Gamma > 0$. The data distributions for our $n$ users $\mathcal{D}_1, \ldots, \mathcal{D}_n$ are possibly distinct distributions over $\mathcal{X} \times \mathcal{Y}$. We further define the distribution sequence $\mathcal{D} = (\mathcal{D}_1, \ldots, \mathcal{D}_n)$.

**Definition 13.** *Let* $(U, v) \in \mathbb{R}^{d \times k} \times \mathbb{R}^k$ *and* $(x, y) \in$ $\mathbb{R}^d \times \{-1, 1\}$ *any data point. We define the **margin loss** as* $\ell_\rho(U, v, z) = \mathbb{1}\left[y\langle x, Uv\rangle \le \rho\right]$ *and denote the **0-1 loss*** $\ell(U, v, z) = \ell_0(U, v, z) = \mathbb{1}\left[y\langle x, Uv\rangle \le 0\right]$.

Let $U \in \mathcal{U}$, $v \in \mathbb{R}^k$, and $S' = (z'_1 \ldots, z'_m) \in \mathcal{Z}^m$. We define $\widehat{L}_\rho(U, v; S') = \frac{1}{m} \sum_{j=1}^m \ell_\rho(U, v, z'_j)$ for any $U, v, S'$. Suppose $V = [v_1, \ldots, v_n]^\top \in \mathcal{V}$ and let $S = (S_1, \ldots, S_n)$ be a sequence of datasets $S_i \in \mathcal{Z}^m$ for all $i \in [n]$. We further define $\widehat{L}_\rho(U, V; S) = \frac{1}{n} \sum_{i=1}^n \widehat{L}_\rho(U, v_i; S_i)$ for all $U, V, S$. Our goal is to privately optimize the population loss $L(U, V; \mathcal{D}) = \frac{1}{n} \sum_{i=1}^n \mathbb{E}_{Z_i \sim \mathcal{D}_i} \left[L(U, v_1, Z_i)\right]$ over $U, V$ with personalization of $V$. We accomplish this via an application of the exponential mechanism over a cover of a dimension-reduced version of $\mathcal{U}$.

**Definition 14.** *Let* $G \subset \mathbb{R}^d$ *be any set of $t$ vectors. Fix* $\tau, \beta \in (0, 1)$. *We call the random matrix* $M \in \mathbb{R}^{k' \times d}$ *a* $(t, \tau, \beta)$-***Johnson-Lindenstrauss (JL) transform** if for any* $u, u' \in G$

$$|\langle Mu, Mu'\rangle - \langle u, u'\rangle| \le \tau \|u\|_2 \|u'\|_2$$

*with probability at least* $1 - \beta$ *over* $M$.

The JL transform is a popular and efficient method of dimension reduction whose existence is ensured by the Johnson-Lindenstrauss lemma (Johnson, 1984; Nelson, 2020; Woodruff et al., 2014). Via a JL transform $M$ we are able to preprocess a feature vector $x$ into a low-dimensional vector $Mx \in \mathbb{R}^{k'}$ for use in our federated learning problem. This preprocessing showcased in Algorithm 3 effectively reduces the dimension of our learning problem.

**Lemma 15.** *(Bassily et al., 2022) Let* $\tau, \beta \in (0, 1)$. *Take* $G \subset \mathbb{R}^d$ *to be any set of $t$ vectors. Setting* $k' = O\left(\frac{\log\left(\frac{t}{\beta}\right)}{\tau^2}\right)$ *for a $k' \times d$ matrix $M$ with entries drawn uniformly and independently from* $\left\{\pm \frac{1}{\sqrt{k'}}\right\}$ *implies that $M$ is a $(t, \tau, \beta)$-JL transform.*

Suppose $M \in \mathbb{R}^{k' \times d}$ is a random matrix with entries drawn uniformly from $\left\{\pm \frac{1}{\sqrt{k'}}\right\}$. Define $\mathcal{U}_M = \{MU : U \in \mathcal{U}\}$. Let $\mathcal{B}_F$ be the $k' \times k$-dimensional Frobenius ball of radius $\sqrt{2k}$. We define $\mathcal{N}^\gamma$ to be a Frobenius norm $\gamma$-cover of $\mathcal{B}_F$. Assuming that $\gamma \le 1$, we have $\mathcal{U}_M \subseteq \mathcal{B}_F$ with high probability. That is, for any $U' \in \mathcal{U}_M$ there exists $\widetilde{U} \in \mathcal{N}^\gamma$ where $\|U' - \widetilde{U}\| \le \gamma$. Moreover, $|\mathcal{N}^\gamma| \le O\left(\left(\frac{\sqrt{k}}{\gamma}\right)^{k' k}\right)$.

Algorithm 3 first takes as input the user data sequence $S$ and score function $f$. It then constructs a JL transform $M$ in the sense of Lemma 15. Algorithm 3 reduces the dimensionality of the users' data in $S$ by applying $M$ to each feature vector in each user's dataset. The exponential mechanism is run over a cover $\mathcal{N}^\gamma$ using a score function $f(\cdot, S_M)$, which will be defined shortly, to obtain a low-dimensional shared

embedding $\widetilde{U} \in \mathbb{R}^{k' \times k}$. This $M^\top \widetilde{U} \in \mathbb{R}^{d \times k}$ is used by Algorithm 3 where the final embedding $U^{\mathrm{priv}} = M^\top \widetilde{U}$ and local vectors $V^{\mathrm{priv}} = [v_1^{\mathrm{priv}}, \ldots, v_n^{\mathrm{priv}}]^\top$ are computed by the users via $U^{\mathrm{priv}}$.

**Remark:** Our algorithm functions as an improper learner in the sense that the learned shared representation $U^{\mathrm{priv}}$ is not necessarily an orthonormal matrix. However, this does not affect the performance of the final user-specific classifiers in terms of expected loss. In particular, Theorem 17 establishes that the expected loss incurred by our algorithm remains close to that of an optimal orthonormal shared representation, ensuring that the lack of orthonormality does not degrade utility.

Assume for simplicity that $m$ is even. We partition $S_i = S_i^0 \cup S_i^1$ where $S_i^0 = \{z_{1,j}, \ldots, z_{\frac{m}{2},j}\}$ and $S_i^1 = \{z_{\frac{m}{2}+1,j}, \ldots, z_{m,j}\}$ for each $i \in [n]$. Further, we denote $S^t = (S_1^t, \ldots, S_n^t)$ where $t \in \{0, 1\}$. Suppose that $S = (S_1, \ldots, S_n) \subset \mathcal{Z}^{nm(d+1)}$ is a sequence of $n$ datasets with $m$ samples each. Let $S_M = ((S_1)_M, \ldots (S_n)_M)$ where $(S_i)_M = \{(Mx_{i,j}, y_{i,j}) : j \in [m]\}$ for all $i \in [n]$. The score function for Algorithm 3 is $f(\widetilde{U}, S_M) = -\frac{1}{n} \sum_{i=1}^{n} \min_{\|v_i\| \leq \Gamma} \widehat{L}_\rho(\widetilde{U}, v_i; (S_i)_M)$ for $\widetilde{U} \in \mathcal{N}^\gamma$. Since we provide a user-level privacy guarantee, we must bound the sensitivity of $f$ over neighboring sequences $S, S'$ where $S'$ differs from $S$ by at most a single user's entire dataset. It can be shown that the sensitivity is bounded $\frac{1}{n}$, which follows easily from the fact that the margin loss is bounded by 1.

---

**Algorithm 3** Private Representation Learning for Personalized Classification

---

**Require:** dataset sequences $S^0$ and $S^1$ of equal size, score function $f(U', \cdot) = -\min_{V \in \mathcal{V}} \widehat{L}_\rho(U', V; \cdot)$ over matrices $U' \in \mathbb{R}^{k' \times k}$, privacy parameter $\epsilon > 0$, target dimension $k' = O\left(\frac{r^2 \Gamma^2 \log(nm/\delta)}{\rho^2}\right)$,

1: Sample $M \in \mathbb{R}^{k' \times d}$ with entries drawn i.i.d uniformly from $\left\{\pm \frac{1}{\sqrt{k'}}\right\}$
2: Let $S_M = ((S_1)_M, \ldots, (S_n)_M)$ where $(S_i)_M = \left\{(Mx, y) : (x, y) \in S_i^0\right\}$ for $i \in [n]$
3: Let $\mathcal{N}^\gamma$ be a Frobenius norm $\gamma$-cover of $\mathcal{B}_F$
4: Run the exponential mechanism over $\mathcal{N}^\gamma$, privacy parameter $\epsilon$, sensitivity $\frac{1}{n}$, and score function $f(U', S_M)$, to select $\widetilde{U} \in \mathcal{N}^\gamma$
5: Let $U^{\mathrm{priv}} \leftarrow M^\top \widetilde{U}$
6: Each user $i \in [n]$ independently computes $v_i^{\mathrm{priv}} \leftarrow \arg\min_{\|v\|_2 \leq \Gamma} \widehat{L}(U^{\mathrm{priv}}, v, S_i^1)$
7: **Return:** $U^{\mathrm{priv}}, V^{\mathrm{priv}} = [v_1^{\mathrm{priv}}, \ldots v_n^{\mathrm{priv}}]^\top$

---

The lemma below is an extension of a fundamental result for the exponential mechanism (McSherry & Talwar, 2007).

Key to obtaining this result is that $\gamma$ is both the error parameter of the JL transform and the radius of the sets in our cover $\mathcal{N}^\gamma$.

**Lemma 16.** *Fix $\epsilon, \rho > 0, \beta \in (0, 1)$. Algorithm 3 is $(\epsilon, 0)$-user-level DP. Sample $S \sim \mathcal{D}^m$. Then, Algorithm 3 returns $U^{\mathrm{priv}}$ from input $S$ such that*

$$\min_{V \in \mathcal{V}} L(U^{\mathrm{priv}}, V; S^0) \leq \min_{(U,V) \in \mathcal{U} \times \mathcal{V}} \widehat{L}_\rho(U, V; S^0)$$
$$+ \widetilde{O}\left(\frac{r^2 \Gamma^2 k}{\epsilon \rho^2 n}\right)$$

*with probability at least $1 - \beta$ over the randomness of $S$ and the internal randomness of the algorithm.*

**Theorem 17.** *Fix $\epsilon, \rho > 0, \beta \in (0, 1)$. Algorithm 3 is $(\epsilon, 0)$-user-level DP in the billboard model. Sample user data $S \sim \mathcal{D}^m$. Then, Algorithm 3 returns $U^{\mathrm{priv}}, V^{\mathrm{priv}}$ from input $S$ such that*

$$L(U^{\mathrm{priv}}, V^{\mathrm{priv}}; \mathcal{D}) \leq \min_{(U,V) \in \mathcal{U} \times \mathcal{V}} \widehat{L}_\rho(U, V; S^0)$$
$$+ \widetilde{O}\left(\frac{r^2 \Gamma^2 k}{n \epsilon \rho^2} + \sqrt{\frac{r^2 \Gamma^2}{m \rho^2}}\right)$$

*with probability at least $1 - \beta$ over the randomness of $S$ and the internal randomness of the algorithm.*

To the best of our knowledge, the above bound is the first margin-based population loss bound that is independent of the data dimension in private personalization with shared embedding.

## 5  Conclusion

In this work, we revisit the problem of model personalization under user-level differential privacy (DP) in the shared representation framework. We propose a novel private federated personalization algorithm that extends the FedRep method while ensuring rigorous privacy guarantees. Our approach efficiently learns a low-dimensional shared embedding and user-specific local models while providing strong privacy-utility trade-offs, even under noisy labels and broader sub-Gaussian data distributions. A key contribution of our work is demonstrating that private model personalization can be achieved in a federated setting with improved risk bounds. Specifically, we show that our approach reduces the privacy error term by a factor of $\widetilde{O}(dk)$ in a natural parameter regime, leading to a higher accuracy in high-dimensional settings. Moreover, our private initialization technique ensures a good starting point for learning shared representations, even under privacy constraints. Additionally, for binary classification, we provide an information-theoretic construction for private model personalization that leverages dimensionality reduction techniques, and hence, derive margin-based dimension-independent risk bound.

While our work provides significant advancements in private federated personalization, several open directions remain for future research: *(i) Relaxing the identity covariance assumption:* Current results assume sub-Gaussian feature distributions with identity covariance. While this assumption simplifies the analysis and aligns with prior work, it does not hold in many real-world applications. A key challenge is extending our methods to handle arbitrary covariance matrices. This could involve leveraging adaptive preconditioning techniques or covariance-aware mechanisms to improve learning under more general feature distributions. *(ii) Extending beyond quadratic loss functions:* Most of efficient private personalization algorithms, including ours, rely on quadratic loss (i.e., least-squares regression) due to its analytical convenience. Developing gradient-based methods for more general loss functions, such as logistic or hinge losses, remains an important open problem. Current techniques for private optimization in these settings are either computationally expensive or lack strong utility guarantees.

## Acknowledgment

The authors' research is supported by NSF Award 2112471 and NSF CAREER Award 2144532.

## Impact Statement

This work advances the theoretical and algorithmic foundations of privacy-preserving personalized learning, a key challenge in the deployment of trustworthy machine learning systems. In domains such as healthcare, education, and mobile applications, users' data are often sensitive and highly personalized—making it critical to ensure privacy while delivering individualized models.

Our contributions demonstrate that personalization and strong privacy guarantees can coexist in federated learning systems, without compromising accuracy. The proposed algorithms address limitations in prior work by supporting a broader range of user data distributions and operating in realistic federated environments. Importantly, we show that the accuracy of learned models can remain robust even in high-dimensional settings or under label noise, which frequently arise in practice.

By enabling theoretically sound, privacy-preserving personalization, this research can help guide the design of secure and reliable machine learning solutions across a wide array of socially impactful applications.

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

# Appendix

## A   Technical Lemmas

**Proposition 18.** *Let $X = (X_1, \ldots, X_a) \in \mathbb{R}^a$ be random vector. Assume that $X_p$ is $R_p$-sub-Gaussian for each $p \in [a]$ and that $X_p, X_q$ are uncorrelated for $p \neq q$. Let $R = (R_1, \ldots, R_a)$. Then, for each $\nu \in \mathbb{R}^a$, the random variable $\langle \nu, X \rangle$ is $(\|\nu\|_2 \max_{p \in [a]} R_a)$-sub-Gaussian.*

*Proof.* Fix an arbitrary $\lambda \in \mathbb{R}$. Observe

$$
\mathbb{E}\left[e^{\lambda \langle \nu, X \rangle}\right] = \mathbb{E}\left[\prod_{p=1}^{a} e^{(\lambda \nu_p) X_p}\right]
$$

$$
= \prod_{p=1}^{a} \mathbb{E}\left[e^{(\lambda \nu_p) X_p}\right]
$$

$$
\leq \prod_{p=1}^{a} e^{\lambda^2 \nu_p^2 R_p^2}
$$

$$
\leq e^{\lambda^2 \|\nu\|_2^2 \max_{p \in [a]} R_a^2}
$$

where the second equality holds by uncorrelatedness, the first inequality from the definition of sub-Gaussian, and the final inequality from definition of $\|\nu\|_2^2$. $\qquad\square$

**Proposition 19.** *Let $X = (X_1, \ldots, X_a) \in \mathbb{R}^a$ be random vector with $\mathbb{E}[X] = \vec{0}$. Assume $X_p, X_q$ are uncorrelated for $p \neq q$. Suppose that $\nu, \pi \in \mathbb{R}^a$ with $\langle \nu, \pi \rangle = 0$. Then, $\langle \nu, X \rangle$ and $\langle \pi, X \rangle$ are uncorrelated.*

*Proof.* By the assumptions of the above proposition

$$
\mathbb{E}\left[\langle \nu, X \rangle \langle \pi, X \rangle\right] = \mathbb{E}\left[\sum_{p=1}^{a} \nu_p X_p \sum_{p=1}^{a} \pi_p X_p\right]
$$

$$
= \mathbb{E}\left[\sum_{p,q \in [a]} \nu_p \pi_q X_p X_q\right]
$$

$$
= \mathbb{E}\left[\sum_{p=1}^{a} \nu_p \pi_p X_p^2\right] + \mathbb{E}\left[\sum_{p \neq q} \nu_p \pi_q X_p X_q\right]
$$

$$
= \sigma^2 \sum_{p=1}^{a} \nu_p \pi_p + \sum_{p \neq q} \nu_p \pi_q \mathbb{E}[X_p] \mathbb{E}[X_q]
$$

$$
= 0
$$

$$
= \mathbb{E}[\langle \nu, X \rangle] \cdot \mathbb{E}[\langle \pi, X \rangle]
$$

where the last three equalities hold by $\langle \nu, \pi \rangle = 0$, the uncorrelatedness of the components of $X$, and the linearity of expectation. $\qquad\square$

**Lemma 20** (McDiarmid's inequality). *Let $f : \mathbb{R}^n \to \mathbb{R}$ be a measurable function. Assume there exists a constant $c_i > 0$ where, for all $x_1, \ldots, x_n \in \mathbb{R}$*

$$
\sup_{x_i' \in \mathbb{R}} |f(x_1, \ldots, x_{i-1}, x_i, x_{i+1}, \ldots, x_n) - f(x_1, \ldots, x_{i-1}, x_i', x_{i+1}, \ldots, x_n)| \leq c_i
$$

*for each $i \in [n]$. Suppose $X_1, \ldots, X_n \in \mathbb{R}$ is a sequence of i.i.d. random variables. Then, for all $s > 0$*

$$
\mathbb{P}\left[f(X_1, \ldots, X_n) - \mathbb{E}[f(X_1, \ldots, X_n)] \geq s\right] \leq e^{-\frac{2s^2}{\sum_{i=1}^{n} c_i}}.
$$

**Definition 21.** *We say that a random variable $X \in \mathbb{R}$ is R-**sub-exponential** if*

$$\mathbb{E}\left[e^{\lambda|X|}\right] \le e^{\lambda R}$$

*for all $\lambda$ such that $0 \le \lambda \le 1/R$.*

For the definition above, we call the smallest such $R$ the sub-exponential norm of an $R$-sub-exponential random variable.

**Lemma 22** (Bernstein inequality). *Let $X_1, \ldots, X_n \in \mathbb{R}^d$ be a sequence of i.i.d. R-sub-exponential random variables. Then, for any $s > 0$*

$$\mathbb{P}\left[\frac{1}{n}\sum_{i=1}^{n} X_i \ge s\right] \le e^{-cn\min\left(\frac{s^2}{R^2}, \frac{s}{R}\right)}$$

*for some constant $c > 0$.*

**Lemma 23** (Lemma 4.4.1 (Vershynin, 2018)). *Let $M \in \mathbb{R}^{a \times b}$ be any matrix. Take $\mathcal{S}^{a-1}, \mathcal{S}^{b-1}$ to be the unit spheres in $\mathbb{R}^a, \mathbb{R}^b$, respectively. Suppose that $\mathcal{N}^a, \mathcal{N}^b$ are Euclidean $\epsilon$-covers over $\mathcal{S}^{a-1}, \mathcal{S}^{b-1}$. Then*

$$\|M\|_2 \le \frac{1}{1-\epsilon} \max_{v \in \mathcal{N}^a, v' \in \mathcal{N}^b} v^\top M v'.$$

**Definition 24.** *A centered random vector $X \in \mathbb{R}^a$ with covariance matrix $\Sigma$ is **isotropic** if*

$$\Sigma = \mathbb{E}\left[XX^\top\right] = I_a.$$

**Definition 25.** *A centered random vector $X \in \mathbb{R}^a$ is R-sub-Gaussian if*

$$\mathbb{E}\left[e^{\langle X,u\rangle}\right] \le e^{\frac{R^2\|u\|_2^2}{2}}$$

*for all $u \in \mathbb{R}^a$.*

Note that when a centered vector $X \in \mathbb{R}^a$ is 1-sub-Gaussian and has covariance matrix $I_a$, the components of $X$ are 1-sub-Gaussian. We use this fact in our results in conjunction with Assumption 5.

**Theorem 26** (Theorem 4.6.1 (Vershynin, 2018)). *Let $M$ be an $a \times b$ matrix whose rows are independent mean-zero K-sub-Gaussian isotropic random vectors in $\mathbb{R}^b$. Then, for any $\alpha \ge 0$, there exists $c > 0$ where*

$$\sqrt{a} - cK^2\left(\sqrt{b} + \alpha\right) \le \sigma_{\min}(M) \le \sigma_{\max}(M) \le \sqrt{a} + cK^2\left(\sqrt{b} + \alpha\right)$$

*with probability $1 - 2e^{-\alpha^2}$.*

**Corollary 27.** *Let $M$ be an $a \times b$ matrix whose rows are independently drawn from $\mathcal{N}(0, \hat{\sigma}^2 I_{a \times a})$. Then, for any $\alpha \ge 0$, we have*

$$\hat{\sigma}\left(\sqrt{a} - \left(\sqrt{b} + \alpha\right)\right) \le \sigma_{\min}(M) \le \sigma_{\max}(M) \le \hat{\sigma}\left(\sqrt{a} + \left(\sqrt{b} + \alpha\right)\right)$$

*with probability at least $1 - 2e^{-\alpha^2}$.*

*Proof.* Observe that we have $M = \hat{\sigma}\bar{M}$ for a matrix $\bar{M} \sim \mathcal{N}^{a \times b}(0, 1)$. As well, note that $\bar{M}$ satisfies Theorem 26 with $K = 1$ since its rows are standard Gaussians.

For any matrix $A \in \mathbb{R}^{a \times b}$, we also have that $\sigma_p(c'A) = c'\sigma_p(A)$ for $c' > 0$ a constant and $\sigma_p(A)$ the $p$-th singular value of $A$. Combining the above reasoning and Theorem 26, there exists a constant $c > 0$ such that

$$\sqrt{a} - c\left(\sqrt{b} + \alpha\right) \le \sigma_{\min}(\bar{M}) \le \sigma_{\max}(\bar{M}) \le \sqrt{a} + c\left(\sqrt{b} + \alpha\right)$$

implies

$$\hat{\sigma}\left(\sqrt{a} - c\left(\sqrt{b} + \alpha\right)\right) \le \sigma_{\min}(M) \le \sigma_{\max}(M) \le \hat{\sigma}\left(\sqrt{a} + c\left(\sqrt{b} + \alpha\right)\right)$$

with probability at least $1 - 2e^{-\alpha^2}$. Furthermore, for standard Gaussians we have $c = 1$. $\qquad\square$

Note that one could simply apply Theorem 26 directly to the matrix $M$ from Corollary 27 by using $K = \hat{\sigma}$, but this does not give the desired result. In particular, we require that $\hat{\sigma}$ scale both terms of the upper bound of $\sigma_{\max}(M)$ in our later analysis of the Gaussian mechanism.

# B  Missing Proofs for Section 3

## B.1  Main algorithm results

We first restate our algorithm and main assumptions.

**Assumption** (Restatement of Assumption 4)**.** *There are known* $\Lambda, \lambda > 0$ *such that* $\Lambda \geq \sigma_{\max,*} \geq \sigma_{\min,*} \geq \lambda$.

**Assumption** (Restatement of Assumption 5)**.** *Let* $i \in [n]$ *and* $x \sim \mathcal{D}_i$. *For all* $i \in [n]$ *the feature distribution* $\mathcal{D}_i$ *has covariance matrix* $I_d$ *and is sub-Gaussian in the sense of* $\mathbb{E}\left[e^{\langle x, u \rangle}\right] \leq e^{\frac{\|u\|_2^2}{2}}$ *for all* $u \in \mathbb{R}^d$.

**Assumption** (Restatement of Assumption 6)**.** *Let* $\mathrm{SG}(R)$ *be any centered $R$-sub-Gaussian distribution over* $\mathbb{R}$. *Sample* $\zeta \sim \mathrm{SG}(R)$. *For all* $i \in [n]$ *and any data points* $(x, y) \sim \mathcal{D}_i$ *we generate the label $y$ for the $i$-th user via*

$$y = x^\top U^* v_i^* + \zeta$$

*where $\zeta$ is independent for all* $x \sim \mathcal{D}_i$.

Unless stated otherwise, in this section, we assume that Assumptions 4, 5, and 6 hold. Furthermore, throughout our proofs we assume $n = \Omega(dk)$. This assumption is reasonable in the context of our work since $n = \Omega(dk)$ is required for non-trivial results in our final bounds.

---

**Algorithm 1** Private FedRep for linear regression

---

**Require:** $S_i = \{(x_{i,1}, y_{i,1}), \ldots, (x_{i,m}, y_{i,m})\}$ data for users $i \in [n]$, learning rate $\eta$, iterations $T$, privacy noise parameter $\hat{\sigma}$, clipping parameters $\psi, \psi_{\text{init}}$, batch size $b \leq \lfloor m/2T \rfloor$, initial embedding $U_{\text{init}} \in \mathbb{R}^{d \times k}$
    Let $S_i^0 \leftarrow \{(x_{i,j}, y_{i,j}) : j \in [m/2]\} \quad \forall i \in [n]$
    Let $S_i^1 \leftarrow S_i \setminus S_i^0 \quad \forall i \in [n]$
    **Initialize:** $U_0 \leftarrow U_{\text{init}}$
    **for** $t = 0, \ldots, T-1$ **do**
      Server sends $U_t$ to clients $[n]$
      **for** Clients $i \in [n]$ in parallel **do**
        Sample two disjoint batches $B_{i,t}$ and $B'_{i,t}$, each of size $b$, without replacement from $S_i^0$
        Update $v_i$ as

$$v_{i,t} \leftarrow \arg\min_{v \in \mathbb{R}^k} \widehat{L}(U_t, v; B_{i,t})$$

        Compute the gradient w.r.t $U$

$$\nabla_{i,t} \leftarrow \nabla_U \widehat{L}(U_t, v_{i,t}; B'_{i,t})$$

        Send $\nabla_{i,t}$ to server
      **end for**
      Server aggregates the client gradients as

$$\hat{U}_{t+1} \leftarrow U_t - \eta \left( \frac{1}{n} \sum_{i=1}^n \mathsf{clip}(\nabla_{i,t}, \psi) + \xi_{t+1} \right)$$

$$U_{t+1}, P_{t+1} \leftarrow \mathrm{QR}(\hat{U}_{t+1})$$

      where $\xi_{t+1} \leftarrow \mathcal{N}^{d \times k}(0, \hat{\sigma}^2)$
    **end for**
    Server sends $U^{\text{priv}} \leftarrow U_T$ to all clients
    **for** Clients $i \in [n]$ independently **do**
      $v_i^{\text{priv}} \leftarrow \arg\min_{v \in \mathbb{R}^k} \widehat{L}(U^{\text{priv}}, v; S_i^1) \quad \forall i \in [n]$
    **end for**
    **Return:** $U^{\text{priv}}, V^{\text{priv}} \leftarrow [v_1^{\text{priv}}, \ldots, v_n^{\text{priv}}]^\top$

---

**Lemma** (Restatement of Lemma 7)**.** *With probability at least* $1 - O(T \cdot n^{-10})$, *we have*

$$\|\nabla_{i,t}\|_F \leq \widetilde{O}\left( (R + \Gamma)\Gamma\sqrt{dk} \right)$$

*for all $i \in [n]$ and all $t \in [T]$ simultaneously.*

*Proof.* The privacy guarantee directly follows from the the privacy guarantee of Gaussian mechanism and advanced composition. In the following, we prove a high probability bound for $\|\nabla_{i,t}\|_F$. Let $B_i' = \{(x_{i,j}', y_{i,j}') : j \in [b]\}$ be a data batch of user $i$ sampled at iteration t as in Algorithm 1. Then we have

$$\nabla_{i,t} = \frac{2}{b} \sum_{j=1}^{b} \left( \langle x_{i,j}', U_t v_{i,t+1} - U^* v_i^* \rangle + \zeta_{i,j} \right) x_{i,j}' v_{i,t+1}^\top$$

Denote $u = U_t v_{i,t+1} - U^* v_i^*$. Throughout the proof we condition on the event

$$\mathcal{E} = \left\{ |\zeta_{i,j}| \leq R\sqrt{26 \log(nb)}, |\langle u, x_{i,j}' \rangle| \leq \|u\|_2 \sqrt{26 \log(nb)}, \text{ and } \|v_{i,t+1}\|_2 \leq \frac{5}{4}\Gamma \text{ for all } (i,j) \in [n] \times [b] \right\}$$

which holds by the sub-Gaussianity of $x_{i,j}'$, the independence between $x_{i,j}'$ and $U_t v_{i,t+1} - U^* v_i^*$, and Proposition 42 with probability at least $1 - O(nb \cdot n^{-14}) \geq 1 - O(n^{-12})$. Given event $\mathcal{E}$, we have

$$\langle x_{i,j}', U_t v_{i,t+1} - U^* v_i^* \rangle + \zeta_{i,j} \leq \frac{5}{2}(R + \Gamma)\sqrt{26 \log(nb)}.$$

Meanwhile, the $(p,q)$-element of $x_{i,j}' v_{i,t+1}^\top$ has

$$|(x_{i,j}')_p (v_{i,t+1})_q| \leq \frac{5}{4}\Gamma\sqrt{26 \log(nb)} \tag{1}$$

for all $i, j$ with probability at least $1 - O(nb \cdot n^{-14})$ by the sub-Gaussianity of the components of $x_{i,j}'$ and conditioning on $\mathcal{E}$. Therefore, taking union bound on $p, q, t$ without conditioning, we have, with probability at least $1 - O(Tdk \cdot n^{-12})$

$$\|\nabla_{i,t}\|_F \leq 82(R + \Gamma)\Gamma\sqrt{dk} \log(nb)$$

for all $i, t$ simultaneously. Finally, we bound $1 - O(Tdk \cdot n^{-12}) \geq 1 - O(T \cdot n^{-10})$. $\square$

**Theorem** (Restatement of Theorem 8). *Algorithm 1 is $(\epsilon, \delta)$-user-level DP in the billboard model by setting $\hat{\sigma} = C\frac{\psi\sqrt{T \log(1/\delta)}}{n\epsilon}$ for some absolute constant $C > 0$.*

*Proof.* Since the computation of $V^{\mathrm{priv}}$ is performed by each user independently, it is sufficient to show that the computation of $U^{\mathrm{priv}}$ satisfies centralized user-level DP.

Given the definition of the clipping function, for each $i \in [n]$, we have $\|\mathrm{clip}(\nabla_{i,t}, \psi)\|_F \leq \psi$ which implies that the sensitivity for each update of the $U_t$ is bounded by $\psi$. By taking $\hat{\sigma} = C\frac{\psi\sqrt{T \log(1/\delta)}}{n\epsilon}$, we obtain the privacy cost of each iteration $t$ is $\rho = O\left(\frac{\epsilon^2}{T \ln(1/\delta)}\right)$ under zero-concentrated differential privacy (z-CDP, (Bun & Steinke, 2016)). By the composition property of z-CDP, we obtain the total privacy cost $\rho = \sum_{t=1}^{T} \rho_t = O\left(\frac{\epsilon^2}{\ln(1/\delta)}\right)$. Then, we can convert the z-CDP guarantee to DP guarantee with the privacy parameter being $\rho + \sqrt{\rho \ln(1/\delta)} = O(\epsilon)$ with sufficiently small $\delta$. Therefore, the computation of $U^{\mathrm{priv}}$ is $(\epsilon, \delta)$-DP and the algorithm is $(\epsilon, \delta)$-DP in the billboard model. $\square$

For the rest of our proof we condition on the event of clipping does not affect the original gradient norm in Algorithm 1. By Lemma 7, this event has probability at least $1 - (T \cdot n^{-10})$ for all $i, t$ simultaneously when selecting $\psi = \widetilde{O}\left((R + \Gamma)\Gamma\sqrt{dk}\right)$.

Let $v_{1,t}, \ldots, v_{n,t}$ be the sequence of local user parameters generated at iteration $t$ of Algorithm 1. Define the matrix $\Sigma_{V_t} = \frac{1}{n}\sum_{i=1}^{n} v_{i,t} v_{i,t}^\top$. Take $B_t' = (B_{1,t}', \ldots, B_{n,t}')$ to be the sequence of batches from of Algorithm 1 used to update $U_t$ at iteration $t$ for each user $i$. Assume that we re-index the batches $B_{i,t}' = \{(x_{i,j}^t, y_{i,j}^t) : j \in [b]\}$ for each $i, t$. Let $U_{t+1}, P_{t+1}$ be the matrices from the *QR* decomposition, and

$$\nabla_t = \frac{1}{n}\sum_{i=1}^{n} \nabla_{i,t} = \frac{1}{nb}\sum_{i=1}^{n}\sum_{j=1}^{b} \nabla_U \ell(U_t, v_{i,t}, (x_{i,j}^t, y_{i,j}^t)) \in \mathbb{R}^{d \times k}$$

the aggregated gradient, all computed in Algorithm 1. For ease of notation we also drop the iteration index $t$ on the data points $(x_{i,j}^t, y_{i,j}^t)$. Recall that $\eta$ is the fixed stepsize for Algorithm 1.

**Lemma 28.** *If $P_{t+1}$ is invertible, then*

$$\text{dist}(U_{t+1}, U^*) \leq \|P_{t+1}^{-1}\|_2 \left( \|I_k - \eta \Sigma_{V_t}\|_2 \, \text{dist}(U_t, U^*) + \eta \|\nabla_t - \mathbb{E}_{B'_t}[\nabla_t]\|_2 + \eta \|\xi_t\|_2 \right).$$

*Proof.* Let $\hat{U}_{t+1}$ be the $t$-th iterate of Algorithm 1 before QR decomposition. Observe

$$
\begin{aligned}
\|(U^*)_\perp^\top \hat{U}_{t+1}\|_2 &= \|(U^*)_\perp^\top (U_t - \eta \nabla_t + \eta \xi_t)\|_2 \\
&\leq \|(U^*)_\perp^\top (U_t - \eta \nabla_t + \eta \mathbb{E}_{B'_t}[\nabla_t] - \eta \mathbb{E}_{B'_t}[\nabla_t])\|_2 + \eta \|\xi_t\|_2 \\
&\leq \|(U^*)_\perp^\top (U_t - \eta \mathbb{E}_{B'_t}[\nabla_t])\|_2 + \eta \|\nabla_t - \mathbb{E}_{B'_t}[\nabla_t]\|_2 + \eta \|\xi_t\|_2
\end{aligned}
$$

since $\|(U^*)_\perp^\top\|_2 = 1$. Then, since the data has covariance matrix $I_d$ and label noise $\zeta_{i,j}$ that satisfies $\mathbb{E}[\zeta_{i,j}] = 0$

$$
\begin{aligned}
\mathbb{E}_{B'_t}[\nabla_t] &= \frac{1}{nb} \sum_{i=1}^n \sum_{j=1}^b \mathbb{E}_{B'_t} \left[ ((x'_{i,j})^\top U_t v_{i,t} - (x'_{i,j})^\top U^* v_i^* + \zeta_{i,j}) x'_{i,j} v_{i,t}^\top \right] \\
&= \frac{1}{nb} \sum_{i=1}^n \sum_{j=1}^b \mathbb{E}_{B'_t} \left[ x'_{i,j} (x'_{i,j})^\top (U_t v_{i,t} - U^* v_i^*) v_{i,t}^\top \right] \\
&= \frac{1}{nb} \sum_{i=1}^n \sum_{j=1}^b \mathbb{E}_{B'_t} \left[ (U_t v_{i,t} - U^* v_i^*) v_{i,t}^\top \right] \\
&= \frac{1}{nb} \sum_{i=1}^n \sum_{j=1}^b U_t v_{i,t} v_{i,t}^\top - U^* v_i^* v_{i,t}^\top
\end{aligned}
$$

where the second equality holds because $(x'_{i,j})^\top U_t v_{i,t} - (x'_{i,j})^\top U^* v_i^* + \zeta_{i,j}$ is a scalar and the fourth equality by the fact that $B'_t$ is independent of $U_t, V_t$. Hence

$$
\frac{1}{nb} \sum_{i=1}^n \sum_{j=1}^b (U^*)_\perp^\top (U_t v_{i,t} v_{i,t}^\top - U^* v_i^* v_{i,t}^\top) = \frac{1}{nb} \sum_{i=1}^n \sum_{j=1}^b (U^*)_\perp^\top U_t v_{i,t} v_{i,t}^\top = (U^*)_\perp^\top U_t \frac{1}{n} \sum_{i=1}^n v_{i,t} v_{i,t}^\top
$$

since $(U^*)_\perp^\top U^* = 0$. Then, for $\Sigma_{V_t} = \frac{1}{n} \sum_{i=1}^n v_{i,t} v_{i,t}^\top$

$$\|(U^*)_\perp^\top (U_t - \eta \mathbb{E}_{B'_{t+1}}[\nabla_t])\|_2 = \|(U^*)_\perp^\top U_t (I_k - \eta \Sigma_{V_t})\|_2 \leq \|I_k - \eta \Sigma_{V_t}\|_2 \|(U^*)_\perp^\top U_t\|_2.$$

Hence

$$\|(U^*)_\perp^\top \hat{U}_{t+1}\|_2 \leq \|I_k - \eta \Sigma_{V_t}\|_2 \|(U^*)_\perp^\top U_t\|_2 + \eta \|\nabla_t - \mathbb{E}_{B'_{t+1}}[\nabla_t]\|_2 + \eta \|\xi_t\|_2. \tag{2}$$

Now, assuming that $P_{t+1}$ is invertible

$$
\begin{aligned}
\|(U^*)_\perp^\top U_{t+1}\|_2 &= \|(U^*)_\perp^\top \hat{U}_{t+1} P_{t+1}^{-1}\|_2 \\
&\leq \|P_{t+1}^{-1}\|_2 \|(U^*)_\perp^\top \hat{U}_{t+1}\|_2.
\end{aligned}
$$

So

$$\|(U^*)_\perp^\top U_{t+1}\|_2 \leq \|P_{t+1}^{-1}\|_2 \|(U^*)_\perp^\top \hat{U}_{t+1}\|_2. \tag{3}$$

Combining (2) and (3) finishes the proof. $\square$

Our proof for our main result follows from bounding the terms and factors on the right-hand side of the inequality in Lemma 28; namely

$$\text{dist}(U_{t+1}, U^*) \leq \|P_{t+1}^{-1}\|_2 \left( \|I_k - \eta \Sigma_{V_t}\|_2 \text{dist}(U_t, U^*) + \eta \|\nabla_t - \mathbb{E}_{B'_t}[\nabla_t]\|_2 + \eta \|\xi_t\|_2 \right). \tag{4}$$

The above inequality has four important components. The term $\|\nabla_t - \mathbb{E}_{B_t'}[\nabla_t]\|_2$ is the deviation of the aggregated gradient from its mean and hence can be bounded via concentration inequalities (see Proposition 30) and the term $\|\xi_t\|_2$ is the magnitude of the Gaussian noise added for privacy, which is not difficult to bound as well (see Proposition 29).

Deriving bounds on $\|P_{t+1}^{-1}\|_2$ and $\|I_k - \eta\Sigma_{V_t}\|_2 \mathrm{dist}(U_t, U^*)$ in (4) is less obvious and require some extra work. We can understand $\|I_k - \eta\Sigma_{V_t}\|_2$ as a measurement of how close $\Sigma_{V_t}$ is to $\eta^{-1}I_k$. So, when $\|I_k - \eta\Sigma_{V_t}\|_2$ is small, the vectors $v_{1,t}, \ldots, v_{n,t}$ are evenly distributed in $\mathbb{R}^k$ with no bias in any direction. For example, if $v_{1,t}, \ldots, v_{n,t}$ are independent with $v_{i,t} \sim \mathcal{N}(0, \eta^{-1}I_k)$ for each $i$, the matrix $I_k - \eta\Sigma_{V_t} = I_k - \frac{\eta}{n}\sum_{i=1}^n v_{i,t}v_{i,t}^\top$ will concentrate around the $(d \times k)$-dimensional zero matrix.

The quantity $\|P_{t+1}^{-1}\|_2$ is more subtle. This quantity is a result of relating $\mathrm{dist}(U_{t+1}, U^*)$ and $\mathrm{dist}(U_t, U^*)$ via $\|(U^*)_\perp^\top \hat{U}_{t+1}\|$ as in our proof of Lemma 28. We can interpret $\|P_{t+1}^{-1}\|_2$ as quantifying how far $\hat{U}_{t+1}$ is from having orthonormal columns. As well, we require that $P_{t+1}^{-1}$ exists in general because the principal angle has $\mathrm{dist}(M, M') = 1$ whenever $\mathrm{rank}(M) \neq \mathrm{rank}(M')$.

Recall that $\psi$ is the gradient clipping parameter in Algorithm 1.

**Proposition 29.** *For all $t$, we have*

$$\|\xi_t\|_2 \leq O\left(\frac{\psi\sqrt{dT\log(n)\log(1/\delta)}}{n\epsilon}\right)$$

*with probability at least $1 - O(n^{-10})$.*

*Proof.* Let $\hat{\sigma} = \frac{\psi\sqrt{2T\log(1.25/\delta)}}{n\epsilon}$ as in the statement of Theorem 8. Let $\alpha > 0$. Then, by Corollary 27

$$\hat{\sigma}\left(\sqrt{d} - \left(\sqrt{k} + \alpha\right)\right) \leq \sigma_{\min}(\xi_t) \leq \sigma_{\max}(\xi_t) \leq \hat{\sigma}\left(\sqrt{d} + \left(\sqrt{k} + \alpha\right)\right)$$

with probability at least $1 - 2e^{-\alpha^2}$. Choosing $\alpha = \sqrt{10d\log n}$ gives us

$$\|\xi_t\|_2 \leq \hat{\sigma}\left(\sqrt{d} + \sqrt{k} + \sqrt{10d\log n}\right)$$

$$= \frac{\psi\left(\sqrt{d} + \sqrt{k} + \sqrt{10d\log n}\right)\sqrt{2T\log(1.25/\delta)}}{n\epsilon}$$

$$\leq O\left(\frac{\psi\sqrt{dT\log(n)\log(1/\delta)}}{n\epsilon}\right)$$

with probability at least $1 - 2e^{-10d\log n}$. $\qquad\square$

**Proposition 30.** *For any $t$, we have*

$$\|\nabla_t - \mathbb{E}_{B_t'}[\nabla_t]\|_2 \leq \widetilde{O}\left(\sqrt{\frac{\eta^2(R^2 + \Gamma^2)\Gamma^2 d}{nb}}\right)$$

*with probability at least $1 - O(n^{-10})$.*

*Proof.* Let $\mathcal{N}^d$ and $\mathcal{N}^k$ be Euclidean $\frac{1}{4}$-covers of the $d$ and $k$-dimensional unit spheres, respectively. Then, by Lemma 23 we have

$$\|\nabla_t - \mathbb{E}_{B_t'}[\nabla_t]\|_2 \leq \frac{4}{3}\max_{a \in \mathcal{N}^d, b \in \mathcal{N}^k} a^\top\left(\nabla_t - \mathbb{E}_{B_t'}[\nabla_t]\right)b.$$

Now

$$a^\top\left(\nabla_t - \mathbb{E}_{B_t'}[\nabla_t]\right)b$$

$$= \frac{2}{nb}\sum_{i=1}^n\sum_{j=1}^b \left(\langle x_{i,j}', U_t v_{i,t} - U^* v_i^*\rangle + \zeta_{i,j}\right)\langle a, x_{i,j}'\rangle\langle v_{i,t}, b\rangle$$

$$- \frac{2}{nb}\sum_{i=1}^n\sum_{j=1}^b \mathbb{E}_{B_t'}\left[\left(\langle x_{i,j}', U_t v_{i,t} - U^* v_i^*\rangle + \zeta_{i,j}\right)\langle a, x_{i,j}'\rangle\langle v_{i,t}, b\rangle\right].$$

We condition on $\|v_{i,t}\|_2 \leq \frac{5}{4}\Gamma$ for each $i$, which has probability at least $1 - O(n^{-14})$ by Proposition 42. Then, we have that

$$\frac{2}{nb}\left(\left(\langle x'_{i,j}, U_t v_{i,t} - U^* v_i^*\rangle + \zeta_{i,j}\right)\langle a, x'_{i,j}\rangle\langle v_{i,t}, b\rangle - \mathbb{E}_{B'_t}\left[\left(\langle x'_{i,j}, U_t v_{i,t} - U^* v_i^*\rangle + \zeta_{i,j}\right)\langle a, x'_{i,j}\rangle\langle v_{i,t}, b\rangle\right]\right)$$

are centered, independent sub-exponential random variables when also conditioning on $U_t$. The sub-exponential norm of these random variables is $\left(\frac{5}{4}\right)^2 \cdot \frac{3(R+\Gamma)\Gamma}{nb}$. Then, given $U_t, V_t$ are independent of $B'_t$ and assuming $nb \geq 20(d+k)\log n$, we get from Lemma 22

$$a^\top\left(\nabla_t - \mathbb{E}_{B'_t}[\nabla_t]\right)b \leq \widetilde{O}\left(\sqrt{\frac{\eta^2(R^2+\Gamma^2)\Gamma^2 d}{nb}}\right)$$

with probability at least $1 - O(n^{-10})$. This bound holds unconditionally since $v_{i,t}$ are bounded with probability at least $1 - O(n^{-10})$ for each $i$. Thus

$$\|\nabla_t - \mathbb{E}_{B'_t}[\nabla_t]\|_2 \leq \widetilde{O}\left(\sqrt{\frac{\eta^2(R^2+\Gamma^2)\Gamma^2 d}{nb}}\right) \tag{5}$$

with probability at least $1 - O(n^{-10})$ by the union bound. $\qquad\square$

In Lemma 9 and Theorem 10, we assume there exist $c_0, c_1 > 1$ such that

$$T = \frac{\log n}{\eta\lambda^2} \leq \min\left(\frac{nm\lambda^2}{c_1 \max\{\Delta_{\epsilon,\delta}, 1\}(R+\Gamma)\Gamma d\sqrt{k}m\log^2(nm) + R^2\Gamma^2 d\sigma_{\min,*}^2\log(nm)},\right.$$
$$\left.\frac{m\Gamma^2\sigma_{\max,*}^2}{c_0\max\{R^2, 1\}\cdot\max\{\Gamma^2, 1\}\gamma^4 k\Gamma^2\log^2 n + R^2 k\sigma_{\max,*}^2\log(nm)}\right). \tag{6}$$

This relationship between $T$ and the problem parameters is equivalent to setting

$$T = \frac{\log n}{\eta\lambda^2} \tag{7}$$

along with assuming the following two conditions on $m$ and $n$:

**Assumption 31.** *Let $E_0 = 1 - \text{dist}^2(U_0, U^*) > 0$. We assume there exists some constant $c_0 > 1$ where*

$$m \geq c_0\left(\frac{\max\{R^2, 1\}\cdot\max\{\Gamma^2, 1\}\gamma^4 k\log^2 n}{E_0^2\sigma_{\max,*}^2} + \frac{R^2 k}{\Gamma^2}\log(nm)\right)T. \tag{8}$$

**Assumption 32.** *Let $\Delta_{\epsilon,\delta} = C\frac{\sqrt{\log(1.25/\delta)}}{\epsilon}$ for some constant $C > 0$ and $E_0 = 1 - \text{dist}^2(U_0, U^*)$. For the user count $n$, we assume, for some constant $c_1 > 1$*

$$n \geq c_1\left(\frac{\max\{\Delta_{\epsilon,\delta}, 1\}(R+\Gamma)\Gamma d\sqrt{k}\log^2(nm)}{E_0^2\lambda^2} + \frac{R^2\Gamma^2 d\log(nm)}{m}\right)T. \tag{9}$$

Our Assumptions 31 and 32 are used repeatedly throughout Appendix B. These conditions are easily leveraged as individual assumptions on $m$ and $n$, unlike the upper bound on $T$. Including the equality (7) allows us to contain all required conditions on $T$, $m$, and $n$ to one convenient setting.

**Lemma 33.** *Let $E_0 = 1 - \text{dist}^2(U_0, U^*)$ and $\psi = \widetilde{O}\left((R+\Gamma)\Gamma\sqrt{dk}\right)$. Suppose Assumption 31 and 32 hold. Then, for any iteration $t$, we have that $P_{t+1}$ is invertible and*

$$\|P_{t+1}^{-1}\|_2 \leq \left(1 - \frac{\eta\sigma_{\min,*}^2 E_0}{\sqrt{2\log n}}\right)^{-\frac{1}{2}}$$

*with probability at least $1 - O(n^{-10})$.*

The proof of Lemma 33 is deferred to Appendix B.3.

**Lemma 34.** *Let $\eta \leq \frac{1}{2\sigma_{\max,*}^2}$, $n \geq e^8$, $E_0 = 1 - \text{dist}^2(U_0, U^*)$, and $\psi = \widetilde{O}\left((R + \Gamma)\Gamma\sqrt{dk}\right)$. Suppose Assumptions 31 and 32 hold along with $T = \frac{\log n}{\eta \sigma_{\min,*}^2}$. Then, for all $t$, we have*

$$\|P_{t+1}^{-1}\|_2 \|I_k - \eta \Sigma_{V_t}\|_2 \leq \sqrt{1 - \frac{\eta \sigma_{\min,*}^2 E_0}{4}}$$

*with probability at least $1 - O(n^{-10})$.*

*Proof.* We will show that

$$\|I_k - \eta \Sigma_{V_t}\|_2 \leq 1 - \eta \sigma_{\min}^2\left(\frac{1}{\sqrt{n}}V_t\right) \leq \left(1 - \frac{\eta \sigma_{\min,*}^2 E_0}{4}\right)$$

with high probability for each $t$. Then selecting $n \geq e^8$ and using Lemma 33 implies the product $\|P_{t+1}^{-1}\|_2 \|I_k - \eta \Sigma_{V_t}\|_2$ satisfies the lemma statement.

Note that $\Sigma_{V_t} = \frac{1}{n}V_t^\top V_t = \frac{1}{n}\sum_{i=1}^n v_{i,t} v_{i,t}^\top$ and define $\sigma_{\min,V_t} = \sigma_{\min}\left(\frac{1}{\sqrt{n}}V_t\right)$. Given $V_t = V^*(U^*)^\top U_t - F$ by Lemma 39 and $V^*$ and $(U^*)^\top U$ are full rank, we have

$$\sigma_{\min}\left(\frac{1}{\sqrt{n}}V_t\right) \geq \sigma_{\min}\left(\frac{1}{\sqrt{n}}V^*(U^*)^\top U_t\right) - \sigma_{\max}\left(\frac{1}{\sqrt{n}}F\right)$$

$$\geq \sigma_{\min}\left(\frac{1}{\sqrt{n}}V^*\right)\sigma_{\min}\left((U^*)^\top U_t\right) - \sigma_{\max}\left(\frac{1}{\sqrt{n}}F\right)$$

Now, via the argument of Lemma 39, we have, with probability at least $1 - O(n^{-10})$

$$\sigma_{\max}\left(\frac{1}{\sqrt{n}}F\right) \leq \frac{\nu \tau_k}{1 - \tau_k}\sigma_{\max,*} + \sqrt{\frac{26R^2 \log(nb)}{(1 - \tau_k)^2 b}} \tag{10}$$

for $\tau_k = c_\tau \sqrt{\frac{35k \log n}{b}}$. Recall that $\gamma = \frac{\sigma_{\max,*}}{\sigma_{\min,*}}$ and $\nu = \frac{\Gamma}{\sigma_{\max,*}}$. Then, there exists a constant $\hat{c} > 0$ such that, by Assumption 31 with $c_0 \geq 16\hat{c}^2$, we have

$$\frac{\nu \tau_k}{1 - \tau_k}\sigma_{\max,*} \leq \hat{c}\sqrt{\frac{E_0^2 \sigma_{\max,*}^2}{c_0 \gamma^4 \log n}} \leq \frac{\sigma_{\max,*} E_0}{4\sqrt{\gamma^4 \log n}} \leq \frac{\sigma_{\min,*} E_0}{4\sqrt{\log n}} \tag{11}$$

and

$$\sqrt{\frac{26R^2 \log(nb)}{(1 - \tau_k)^2 b}} \leq \hat{c}\sqrt{\frac{E_0^2 \sigma_{\max,*}^2}{c_0 \gamma^4 k \log n}} \leq \frac{\sigma_{\max*} E_0}{4\sqrt{\gamma^4 k \log n}} \leq \frac{\sigma_{\min,*} E_0}{4\sqrt{\log n}} \tag{12}$$

since $m \geq c_0 \left(\frac{\max\{R^2,1\}\cdot\max\{\Gamma^2,1\}\gamma^4 k \log^2 n}{E_0^2 \sigma_{\max,*}^2}\right) T$ and $\gamma \geq 1$.

Furthermore

$$\sigma_{\min}\left((U^*)^\top U_t\right) \geq \sqrt{1 - \|(U^*)_\perp^\top U_t\|_2^2} = \sqrt{1 - \text{dist}^2(U_t, U^*)}$$

Therefore, we obtain

$$\sigma_{\min,V_t} \geq \sigma_{\min}\left(\frac{1}{\sqrt{n}}V^*\right)\sigma_{\min}\left((U^*)^\top U_t\right) - \sigma_{\max}\left(\frac{1}{\sqrt{n}}F\right)$$

$$\geq \sigma_{\min,*}\left(\sqrt{1 - \text{dist}^2(U_t, U^*)} - \frac{E_0}{2\sqrt{\log n}}\right)$$

with probability at least $1 - O(n^{-10})$. Thus

$$1 - \eta\sigma_{\min,V_t}^2 \leq 1 - \eta\sigma_{\min,*}^2 \left( \sqrt{1 - \text{dist}^2(U_t, U^*)} - \frac{E_0}{2\sqrt{\log n}} \right)^2. \tag{13}$$

**Claim 1:**   We have

$$\|I_k - \eta\Sigma_{V_t}\|_2 \leq 1 - \eta\sigma_{\min,V_t}^2.$$

with probability at least $1 - O(n^{-10})$.

We prove Claim 1 by showing $\frac{1}{2\sigma_{\max,*}^2} \leq \frac{2}{\sigma_{\max,V_t}^2 + \sigma_{\min,V_t}^2}$, which implies

$$\|I_k - \eta\Sigma_{V_t}\|_2 \leq \max\left\{ |1 - \eta\sigma_{\max,V_t}^2|, |1 - \eta\sigma_{\min,V_t}^2| \right\} \leq 1 - \eta\sigma_{\min,V_t}^2.$$

First, we reuse

$$\sigma_{\max}\left( \frac{1}{\sqrt{n}}F \right) \leq \frac{\nu\tau_k}{1 - \tau_k}\sigma_{\max,*} + \sqrt{\frac{26R^2\log(nb)}{(1 - \tau_k)^2 b}}$$

as in (10). By Lemma 39 and the triangle inequality along with the submultiplicativity of the spectral norm

$$\begin{aligned}
\sigma_{\max,V_t} &= \sigma_{\max}\left( \frac{1}{\sqrt{n}}V_t \right) \\
&\leq \sigma_{\max}\left( \frac{1}{\sqrt{n}}V^*(U^*)^\top U_t \right) + \sigma_{\max}\left( \frac{1}{\sqrt{n}}F \right) \\
&\leq \sigma_{\max,*} + \sigma_{\max}\left( \frac{1}{\sqrt{n}}F \right).
\end{aligned}$$

So

$$\sigma_{\min,V_t} \leq \sigma_{\max,*} + \sigma_{\max}\left( \frac{1}{\sqrt{n}}F \right). \tag{14}$$

By (11) and (12), we have

$$\sigma_{\max}\left( \frac{1}{\sqrt{n}}F \right) \leq \frac{\sigma_{\max,*}}{2}$$

and thus via (14)

$$\sigma_{\min,V_t}^2 \leq 2\sigma_{\max,*}^2.$$

Since $\sigma_{\min,V_t} \leq \sigma_{\max,V_t}$ by definition, we have $\sigma_{\min,V_t}^2 + \sigma_{\max,V_t}^2 \leq 4\sigma_{\max,*}^2$. This gives us $\frac{1}{2\sigma_{\max,*}^2} \leq \frac{2}{\sigma_{\max,V_t}^2 + \sigma_{\min,V_t}^2}$ with the required probability. This proves Claim 1.

Next, we will prove the following claim using induction.

**Claim 2:**   Let $\alpha = \frac{2}{5T\sqrt{\log(nm)}} - \frac{1}{25T^2\log(nm)}$. We have $\sqrt{1 - \text{dist}^2(U_t, U^*)} \geq \sqrt{(1 - t\alpha)E_0}$ for each iteration $t \in [T]$.

**Base case:** when $t = 0$, the inequality $\sqrt{1 - \text{dist}^2(U_0, U^*)} \geq \sqrt{E_0}$ is clearly true since $E_0 = 1 - \text{dist}^2(U_0, U^*)$.

**Inductive hypothesis:** we assume $\sqrt{1 - \|(U^*)_\perp^\top U_t\|_2^2} \geq \sqrt{(1 - t\alpha)E_0}$ for some $t \in [T]$.

**Inductive Step:** note $t\alpha < \frac{1}{2}$ for any $t \in [T]$. Our assumption on $\sqrt{1 - \|(U^*)_\perp^\top U_t\|_2^2}$, (13), and assuming $n \geq e^{E_0}$ imply

$$\sigma_{\min,V_t}^2 \geq \sigma_{\min,*}^2 \left( \sqrt{(1 - t\alpha)E_0} - \frac{E_0}{2\sqrt{\log n}} \right)^2. \tag{15}$$

Now, since $\eta \leq \frac{1}{2\sigma_{\max,*}^2} \leq \frac{2}{\sigma_{\max,V_t}^2 + \sigma_{\min,V_t}^2}$, Weyl's inequality and (15) give us

$$\|I_k - \eta\Sigma_{V_t}\|_2 \leq 1 - \eta\sigma_{\min,V_t}^2 \leq 1 - \eta\sigma_{\min,*}^2 \left( \sqrt{(1 - t\alpha)E_0} - \frac{E_0}{2\sqrt{\log n}} \right)^2.$$

By Lemma 33

$$\|P_t^{-1}\|_2 \leq \left(1 - \frac{\eta\sigma_{\min,*}^2 E_0}{\sqrt{2\log n}}\right)^{-\frac{1}{2}}$$

with probability at least $1 - O(n^{-10})$. This event is a superset of the probabilistic bounds from earlier in this lemma. Then

$$\|(U^*)_\perp^\top U_{t+1}\|_2 \leq \left(1 - \frac{\eta\sigma_{\min,*}^2 E_0}{\sqrt{2\log n}}\right)^{-\frac{1}{2}} \left(1 - \eta\sigma_{\min,*}^2 \left(\sqrt{(1-t\alpha)E_0} - \frac{E_0}{2\sqrt{\log n}}\right)^2\right) \|(U^*)_\perp^\top U_t\|_2 \tag{16}$$
$$+ \widetilde{O}\left(\frac{\eta(R+\Gamma)\Gamma d\sqrt{k}}{n\epsilon} + \sqrt{\frac{\eta^2(R^2+\Gamma^2)\Gamma^2 d}{nb}}\right)$$

with probability at least $1 - O(n^{-10})$. Assuming $n \geq e^{E_0/(\sqrt{2}-1/2)^2}$, we have

$$1 - \eta\sigma_{\min,*}^2 \left(\sqrt{(1-t\alpha)E_0} - \frac{E_0}{2\sqrt{\log n}}\right)^2 \leq 1 - \eta\sigma_{\min,*}^2\left(\sqrt{E_0/2} - \frac{E_0}{2\sqrt{\log n}}\right)^2 \leq 1 - \frac{\eta\sigma_{\min,*}^2 E_0}{4}.$$

As well, if $n \geq e^8$, we have

$$\left(1 - \frac{\eta\sigma_{\min,*}^2 E_0}{\sqrt{2\log n}}\right)^{-\frac{1}{2}} \left(1 - \eta\sigma_{\min,*}^2\left(\sqrt{(1-t\alpha)E_0} - \frac{E_0}{2\sqrt{\log n}}\right)^2\right)$$
$$\leq \left(1 - \frac{\eta\sigma_{\min,*}^2 E_0}{\sqrt{2\log n}}\right)^{-\frac{1}{2}} \left(1 - \eta\sigma_{\min,*}^2\left(\sqrt{E_0/2} - \frac{E_0}{2\sqrt{\log n}}\right)^2\right)$$
$$\leq \left(1 - \frac{\eta\sigma_{\min,*}^2 E_0}{\sqrt{2\log n}}\right)^{-\frac{1}{2}} \left(1 - \frac{\eta\sigma_{\min,*}^2 E_0}{4}\right)$$
$$\leq \left(1 - \frac{\eta\sigma_{\min,*}^2 E_0}{4}\right)^{-\frac{1}{2}} \left(1 - \frac{\eta\sigma_{\min,*}^2 E_0}{4}\right)$$
$$= \sqrt{1 - \frac{\eta\sigma_{\min,*}^2 E_0}{4}}.$$

We now use the lower bound assumptions of the user count $n$ and data sample count $m$. Recall the exact statements of Assumption 31 and 32. That is, there exist $c_0, c_1 > 1$ such that

$$m \geq c_0 \left(\frac{\max\{R^2, 1\} \cdot \max\{\Gamma^2, 1\}\gamma^4 k \log^2 n}{E_0^2 \sigma_{\max,*}^2} + \frac{R^2 k}{\Gamma^2}\log(nm)\right) T$$

and

$$n \geq c_1 \left(\frac{\max\{\Delta_{\epsilon,\delta}, 1\}(R+\Gamma)\Gamma d\sqrt{k}\log^2(nm)}{E_0^2 \lambda^2} + \frac{R^2\Gamma^2 d\log(nm)}{m}\right) T.$$

The problem parameters that lower bound $m, n$ now come into use during this proof. Furthermore, recall that by Assumption 4 we know some $\lambda > 0$ such that $\lambda \leq \sigma_{\min,*}$. Then, by Assumptions 31 and 32 along with $n \geq e^8$ and $T = \frac{\log n}{\eta\lambda^2}$, there

exists a constant $\hat{c} > 0$ such that

$$
\begin{aligned}
\|(U^*)_\perp^\top U_{t+1}\|_2 &\leq \left(1 - \frac{\eta\sigma_{\min,*}^2 E_0}{4}\right)^{\frac{1}{2}} \|(U^*)_\perp^\top U_t\|_2 \\
&\quad + O\left(\left(1 - \frac{\eta\sigma_{\min,*}^2 E_0}{\sqrt{2\log n}}\right)^{-\frac{1}{2}} \frac{\eta(R+\Gamma)\Gamma d\sqrt{Tk\log(1/\delta)}\log(nb)}{n\epsilon}\right. \\
&\quad + \left(1 - \frac{\eta\sigma_{\min,*}^2 E_0}{\sqrt{2\log n}}\right)^{-\frac{1}{2}} \left.\sqrt{\frac{\eta^2(R^2+\Gamma^2)\Gamma^2 d\log n}{nb}}\right) \\
&\leq \left(1 - \frac{\eta\sigma_{\min,*}^2 E_0}{4}\right)^{\frac{1}{2}} \|(U^*)_\perp^\top U_t\|_2 + O\left(\frac{\eta(R+\Gamma)\Gamma d\sqrt{Tk\log(1/\delta)}\log(nb)}{n\epsilon}\right. \\
&\quad + \left.\sqrt{\frac{\eta^2(R^2+\Gamma^2)\Gamma^2 d\log(nm)}{nb}}\right) \\
&\leq \left(1 - \frac{\eta\sigma_{\min,*}^2 E_0}{4}\right)^{\frac{1}{2}} \|(U^*)_\perp^\top U_t\|_2 + O\left(\frac{\eta\lambda^2 E_0^2 \sqrt{T}}{c_1 T\log(nm)}\right. \\
&\quad + \left.\sqrt{\frac{E_0^4 \eta^2(R+\Gamma)\Gamma\sigma_{\max,*}^2\lambda^2 T}{c_0 c_1 \max\{R^2,1\}\cdot\max\{\Gamma^2,1\}\gamma^4 k^{\frac{3}{2}} T^2\log(nm)\log^2 n}}\right) \\
&\leq \left(1 - \frac{\eta\sigma_{\min,*}^2 E_0}{4}\right)^{\frac{1}{2}} \|(U^*)_\perp^\top U_t\|_2 + O\left(\frac{\sqrt{\eta}\sigma_{\min,*} E_0^2}{c_1 T\sqrt{\log(nm)}}\right. \\
&\quad + \left.\sqrt{\frac{E_0^4 \eta(R+\Gamma)\Gamma\sigma_{\max,*}^2\log n}{c_0 c_1 \max\{R^2,1\}\cdot\max\{\Gamma^2,1\}\gamma^4 k^{\frac{3}{2}} T^2\log(nm)\log^2 n}}\right) \\
&= \left(1 - \frac{\eta\sigma_{\min,*}^2 E_0}{4}\right)^{\frac{1}{2}} \|(U^*)_\perp^\top U_t\|_2 + \frac{\hat{c} E_0^2}{c_1\gamma T\sqrt{\log(nm)}} + \hat{c}\sqrt{\frac{E_0^4}{c_0 c_1\gamma^4 k^{\frac{3}{2}} T^2\log(nm)\log n}}
\end{aligned}
$$ (17)

since $\eta \leq \frac{1}{2\sigma_{\max,*}^2}$ and $\lambda \leq \sigma_{\min,*}$. Recall that $\gamma, k \geq 1$. Using the choice of $c_0, c_1 \geq 10\hat{c}$ in (17), we have

$$
\begin{aligned}
\|(U^*)_\perp^\top U_{t+1}\|_2 &\leq \left(1 - \frac{\eta\sigma_{\min,*}^2 E_0}{4}\right)^{\frac{1}{2}} \|(U^*)_\perp^\top U_t\|_2 + \frac{E_0^2}{10\gamma T\sqrt{\log(nm)}} + \sqrt{\frac{E_0^4}{100\gamma^4 k^{\frac{3}{2}} T^2\log(nm)\log n}} \\
&\leq \|(U^*)_\perp^\top U_t\|_2 + \frac{E_0^2}{5T\sqrt{\log(nm)}}.
\end{aligned}
$$

Thus, given $E_0 \leq 1$

$$
\begin{aligned}
1 - \|(U^*)_\perp^\top U_{t+1}\|_2^2 &\geq 1 - \left(\|(U^*)_\perp^\top U_t\|_2 + \frac{E_0^2}{5T\sqrt{\log(nm)}}\right)^2 \\
&= 1 - \|(U^*)_\perp^\top U_t\|_2^2 - \frac{2E_0^2\|(U^*)_\perp^\top U_t\|_2^2}{5T\sqrt{\log(nm)}} - \frac{E_0^4}{25T^2\log(nm)} \\
&\geq 1 - \|(U^*)_\perp^\top U_t\|_2^2 - E_0\left(\frac{2}{5T\sqrt{\log(nm)}} + \frac{1}{25T^2\log(nm)}\right).
\end{aligned}
$$

Then, by our assumptions that $\sqrt{1 - \|(U^*)_\perp^\top U_t\|_2^2} \geq \sqrt{(1-t\alpha)E_0}$, we have

$$
1 - \|(U^*)_\perp^\top U_{t+1}\|_2^2 \geq 1 - \|(U^*)_\perp^\top U_{t+1}\|_2^2 - \alpha E_0 \geq (1-t\alpha)E_0 - \alpha E_0 = (1-(t+1)\alpha)E_0.
$$

Thus, by inductive hypothesis

$$\sigma_{\min}\left((U^*)^\top U_t\right) \geq \sqrt{1 - \text{dist}^2(U_t, U^*)} \geq \sqrt{(1 - t\alpha)E_0}$$

for any $t \geq 0$. This proves Claim 2.

Observe that $t\alpha \leq T\alpha = T\frac{10T\sqrt{\log(nm)}+1}{25T^2\log(nm)} < \frac{1}{2}$ for all $t$. Using this, Claim 2 implies

$$\sigma_{\min}\left((U^*)^\top U_t\right) \geq \sqrt{1 - \text{dist}^2(U_t, U^*)} \geq \sqrt{(1 - t\alpha)E_0} \geq \sqrt{E_0/2} \tag{18}$$

for each $t \geq 0$. Via (18) and $n \geq e^8$ along with $\eta \leq \frac{1}{2\sigma_{\max,*}^2}$

$$\|P_{t+1}^{-1}\|_2\|I_k - \eta\Sigma_{V_t}\|_2 \leq \sqrt{1 - \frac{\eta\sigma_{\min,*}^2 E_0}{4}}$$

with probability at least $1 - O(n^{-10})$ for any $t \in [T-1]$. This proves the lemma. □

**Lemma** (Restatement of Lemma 9). *Let $\eta \leq \frac{1}{2\sigma_{\max,*}^2}$. Suppose $1 - \text{dist}^2(U_0, U^*) \geq c$ for some constant $c > 0$. Set the clipping parameter $\psi = \widetilde{O}\left((R + \Gamma)\Gamma\sqrt{dk}\right)$. Assume $T = \frac{\log n}{\eta\lambda^2} = \widetilde{O}\left(\min\left(\frac{nm\lambda^2}{\max\{\Delta_{\epsilon,\delta},1\}(R+\Gamma)\Gamma d\sqrt{km}+R^2\Gamma^2 d\sigma_{\min,*}^2}\right)\right.$,*

$$\left.\frac{m\Gamma^2\sigma_{\max,*}^2}{\max\{R^2,1\}\cdot\max\{\Gamma^2,1\}\gamma^4 k\Gamma^2+R^2 k\sigma_{\max,*}^2}\right)\right)$$ *and Assumptions 5 and 6 hold for all user data. Set $\hat{\sigma}$ as in Theorem 8 and batch size $b = \lfloor m/2T \rfloor$. Then, $U^{\text{priv}}$, the first output of Algorithm 1, satisfies*

$$\text{dist}(U^{\text{priv}}, U^*) \leq \left(1 - \frac{c\eta\sigma_{\min,*}^2}{4}\right)^{\frac{T}{2}}\text{dist}(U_0, U^*) + \widetilde{O}\left(\frac{(R+\Gamma)\Gamma d\sqrt{kT}}{n\epsilon\sigma_{\min,*}^2} + \sqrt{\frac{(R^2+\Gamma^2)\Gamma^2 dT}{nm\sigma_{\min,*}^4}}\right)$$

*with probability at least $1 - O(T \cdot n^{-10})$.*

*Proof.* By Lemma 28

$$\|(U^*)_\perp^\top U_{t+1}\|_2 \leq \|P_{t+1}^{-1}\|_2\left(\|I_k - \eta\Sigma_{V_t}\|_2\|(U^*)_\perp^\top U_t\|_2 + \eta\|\nabla_t - \mathbb{E}_{B_t'}[\nabla_t]\|_2 + \eta\|\xi_t\|_2\right). \tag{19}$$

We must bound all three terms on the right-hand side of (19). Combining (19) and Proposition 30

$$\|(U^*)_\perp^\top U_{t+1}\|_2 \leq \|P_{t+1}^{-1}\|_2\left(\|I_k - \eta\Sigma_{V_t}\|_2\|(U^*)_\perp^\top U_t\|_2 + \widetilde{O}\left(\sqrt{\frac{\eta^2(R^2+\Gamma^2)\Gamma^2 d}{nb}}\right) + \eta\|\xi_t\|_2\right) \tag{20}$$

with probability at least $1 - O(n^{-10})$. Combining our choice of $\psi$ with Proposition 29, via the union bound

$$\|(U^*)_\perp^\top U_{t+1}\|_2 \leq \|P_{t+1}^{-1}\|_2\left(\|I_k - \eta\Sigma_{V_t}\|_2\|(U^*)_\perp^\top U_t\|_2 + \widetilde{O}\left(\frac{\eta(R+\Gamma)\Gamma d\sqrt{kT}}{n\epsilon} + \sqrt{\frac{\eta^2(R^2+\Gamma^2)\Gamma^2 d}{nb}}\right)\right) \tag{21}$$

for all $t \in [T-1]$ simultaneously with probability at least $1 - O(T \cdot n^{-10})$.

To bound the right-hand side of the above inequality, we need to bound $\|P_{t+1}^{-1}\|_2$. By Lemma 33, with probability at least $1 - O(T \cdot n^{-10})$, we have $\|P_{t+1}^{-1}\|_2 \leq \left(1 - \frac{\eta\sigma_{\min,*}^2 E_0}{\sqrt{2\log n}}\right)^{-\frac{1}{2}} = O(1)$ for all $t$ simultaneously, where the last equality follows from the fact that $\eta \leq \frac{1}{2\sigma_{\max,*}^2}$, $E_0 \leq 1$, and $\log n \geq 1$ (assuming, without loss of generality, that $n \geq 2$). Further, by Lemma 33

$$\|(U^*)_\perp^\top U_{t+1}\|_2 \leq \left(1 - \frac{\eta\sigma_{\min,*}^2 E_0}{4}\right)^{\frac{1}{2}}\|(U^*)_\perp^\top U_t\|_2 + \widetilde{O}\left(\frac{\eta(R+\Gamma)\Gamma d\sqrt{kT}}{n\epsilon} + \sqrt{\frac{\eta^2(R^2+\Gamma^2)\Gamma^2 d}{nb}}\right) \tag{22}$$

for all $t$ simultaneously with probability at least $1 - O(T \cdot n^{-10})$.

Now, what remains is to obtain the tightest possible bound on the recursion from Lemma 22 via the summation of a geometric sum. Since $\text{dist}(U^*, U_t) = \|(U^*)_\perp^\top U_t\|_2 \leq 1$, we have, for all $t$ simultaneously

$$
\begin{aligned}
&\text{dist}(U^*, U_{t+1}) \\
&\leq \left(1 - \frac{\eta \sigma_{\min,*}^2 E_0}{4}\right)^{\frac{1}{2}} \text{dist}(U^*, U_t) + \widetilde{O}\left(\frac{\eta(R+\Gamma)\Gamma d\sqrt{kT}}{n\epsilon} + \sqrt{\frac{\eta^2(R^2+\Gamma^2)\Gamma^2 d}{nb}}\right) \\
&\leq \left(1 - \frac{\eta \sigma_{\min,*}^2 E_0}{4}\right)^{\frac{T}{2}} \text{dist}(U^*, U_t) \\
&\quad + \widetilde{O}\left(\sum_{t=0}^{T-1}\left(1 - \frac{\eta \sigma_{\min,*}^2 E_0}{4}\right)^{\frac{t}{2}}\left(\frac{\eta(R+\Gamma)\Gamma d\sqrt{kT}}{n\epsilon} + \sqrt{\frac{\eta^2(R^2+\Gamma^2)\Gamma^2 d}{nb}}\right)\right) \\
&\leq \left(1 - \frac{\eta \sigma_{\min,*}^2 E_0}{4}\right)^{\frac{T}{2}} \text{dist}(U^*, U_t) \\
&\quad + \widetilde{O}\left(\frac{1}{1 - \sqrt{1 - \eta \sigma_{\min,*}^2 E_0/4}}\left(\frac{\eta(R+\Gamma)\Gamma d\sqrt{kT}}{n\epsilon} + \sqrt{\frac{\eta^2(R^2+\Gamma^2)\Gamma^2 d}{nb}}\right)\right) \\
&\leq \left(1 - \frac{c\eta \sigma_{\min,*}^2}{4}\right)^{\frac{T}{2}} \text{dist}(U^*, U_0) + \widetilde{O}\left(\frac{(R+\Gamma)\Gamma d\sqrt{kT}}{n\epsilon \sigma_{\min,*}^2} + \sqrt{\frac{(R^2+\Gamma^2)\Gamma^2 d}{nb\sigma_{\min,*}^4}}\right)
\end{aligned}
$$

with probability at least $1 - O(T \cdot n^{-10})$, given that $\frac{1}{1-\sqrt{1-\frac{1}{x}}} \leq 2x$ for any $x > 1$ and since there exists $c > 0$ such that $E_0 \geq c$. Selecting batch size $b = \lfloor m/2T \rfloor$ completes the proof of the lemma. $\qquad\square$

**Lemma 35** (Adaptation of Theorem 4.2 (Jain et al., 2021)). *Let $X$ be a matrix with rows sampled from $\mathcal{D}_i$ and $\vec{y}$ the vector of its labels generated according to Assumption 6. Suppose that $(x, y)$ with $x \sim \mathcal{D}_i$ is a data point independent of $X$ with label vector $y$. Take $\bar{U}$ to be any matrix with orthonormal columns and $\bar{v} \in \arg\min_v \|v^\top \bar{U}^\top X - \vec{y}\|_2^2$. Then*

$$
\mathbb{E}\left[\ell(\bar{v}, \bar{U}, (x, y)) - \ell(v^*, U^*, (x, y))\right] \leq \Gamma^2 \|(\bar{U}\bar{U}^\top - I)U^*\|_2^2 + \frac{R^2 k}{m}.
$$

*Proof.* Assume that $X \sim \mathcal{D}_i^m$ is a data matrix with label vector $\vec{y}$ generated as in Assumption 6 using noise vector $\vec{\zeta} \sim \text{SG}(R^2)^m$. Let $x \sim \mathcal{D}_i$ be a data point that is independent of $X$. By definition of our loss function $\ell$ and data generation method

$$
\begin{aligned}
\mathbb{E}_{x \sim \mathcal{D}_i, \zeta}\left[\ell(v, U, (x, y))\right] &= \mathbb{E}_{x \sim \mathcal{D}_i}\left[\left(x^\top U v - x^\top U^* v^* + \zeta\right)^2\right] \\
&= (Uv - U^*v^*)^\top \mathbb{E}_{x \sim \mathcal{D}_i}\left[xx^\top\right](Uv - U^*v^*) + \mathbb{E}[\zeta^2] \\
&= \|Uv - U^*v^*\|_2^2 + \mathbb{E}[\zeta^2]
\end{aligned}
$$

for any fixed $U \in \mathbb{R}^{d \times k}, v \in \mathbb{R}^k$. Since $\bar{v} \in \arg\min_v \|v^\top \bar{U}^\top X - \vec{y}\|_2^2$ we have $\bar{v} = \left(\bar{U}^\top XX^\top \bar{U}\right)^{-1} \bar{U}^\top X\vec{y}$, where this

inverse exists by our assumption that $k \ll m$. Then

$$
\begin{aligned}
\mathbb{E}\left[\left\|\bar{U}\bar{v} - U^*v^*\right\|_2^2\right] &= \mathbb{E}\left[\left\|\bar{U}\left(\bar{U}^\top XX^\top \bar{U}\right)^{-1}\bar{U}^\top X\vec{y} - U^*v^*\right\|_2^2\right] \\
&= \mathbb{E}\left[\left\|\bar{U}\left(\bar{U}^\top XX^\top \bar{U}\right)^{-1}\bar{U}^\top XX^\top U^*v^* - U^*v^* + \bar{U}\left(\bar{U}^\top XX^\top \bar{U}\right)^{-1}\bar{U}^\top X\zeta\right\|_2^2\right] \\
&\leq \mathbb{E}\left[\left\|\bar{U}\bar{U}^\top U^*v^* - U^*v^*\right\|_2^2\right] + \frac{R^2 k}{m} \\
&= \left\|\bar{U}\bar{U}^\top U^*v^* - U^*v^*\right\|_2^2 + \frac{R^2 k}{m} \\
&\leq \Gamma^2\left\|(\bar{U}\bar{U}^\top - I)U^*\right\|_2^2 + \frac{R^2 k}{m}
\end{aligned}
$$

where the first inequality follows from the fact that $\left(\bar{U}^\top XX^\top \bar{U}\right)^{-1}\bar{U}^\top XX^\top \bar{U} = I$ implies $\bar{U}^\top = \left(\bar{U}^\top XX^\top \bar{U}\right)^{-1}\bar{U}^\top XX^\top$ by orthonormality of the columns of $\bar{U}$ and the expected mean square estimation error of sub-Gaussian noise. Combining our two inequalities with $\mathbb{E}\left[\ell(v^*, U^*, (x, y))\right] = \mathbb{E}[\zeta^2]$ completes the proof. $\qquad\square$

**Theorem** (Restatement of Theorem 10). *Suppose all conditions of Lemma 9 hold with $\eta = \frac{1}{2\Lambda^2}$, $T = \Theta\left(\frac{\Lambda^2 \log(n^3)}{\lambda^2}\right)$, and that Assumption 4 holds as well. Then, $U^{\mathrm{priv}}$ and $V^{\mathrm{priv}}$, the outputs of Algorithm 1, satisfy*

$$
\begin{aligned}
&L(U^{\mathrm{priv}}, V^{\mathrm{priv}}; \mathcal{D}) - L(U^*, V^*; \mathcal{D}) \\
&\leq \widetilde{O}\left(\frac{(R^2 + \Gamma^2)\Gamma^4\Lambda^2 d^2 k}{n^2\epsilon^2\sigma_{\min,*}^4\lambda^2} + \frac{(R^2 + \Gamma^2)\Gamma^4\Lambda^2 d}{nm\sigma_{\min,*}^4\lambda^2}\right) \\
&\quad + \frac{R^2 k}{m}
\end{aligned}
$$

*with probability at least $1 - O(T \cdot n^{-10})$. Furthermore, Algorithm 1 is $(\epsilon, \delta)$-user-level DP.*

*Proof.* Via Lemma 9 and $\eta \leq \frac{1}{2\sigma_{\max,*}^2}$

$$
\mathrm{dist}(U^{\mathrm{priv}}, U^*) \leq \left(1 - \frac{c\eta\sigma_{\min,*}^2}{4}\right)^{\frac{T}{2}} \mathrm{dist}(U_0, U^*) + \widetilde{O}\left(\frac{(R + \Gamma)\Gamma d\sqrt{kT}}{n\epsilon\sigma_{\min,*}^2} + \sqrt{\frac{(R^2 + \Gamma^2)\Gamma^2 dT}{nm\sigma_{\min,*}^4}}\right)
$$

with probability at least $1 - O(T \cdot n^{-10})$. Note that $\mathrm{dist}(U_T, U^*) = \left\|(U_T U_T^\top - I)U^*\right\|_2$. Applying Lemma 35 and plugging our choice of $T$ in this bound finishes the proof. $\qquad\square$

**Theorem 36.** *(Theorem 4 (Tripuraneni et al., 2021)) Given a new user with a dataset $S_{n+1}$ of size $m_2$ whose elements sampled from distribution $\mathcal{D}_{n+1}$ where assumptions 6 and 5 hold with $\|v_{n+1}^*\| \leq \Gamma$. Then if $\mathrm{dist}(U^{\mathrm{priv}}, U^*) \leq \delta$ and $m_2 \geq k \log m_2$, let $v_{n+1}^{\mathrm{priv}} = \arg\min_{v \in \mathbb{R}^k} \widehat{L}(U^{\mathrm{priv}}, v; S_{n+1})$, then we have*

$$
\|U^*v_{n+1}^* - U^{\mathrm{priv}}v_{n+1}^{\mathrm{priv}}\|^2 \leq \tilde{O}\left(\Gamma^2\delta^2 + \frac{R^2 k}{m_2}\right)
$$

*with probability at least $1 - O(m_2^{-100})$*

**Corollary** (Restatement of Corollary 11). *Suppose all conditions of Theorem 10 hold. Consider a new client with a dataset $S_{n+1} \sim \mathcal{D}_{n+1}^{m'}$, where Assumptions 5 and 6 hold with $\|v_{n+1}^*\|_2 \leq \Gamma$. Let $U^{\mathrm{priv}}$ be the output of Algorithm 1 and $v_{n+1}^{\mathrm{priv}} = \arg\min_{v \in \mathbb{R}^k} \hat{L}(U^{\mathrm{priv}}, v; S_{n+1})$, if $m' \gtrsim k \log m'$. We have*

$$
\begin{aligned}
&L(U^{\mathrm{priv}}, v_{n+1}^{\mathrm{priv}}; \mathcal{D}_{n+1}) - L(U^*, v_{n+1}^*; \mathcal{D}_{n+1}) \\
&\leq \widetilde{O}\left(\frac{(R^2 + \Gamma^2)\Gamma^4\Lambda^2 d^2 k}{n^2\epsilon^2\sigma_{\min,*}^4\lambda^2} + \frac{(R^2 + \Gamma^2)\Gamma^4\Lambda^2 d}{nm\sigma_{\min,*}^4\lambda^2} + \frac{R^2 k}{m'}\right)
\end{aligned}
$$

*with probability at least $1 - O(T \cdot n^{-10} + m'^{-100})$.*

*Proof.* From the proof in Lemma 35, we have

$$L(U^{\mathrm{priv}}, v_{n+1}^{\mathrm{priv}}; \mathcal{D}_{n+1}) - L(U^*, v_{n+1}^*; \mathcal{D}_{n+1}) \leq \|U^* v_{n+1}^* - U^{\mathrm{priv}} v_{n+1}^{\mathrm{priv}}\|^2 + \mathbb{E}[\eta^2] - \mathbb{E}[\eta^2]$$
$$= \|U^* v_{n+1}^* - U^{\mathrm{priv}} v_{n+1}^{\mathrm{priv}}\|^2$$

Then by Lemma 9 and our choice of $T$, we obtain

$$\mathrm{dist}(U^{\mathrm{priv}}, U^*) \leq \widetilde{O}\left( \frac{(R+\Gamma)\Gamma d\Lambda\sqrt{k}}{n\epsilon\sigma_{\min,*}^2 \lambda} + \sqrt{\frac{(R^2+\Gamma^2)\Gamma^2 d\Lambda^2}{nm\sigma_{\min,*}^4 \lambda^2}} \right)$$

Then we plug the bound of $\mathrm{dist}(U^{\mathrm{priv}}, U^*)$ into Theorem 36, and obtain

$$L(U^{\mathrm{priv}}, v_{n+1}^{\mathrm{priv}}; \mathcal{D}_{n+1}) - L(U^*, v_{n+1}^*; \mathcal{D}_{n+1}) \leq \|U^* v_{n+1}^* - U^{\mathrm{priv}} v_{n+1}^{\mathrm{priv}}\|^2$$
$$\leq \widetilde{O}\left( \frac{(R^2+\Gamma^2)\Gamma^4\Lambda^2 d^2 k}{n^2\epsilon^2\sigma_{\min,*}^4 \lambda^2} + \frac{(R^2+\Gamma^2)\Gamma^4\Lambda^2 d}{nm\sigma_{\min,*}^4 \lambda^2} + \frac{R^2 k}{m_2} \right).$$

$\square$

## B.2 Private initialization results

**Theorem 37** (Theorem 2.1 (Duchi et al., 2022)). *Let $M, \hat{M} \in \mathbb{R}^{d\times d}$ be symmetric matrices. Suppose $p \in [k]$ for $k$ a positive integer with $k < d$. Denote by $\lambda_p$ the $p$-th largest eigenvalue of $M$. Let $A, \hat{A}$ be matrices with columns the top-$k$ eigenvectors of $M, \hat{M}$, respectively. Then, $\lambda_k - \lambda_{k+1} > 0$ implies*

$$\mathrm{dist}(A, \hat{A}) \leq \frac{2\|M - \hat{M}\|_2}{\lambda_k - \lambda_{k+1}}.$$

*Proof.* The theorem holds trivially when $\frac{2\|M-\hat{M}\|_2}{\lambda_k-\lambda_{k+1}} > 1$. So, we focus on the case $\frac{2\|M-\hat{M}\|_2}{\lambda_k-\lambda_{k+1}} \leq 1$.

Let $\hat{\lambda}_p$ be the $p$-th largest eigenvalue of $\hat{M}$. By Weyl's inequality and $\frac{2\|M-\hat{M}\|_2}{\lambda_k-\lambda_{k+1}} \leq 1$, we have

$$\hat{\lambda}_{k+1} - \lambda_{k+1} \leq \|M - \hat{M}\|_2 \leq \frac{\lambda_k - \lambda_{k+1}}{2}.$$

This and the assumption $\lambda_k - \lambda_{k+1} > 0$ imply

$$\lambda_{k+1} - \hat{\lambda}_{k+1} \geq \frac{\lambda_k - \lambda_{k+1}}{2} > 0. \tag{23}$$

By the Davis-Kahan theorem (Stewart, 1990)

$$\|A_\perp^\top \hat{A}\|_2 \leq \frac{2\|M - \hat{M}\|_2}{\lambda_k - \hat{\lambda}_{k+1}}. \tag{24}$$

Combining (23) and (24) along with $\|A_\perp^\top \hat{A}\|_2 = \mathrm{dist}(A, \hat{A})$ finishes the proof. $\square$

We use a slight adaptation of Theorem L.1 from (Duchi et al., 2022). The statement of the result below is different from the original result by a single step. This step is where the authors use the Davis-Kahan theorem to obtain a bound on the principal angle. We instead state their Theorem L.1 before applying Davis-Kahan for use in our private initialization guarantee.

**Theorem 38** (Adaptation of Theorem L.1 (Duchi et al., 2022)). *Let $S = (S_1, \ldots, S_n)$ be a sequence of datasets where $S_i = \{(x_{i,1}, y_{i,1}), \ldots, (x_{i,m/2}, y_{i,m/2})\}$ are sampled according to Assumptions 5 and 6 for each $i \in [n]$. Define $Z_i = \frac{2}{m(m-1)} \sum_{j_1,j_2 \in [m/2]: j_1 \neq j_2} y_{i,j_1} y_{i,j_2} x_{i,j_1} x_{i,j_2}^\top$, $Z = \frac{1}{n}\sum_{i=1}^n Z_i$, and $\bar{Z} = \frac{1}{n}\sum_{i=1}^n \mathbb{E}[Z_i]$. Then, we have*

$$\|Z - \bar{Z}\|_2 \leq O\left( \log^3(nd)\sqrt{\frac{(R^2+\Gamma^2)\Gamma^2 d}{mn}} \right)$$

*with probability at least $1 - O(n^{-10})$.*

---

**Algorithm 2 Private Initialization** for Private FedRep

---

**Require:** $S_i = \{(x_{i,1}, y_{i,1}), \ldots, (x_{i,m/2}, y_{i,m/2})\}$ data for users $i \in [n]$, privacy parameters $\epsilon, \delta$, clipping bound $\psi_{\text{init}}$, rank $k$

1: Let $\hat{\sigma}_{\text{init}} \leftarrow \frac{\psi_{\text{init}}\sqrt{\log\left(\frac{1.25}{\delta}\right)}}{n\epsilon}$

2: Let $\xi_{\text{init}} \leftarrow \mathcal{N}^{d\times d}(\vec{0}, \hat{\sigma}_{\text{init}}^2)$

3: **for** Clients $i \in [n]$ in parallel **do**

4:    Send $Z_i \leftarrow \frac{2}{m(m-1)} \sum_{j_1 \neq j_2} y_{i,j_1} y_{i,j_2} x_{i,j_1} x_{i,j_2}^\top$ to server

5: **end for**

6: Server aggregates $Z_i$ and add noise for privatization

$$\hat{Z} = \frac{1}{n}\sum_{i=1}^{n} \text{clip}(Z_i, \psi_{\text{init}}) + \xi_{\text{init}}$$

7: Server computes

$$U_{\text{init}} D U_{\text{init}}^\top \leftarrow \text{rank-}k\text{-SVD}(\hat{Z})$$

8: **Return:** $U_{\text{init}}$

---

**Lemma** (Restatement of Lemma 12). *Suppose that Assumptions 5 and 6 hold. Let $U_{\text{init}}$ be the output of Algorithm 2. Then, by setting $\psi_{\text{init}} = \widetilde{O}((R^2 + \Gamma^2)d)$, we have*

$$\text{dist}(U_{\text{init}}, U^*) \leq \widetilde{O}\left(\frac{(R^2 + \Gamma^2)d^{3/2}}{n\epsilon\sigma_{\min,*}^2} + \sqrt{\frac{(R^2 + \Gamma^2)\Gamma^2 d}{mn\sigma_{\min,*}^4}}\right)$$

*with probability at least $1 - O(n^{-10})$. Furthermore, Algorithm 2 is $(\epsilon, \delta)$ user-level DP.*

*Proof.* The privacy guarantee follows directly from the guarantee of Gaussian mechanism. For our utility guarantee, we condition on the event

$$\mathcal{E} = \left\{|y_{i,j}| \leq O((\Gamma + R)\log(mn)), \left|x_{i,j}^p\right| \leq O\left(\sqrt{\log dmn}\right) \text{ for all } (i, j, p) \in [n] \times [m] \times [d] \text{ simultaneously}\right\}$$

where $x_{i,j}^p$ represents the $p$-th entry of $x_{i,j}$. The condition $\mathcal{E}$ holds with probability at least $1 - O(n^{-10})$.

Conditioning on event $\mathcal{E}$, we obtain

$$\|Z_i\|_F \leq \widetilde{O}\left((R^2 + \Gamma^2)d\right)$$

and the clipping will not change the gradient norm. Let $Z = \frac{1}{n}\sum_{i=1}^{n} Z_i$, $\bar{Z} = \frac{1}{n}\sum_{i=1}^{n} \mathbb{E}[Z_i]$. Via Corollary 27 and the fact that the clipping does not affect the bound, we have

$$\|\hat{Z} - Z\|_2 = \|\xi_{\text{init}}\|_2 \leq O\left(\sqrt{d}\hat{\sigma}_{\text{init}}\right) = O\left(\frac{(R^2 + \Gamma^2)d^{3/2}\sqrt{\log n}\log^2(dmn)}{n\epsilon}\right) \tag{25}$$

with probability at least $1 - 2e^{-10\log n}$.

By Theorem 38, we have

$$\|Z - \bar{Z}\|_2 \leq O\left(\log^3(nd)\sqrt{\frac{(R^2 + \Gamma^2)\Gamma^2 d}{mn}}\right) \tag{26}$$

with probability over $1 - O(n^{-10})$. Therefore, by (25) and (26), we have

$$\|\hat{Z} - \bar{Z}\|_2 \leq \|\hat{Z} - Z\|_2 + \|\hat{Z} - Z\|_2$$
$$\leq O\left(\frac{(R^2 + \Gamma^2)d^{3/2}\sqrt{\log n}\log^2(dmn)}{n\epsilon} + \log^3(nd)\sqrt{\frac{(R^2 + \Gamma^2)\Gamma^2 d}{mn}}\right) \tag{27}$$

with probability at least $1 - O(n^{-10})$ via the union bound. Finally, using Theorem 37 and the fact that $\bar{Z} = \frac{1}{n}\sum_{i=1}^{n}(U^*v_i^*)(U^*v_i^*)^\top$ with (27), we obtain

$$\text{dist}(U_{\text{init}}, U^*) \leq \frac{2\|\hat{Z} - \bar{Z}\|_2}{\sigma_{\min,*}^2}$$

$$\leq O\left(\frac{(R^2 + \Gamma^2)d^{3/2}\sqrt{\log n}\log^2(dmn)}{n\epsilon\sigma_{\min,*}^2} + \log^3(nd)\sqrt{\frac{(R^2+\Gamma^2)\Gamma^2 d}{mn\sigma_{\min,*}^4}}\right)$$

with probability at least $1 - O(n^{-10})$.

$\square$

### B.3 Auxiliary lemmas

The results of this section include multiple adaptations of those from (Collins et al., 2021) such as Lemma 43 and Lemma 33. Our proofs, when they are adaptations, are substantially more complex due to the addition of label noise and differential privacy to design Private FedRep (Algorithm 1).

This section has the following structure. We first characterize the solution of the local minimization step of Algorithm 1 in Lemma 39. Next, we introduce in Proposition 40 terms quantifying the label noise terms that periodically appear throughout our proofs. Using this proposition, we give a bound on the error from estimating $v_1^*, \ldots, v_n^*$ incurred during the local minimization step of Algorithm 1 in Lemma 41. Using similar methods, in Lemma 43 we bound the spectral distance of the linear operator that defines the gradient step of Algorithm 1 from the identity operator. Finally, using all of these intermediate results allows us to prove Lemma 33, a key lemma in our main proof of Section B.1.

Take $\mathcal{I}_t, \mathcal{I}_t'$ to be the index sets of our batches in Algorithm 1. Let $B_{i,t} = \{(x_{i,j}, y_{i,j}) : j \in \mathcal{I}_t\}$ and $B_{i,t}' = \{(x_{i,j}', y_{i,j}') : j \in \mathcal{I}_t'\}$. We omit iterations $t$ on our quantities for ease of notation. Further, we reindex the elements of $B_{i,t}, B_{i,t}'$ to $B_{i,t} = \{(x_{i,j}, y_{i,j}) : j \in [b]\}$ and $B_{i,t}' = \{(x_{i,j}', y_{i,j}') : j \in [b]\}$. Let $A_{i,j} = e_i x_{i,j}^\top$, $A_{i,j}' = e_i x_{i,j}'^\top$ for each $(i, j) \in [n] \times \mathcal{I}_t$. Define $\mathcal{A} : \mathbb{R}^{n \times d} \to \mathbb{R}^{nb}$ where $\mathcal{A}(M) = (\langle A_{i,j}, M\rangle_F)_{(i,j)\in[n]\times\mathcal{I}_t}$ for all matrices $M \in \mathbb{R}^{n \times d}$. We analogously define the operator $\mathcal{A}'$ with respect to the matrices $A_{i,j}'$.

Denote $(\mathcal{A}')^\dagger : \mathbb{R}^{nb} \to \mathbb{R}^{n \times d}$ the adjoint operator of $\mathcal{A}'$ defined as $(\mathcal{A}')^\dagger(M) = \sum_{i=1}^{n}\sum_{j=1}^{b}\langle A_{i,j}', M\rangle A_{i,j}'$. In this sense $(\mathcal{A})^\dagger\mathcal{A} : \mathbb{R}^{n \times d} \to \mathbb{R}^{n \times d}$ is a single operator. Furthermore, recall that $\xi_t \sim \mathcal{N}(0, \hat{\sigma}^2)^{d \times k}$ and choose $\hat{\sigma} = \tilde{O}\left(\frac{(R+\Gamma)\Gamma\sqrt{dk}}{n\epsilon}\right)$. Define the following recursion from Algorithm 1.

---

**Algorithm 1 recursion**

$$V_{t+1} \leftarrow \underset{V \in \mathbb{R}^{n \times k}}{\arg\min} \frac{1}{nb}\left\|\mathcal{A}(V^*(U^*)^\top - V(U_t)^\top) + \vec{\zeta}\right\|_2^2$$

$$\hat{U}_{t+1} \leftarrow U_t - \frac{\eta}{nb}\left((\mathcal{A}')^\dagger\mathcal{A}'(V_{t+1}(U_t)^\top - V^*(U^*)^\top)\right)^\top V_{t+1} \tag{28}$$

$$- \frac{2\eta}{nb}\nabla_U\langle\mathcal{A}'(V_{t+1}(U_t)^\top), \vec{\zeta}\rangle + \eta\xi_t$$

$$U_{t+1}, P_{t+1} \leftarrow \text{QR}(\hat{U}_{t+1}).$$

---

We will now state a theorem that gives an analytic form for $V_{t+1}$. Suppose $p, q \in [d]$. Let $u_{q,t}, u_q^*$ be the $q$-th column of

$U_t, U^*$, respectively. Define

$$G_{p,q} = \frac{1}{b} \sum_{i=1}^{n} \sum_{j=1}^{b} \left( A_{i,j} u_{p,t} u_{q,t}^\top (A_{i,j})^\top \right) \in \mathbb{R}^{n \times n}$$

$$C_{p,q} = \frac{1}{b} \sum_{i=1}^{n} \sum_{j=1}^{b} \left( A_{i,j} u_{p,t} (u_q^*)^\top (A_{i,j})^\top \right) \in \mathbb{R}^{n \times n}$$

$$(29)$$

$$D_{p,q} = \langle u_{p,t}, u_q^* \rangle I_{n \times n} \in \mathbb{R}^{n \times n}$$

$$W_p = \frac{2}{b} \sum_{i=1}^{n} \sum_{j=1}^{b} \zeta_{i,j} A_{i,j} u_{p,t} \in \mathbb{R}^n.$$

Take $G, C, D$ to be $nk \times nk$ block matrices with blocks $G_{p,q}, C_{p,q}, C_{p,q}$ and $W$ an $nk$-dimensional vector created by concatenating $W_p$ for each $p \in [k]$. Denote $\widetilde{v}^* = \text{Vec}(V^*)$, a column vector of dimension $nk$.

For the following lemma we define

$$F = [(G^{-1}((GD - C)\widetilde{v}^* + W))_1 \dots (G^{-1}((GD - C)\widetilde{v}^* + W))_k] \in \mathbb{R}^{n \times k} \tag{30}$$

where $(G^{-1}((GD - C)\widetilde{v}^* + W))_p$ is the $p$-th $n$-dimensional block of the $nk$-dimensional vector $G^{-1}((GD - C)\widetilde{v}^* + W)$.

**Lemma 39.** *The matrix $V_{t+1}$ satisfies*

$$V_{t+1} = V^*(U^*)^\top U_t - F$$

*at each iteration $t + 1$ for the error matrix $F \in \mathbb{R}^{n \times k}$.*

*Proof.* Note that $V_{t+1}$ minimizes the function $F(V, U_t) = \frac{2}{nb} \left\| \mathcal{A}(V^*(U^*)^\top - V(U_t)^\top) + \zeta \right\|_2^2$ and so $\nabla_{v_p} F(V_{t+1}, U_t) = 0$ for $v_p$ the $p$-th column of $V$ for each $p \in [k]$. Recall our definition of $A_{i,j} = e_i x_{i,j}^\top$ for $e_i$ the $i$-th $n$-dimensional standard basis vector. Given that

$$\left\| \mathcal{A}(V^*(U^*)^\top - V(U_t)^\top) + \zeta \right\|_2^2 = \left\| \mathcal{A}(V^*(U^*)^\top - V(U_t)^\top) \right\|_2^2 + 2\langle \mathcal{A}(V^*(U^*)^\top - V(U_t)^\top), \zeta \rangle + \|\zeta\|_2^2$$

we have for $u_{q,t}$ the $q$-th column of $U_t$

$$0 = \nabla_{v_p} F(V_{t+1}, U_t)$$
$$= \frac{2}{nb} \sum_{q=1}^{k} \sum_{i=1}^{n} \sum_{j=1}^{b} \left( u_{q,t}^\top (A_{i,j})^\top v_{q,t+1} - (u_q^*)^\top (A_{i,j})^\top v_q^* \right) A_{i,j} u_{p,t} + \frac{4}{nb} \nabla_{v_p} \langle \mathcal{A}(V^*(U^*)^\top - V_{t+1}(U_t)^\top), \zeta \rangle.$$

Let $(M)_{*,p}$ be the $p$-th column of a matrix $M$. Then

$$\nabla_{v_p} \langle \mathcal{A}(V^*(U^*)^\top - V_{t+1}(U_t)^\top), \zeta \rangle = -\nabla_{v_p} \langle \mathcal{A}(V_{t+1}(U_t)^\top), \zeta \rangle$$
$$= -\nabla_{v_p} \langle ((\langle A_{i,j}, V_{t+1}(U_t)^\top \rangle_F)_{(i,j) \in [n] \times \mathcal{I}_t}, \zeta \rangle$$
$$= -\nabla_{v_p} \sum_{(i,j) \in [n] \times [b]} \zeta_{i,j} \langle A_{i,j}, V_{t+1}(U_t)^\top \rangle_F$$
$$= -\nabla_{v_p} \left\langle \sum_{(i,j) \in [n] \times [b]} \zeta_{i,j} A_{i,j}, V_{t+1}(U_t)^\top \right\rangle_F$$
$$= -\nabla_{v_p} \left\langle \sum_{(i,j) \in [n] \times [b]} \zeta_{i,j} A_{i,j} U_t, V_{t+1} \right\rangle_F$$
$$= \left( -\sum_{(i,j) \in [n] \times [b]} \zeta_{i,j} A_{i,j} U_t \right)_{*,p}$$

the $p$-th column of the matrix $-\sum_{(i,j)\in[n]\times[b]}\zeta_{i,j}A_{i,j}U_t \in \mathbb{R}^{n\times k}$. Hence

$$\frac{1}{b}\sum_{q=1}^{k}\sum_{i=1}^{n}\sum_{j=1}^{b}\left(A_{i,j}u_{p,t}u_{q,t}^{\top}(A_{i,j})^{\top}\right)v_{q,t+1} = \frac{1}{b}\sum_{q=1}^{k}\sum_{i=1}^{n}\sum_{j=1}^{b}\left(A_{i,j}u_{p,t}(u_q^*)^{\top}(A_{i,j})^{\top}\right)v_q^* + \frac{2}{b}\sum_{i=1}^{n}\sum_{j=1}^{b}\zeta_{i,j}A_{i,j}u_{p,t}.$$

Then, for $\widetilde{v}_{t+1} = (v_{1,t+1}^{\top},\ldots,v_{k,t+1}^{\top})^{\top} \in \mathbb{R}^{nk}$ and $\widetilde{v}^* = ((v_1^*)^{\top},\ldots,(v_k^*)^{\top})^{\top} \in \mathbb{R}^{nk}$ we have

$$\widetilde{v}_{t+1} = G^{-1}C(\widetilde{v}^* + W) = D\widetilde{v}^* - G^{-1}((GD - C)\widetilde{v}^* + W)$$

conditioned on the event that $G^{-1}$ exists. We denote $F = [(G^{-1}((GD - C)\widetilde{v}^* + W))_1 \ldots (G^{-1}((GD - C)\widetilde{v}^* + W))_k] \in \mathbb{R}^{n\times k}$ where $(G^{-1}((GD - C)\widetilde{v}^* + W))_p$ is the $p$-th $n$-th dimensional block of the $nk$-dimensional vector $G^{-1}((GD - C)\widetilde{v}^* + W)$. Recalling the definition of $D$, we have that $V_{t+1} = V^*(U^*)^{\top}U_t - F$. □

In order to evaluate the final bounds with label noise included we must bound the following terms

$$\frac{1}{b}\nabla_V\langle\mathcal{A}(V_{t+1}(U_t)^{\top}),\vec{\zeta}\rangle$$

and

$$\frac{1}{nb}\nabla_U\langle\mathcal{A}(V_{t+1}(U_t)^{\top}),\vec{\zeta}\rangle$$

in spectral norm.

**Proposition 40.** *With probability $1 - O(n^{-11})$, we have*

$$(1)\quad \left\|\frac{1}{b}\nabla_V\langle\mathcal{A}(V_{t+1}(U_t)^{\top}),\vec{\zeta}\rangle\right\|_2 \leq \sqrt{\frac{26R^2n\log(nb)}{b}}.$$

*Furthermore, with probability at least $1 - O(n^{-10})$, we have*

$$(2)\quad \left\|\frac{1}{nb}\nabla_U\langle\mathcal{A}(V_{t+1}(U_t)^{\top}),\vec{\zeta}\rangle\right\|_2 \leq \frac{4}{3}\sqrt{\frac{2\cdot 15R^2\Gamma^2 d\log n}{nb}}.$$

*Proof.* **Claim 1:** With probability $1 - e^{-11\log(nb)}$, we have

$$\left\|\frac{1}{b}\nabla_V\langle\mathcal{A}(V_{t+1}(U_t)^{\top}),\vec{\zeta}\rangle\right\|_2 \leq \sqrt{\frac{26R^2n\log(nb)}{b}}.$$

Note that for any given $p \in [n]$ and $q \in [k]$

$$\left(\frac{1}{b}\nabla_V\langle\mathcal{A}(V_{t+1}(U_t)^{\top}),\vec{\zeta}\rangle\right)_{p,q} = \left(\frac{1}{b}\nabla_V\langle(\langle e_ix_{i,j}^{\top},V_{t+1}(U_t)^{\top}\rangle_F)_{(i,j)\in[n]\times[m]},\vec{\zeta}\rangle\right)_{p,q}$$

$$= \left(\frac{1}{b}\sum_{(i,j)\in[n]\times[m]}\zeta_{i,j}\nabla_V(\langle e_ix_{i,j}^{\top},V_{t+1}(U_t)^{\top}\rangle_F)_{(i,j)}\right)_{p,q}$$

$$= \left(\frac{1}{b}\sum_{(i,j)\in[n]\times[m]}\zeta_{i,j}\nabla_V(\langle e_ix_{i,j}^{\top}U_t,V_{t+1}\rangle_F)_{(i,j)}\right)_{p,q}$$

$$= \left(\frac{1}{b}\sum_{(i,j)\in[n]\times[m]}\zeta_{i,j}e_ix_{i,j}^{\top}U_t\right)_{p,q}$$

$$= \frac{1}{b}\sum_{j=1}^{b}\zeta_{p,j}\langle x_{p,j},u_{q,t}\rangle$$

for $u_{q,t}$ the $q$-th column of $U_t$. Observe that $\zeta_{p,j}$ is independent of both $x_{p,j}$ and $u_{q,t}$ for all $j \in [m]$. Condition on the event

$$\mathcal{E} = \{|\zeta_{i,j}| \leq R\sqrt{26\log(nb)} \text{ for all } (i,j) \in [n] \times [m]\}$$

which has probability at least $1 - e^{-13\log(nb)}$. Via this conditioning the random variable $\zeta_{p,j}\langle x_{p,j}, u_{q,t}\rangle$ is $R\sqrt{26\log(nb)}$-sub-Gaussian given that $\langle x_{p,j}, u_{q,t}\rangle$ is 1-sub-Gaussian. Note as well $\zeta_{p,j}\langle x_{p,j}, u_{q,t}\rangle$ is mean zero and independent for each $j$. So, $\frac{1}{b}\sum_{j=1}^b \zeta_{p,j}\langle x_{p,j}, u_{q,t}\rangle$ is centered, $\sqrt{\frac{26R^2\log(nb)}{b}}$-sub-Gaussian, and independent for every $p$.

Let $\bar{\sigma}^2$ be the variance of $\frac{1}{b}\sum_{j=1}^b \zeta_{p,j}\langle x_{p,j}, u_{q,t}\rangle$, which has $\bar{\sigma} \leq \sqrt{\frac{26R^2\log(nb)}{b}}$. Then, $\left(\frac{1}{m\bar{\sigma}}\sum_{j=1}^b \zeta_{p,j}\langle x_{p,j}, u_{q,t}\rangle\right)_{q\in[k]}$ has covariance matrix $I_k$. By the one-sided version of Theorem 26

$$\sigma_{\max}\left(\frac{1}{m\bar{\sigma}}\nabla_V\langle\mathcal{A}(V_{t+1}(U_t)^\top),\vec{\zeta}\rangle\right) \leq O\left(\sqrt{n} + \sqrt{k} + w\right)$$

with probability at least $1 - e^{-\alpha^2}$. Multiplying through by $\bar{\sigma}$ and setting $\alpha = \sqrt{n}$, we have

$$\left\|\frac{1}{b}\nabla_V\langle\mathcal{A}(V_{t+1}(U_t)^\top),\vec{\zeta}\rangle\right\|_2 \leq \sqrt{\frac{26R^2 n\log(nb)}{b}}$$

with probability at least $1 - e^{-n}$ conditioned on the event $\mathcal{E}$. So, in general

$$\left\|\frac{1}{b}\nabla_V\langle\mathcal{A}(V_{t+1}(U_t)^\top),\vec{\zeta}\rangle\right\|_2 \leq \sqrt{\frac{26R^2 n\log(nb)}{b}}$$

with probability at least $1 - e^{-n} - e^{-12\log(nb)} \geq 1 - e^{-11\log(nb)}$. This proves our first claim.

**Claim 2:** With probability at least $1 - O(n^{-10})$, we have

$$\left\|\frac{1}{nb}\nabla_U\langle\mathcal{A}(V_{t+1}(U_t)^\top),\vec{\zeta}\rangle\right\|_2 \leq \frac{4}{3}\sqrt{\frac{2\cdot 15R^2\Gamma^2 d\log n}{nb}}.$$

Note that

$$\frac{1}{nb}\nabla_U\langle\mathcal{A}(V_{t+1}(U_t)^\top),\vec{\zeta}\rangle = \nabla_U\left\langle\frac{1}{nb}\sum_{i=1}^n\sum_{j=1}^b \zeta_{i,j}A_{i,j}, V_{t+1}(U_t)^\top\right\rangle_F$$

$$= \nabla_U\left\langle\frac{1}{nb}\sum_{i=1}^n\sum_{j=1}^b \zeta_{i,j}(V_{t+1})^\top A_{i,j}, (U_t)^\top\right\rangle_F$$

$$= \nabla_U\left\langle\frac{1}{nb}\sum_{i=1}^n\sum_{j=1}^b \zeta_{i,j}(A_{i,j})^\top V_{t+1}, U_t\right\rangle_F$$

$$= \frac{1}{nb}\sum_{i=1}^n\sum_{j=1}^b \zeta_{i,j}(A_{i,j})^\top V_{t+1}.$$

Observe that since $(A_{i,j})^\top$ is a matrix with one non-zero column $x'_{i,j}$, we have $(A_{i,j})^\top V_{t+1} = x'_{i,j}(v_{i,t+1})^\top$ where $v_{i,t+1}$ is the $i$-th row of $V_{t+1}$. Let $\mathcal{N}^d$ and $\mathcal{N}^k$ be Euclidean $\frac{1}{4}$-covers of the $d$ and $k$-dimensional unit spheres, respectively. Then, by Lemma 23

$$\left\|\frac{1}{nb}\sum_{i=1}^n\sum_{j=1}^b \zeta_{i,j}(A_{i,j})^\top V_{t+1}\right\|_2 = \left\|\frac{1}{nb}\sum_{i=1}^n\sum_{j=1}^b \zeta_{i,j}x'_{i,j}(v_{i,t+1})^\top\right\|_2$$

$$= \frac{4}{3}\max_{a\in\mathcal{N}^d, b\in\mathcal{N}^k} a^\top\left(\frac{1}{nb}\sum_{i=1}^n\sum_{j=1}^b \zeta_{i,j}x'_{i,j}(v_{i,t+1})^\top\right)b$$

$$= \frac{4}{3}\max_{a\in\mathcal{N}^d, b\in\mathcal{N}^k}\left(\frac{1}{nb}\sum_{i=1}^n\sum_{j=1}^b \zeta_{i,j}\langle a, x'_{i,j}\rangle\langle v_{i,t+1}, b\rangle\right).$$

Let $\|v_{i,t+1}\|_2 \le \frac{5}{4}\Gamma$ for all $i$ with probability at least $1 - O(n^{-14})$ via Proposition 42. As well, we condition on the event

$$\mathcal{E} = \left\{ \|v_{i,t+1}\|_2 \le \frac{5}{4}\Gamma \text{ for all } i \text{ simultaneously} \right\}$$

which has probability $\mathbb{P}[\mathcal{E}] \ge 1 - O(n^{-13})$.

Since $|\langle v_{i,t+1}, b \rangle| \le \frac{25}{16}\Gamma$ by Cauchy-Schwarz, the random variable $\zeta_{i,j}\langle a, x'_{i,j}\rangle\langle v_{i,t+1}, b\rangle$ is sub-exponential. Furthermore, the variable $\frac{1}{nb}\zeta_{i,j}\langle a, x'_{i,j}\rangle\langle v_{i,t+1}, b\rangle$ has sub-exponential norm bounded by $\frac{125R\Gamma}{64nb} \le \frac{2R\Gamma}{nb}$ and is mean zero. Following from Lemma 22 conditioned on $\mathcal{E}$ and for $nb \ge 15d\log n$, we have

$$\frac{4}{3nb} \sum_{i=1}^{n} \sum_{j=1}^{b} \zeta_{i,j}\langle a, x'_{i,j}\rangle\langle v_{i,t+1}, b\rangle \le \frac{4}{3}\sqrt{\frac{2 \cdot 15R^2\Gamma^2 d\log n}{nb}}$$

with probability at least $1 - e^{-15d\log n}$. Via the union bound over $\mathcal{N}^d, \mathcal{N}^k$

$$\left\| \frac{1}{nb} \sum_{i=1}^{n} \sum_{j=1}^{b} \zeta_{i,j}(A_{i,j})^\top V_{t+1} \right\|_2 \le \frac{4}{3} \max_{a \in \mathcal{N}^d, b \in \mathcal{N}^k} \frac{1}{nb} \sum_{i=1}^{n} \sum_{j=1}^{b} \zeta_{i,j}\langle a, x'_{i,j}\rangle\langle v_{i,t+1}, b\rangle$$

$$\le \frac{4}{3}\sqrt{\frac{2 \cdot 15R^2\Gamma^2 d\log n}{nb}}$$

with probability at least $1 - 9^{d+k}e^{-15d\log n} \ge 1 - e^{-10d\log n}$. Let $E$ be the event

$$E = \left\{ \left\| \frac{1}{nb} \sum_{i=1}^{n} \sum_{j=1}^{b} \zeta_{i,j}(A_{i,j})^\top V_{t+1} \right\|_2 \le \frac{4}{3}\sqrt{\frac{2 \cdot 15R^2\Gamma^2 d\log n}{nb}} \right\}.$$

Applying the fact that $\mathbb{P}[E^c] \le \mathbb{P}[E^c|\mathcal{E}] + \mathbb{P}[\mathcal{E}^c] \le e^{-10d\log n} + O(n^{-13}) = O(n^{-10})$ finishes the second claim. $\qquad\square$

Recall the following definitions given prior to Lemma 39 where $A_{i,j} = e_i x_{i,j}^\top$. Denote

$$G_{p,q} = \frac{1}{b} \sum_{i=1}^{n} \sum_{j=1}^{b} \left( A_{i,j} u_{p,t} u_{q,t}^\top (A_{i,j})^\top \right) \in \mathbb{R}^{n \times n}$$

$$C_{p,q} = \frac{1}{b} \sum_{i=1}^{n} \sum_{j=1}^{b} \left( A_{i,j} u_{p,t} (u_q^*)^\top (A_{i,j})^\top \right) \in \mathbb{R}^{n \times n} \tag{31}$$

$$D_{p,q} = \langle u_{p,t}, u_q^* \rangle I_{n \times n} \in \mathbb{R}^{n \times n}$$

$$W_p = \frac{2}{b} \sum_{i=1}^{n} \sum_{j=1}^{b} \zeta_{i,j} A_{i,j} u_{p,t} \in \mathbb{R}^n.$$

Take $G, C, D$ to be $nk \times nk$ block matrices with blocks $G_{p,q}, C_{p,q}, C_{p,q}$ and $W$ an $nk$-dimensional vector created by concatenating $W_p$ for each $p \in [k]$. Let $G^i, C^i, D^i$ be the $k \times k$ matrices formed by taking the $i$-th diagonal element from each $G_{p,q}, C_{p,q}, D_{p,q}$, respectively.

**Lemma 41.** *Let $\tau_k = c_\tau\sqrt{\frac{35k\log n}{b}}$ for some $c > 0$. Then*

$$(1) \quad \|G^{-1}\|_2 \le \frac{1}{1 - \tau_k}$$

*with probability at least $1 - O(n^{-13})$.*

*Furthermore,*

$$(2) \quad \|F\|_F \le \frac{\nu\tau_k}{1 - \tau_k}\|V^*\|_2\mathrm{dist}(U_t, U^*) + \sqrt{\frac{26R^2 nk\log(nb)}{(1 - \tau_k)^2 b}}$$

*with probability at least $1 - O(n^{-11})$.*

*Proof.* **Claim 1:** We have

$$\|G^{-1}\|_2 \le \frac{1}{1 - \tau_k}$$

with probability at least $1 - e^{-13k \log n}$.

Let $a$ be a normalized vector in $nk$ dimensions. Define $a^i \in \mathbb{R}^k$ to be the sub-vector of $a$ constructed by choosing each $((p-1)n + i)$-th component for $p = 1, \ldots, k$. Observe that

$$\sigma_{\min}(G) = \min_{a:\|a\|_2=1} a^\top G a$$

$$= \min_{a:\|a\|_2=1} \sum_{i=1}^{n} a^{i^\top} G^i a^i$$

$$\ge \min_{i \in [n]} \sigma_{\min}(G^i).$$

Let $\Pi^i = \frac{1}{b} \sum_{j=1}^{b} x_{i,j} x_{i,j}^\top$. By our definition below (31), we have $G^i = U_t^\top \Pi^i U_t$. Note that $\frac{1}{\sqrt{b}} U_t^\top x_{i,j}$ is $\frac{1}{\sqrt{b}}$-sub-Gaussian and independent for each $i, j$. Assume $b \ge k$. Then, using a one-sided form of Theorem 26, there exists a constant $c_\tau > 0$ where

$$\sigma_{\min}(U_t^\top \Pi^i U_t) \ge 1 - c_\tau \left( \sqrt{\frac{k}{b}} + \frac{w}{\sqrt{b}} \right)$$

with probability at least $1 - e^{-\alpha^2}$. Setting $\alpha = \sqrt{14k \log n}$ gives us

$$\sigma_{\min}(G^i) \ge 1 - \tau_k$$

with probability $1 - e^{-14k \log n}$ for $\tau_k$ as in the lemma statement. Via the union bound over $i \in [n]$

$$\sigma_{\min}(G) \ge 1 - \tau_k$$

with probability at least $1 - e^{-13k \log n}$.

**Claim 2:** We have

$$\|F\|_F \le \frac{\nu \tau_k}{1 - \tau_k} \|V^*\|_2 \mathrm{dist}(U_t, U^*) + \sqrt{\frac{26 R^2 nk \log(nb)}{(1 - \tau_k)^2 b}}$$

with probability at least $1 - 2e^{-13k \log n} - e^{-11 \log(nb)}$.

The proof follows from bounding $H^i = G^i C^i - D^i$ for each $i \in [n]$ in spectral norm with Lemma 22 and exploiting the definition of our parameter $\nu = \frac{\max_{i \in [n]} \|v_i^*\|_2}{\sigma_{\max,*}}$.

Let $X_i$ be the design matrix for $x_{i,1}, \ldots, x_{i,b}$. Note that, by the definitions below (31), we have

$$G^i D^i - C^i = (U_t)^\top \Pi^i U_t (U_t)_t^\top U^* - U^\top \Pi^i U^* = \frac{1}{b} (U_t)^\top X_i^\top X_i (U_t (U_t)^\top - I_d) U^*.$$

Then

$$\|(GD - C)\mathrm{Vec}(V^*)\|_2^2 = \sum_{i=1}^{n} \|H^i (v_i^*)^\top\|_2^2$$

$$\le \sum_{i=1}^{n} \|H^i\|_2^2 \|v_i^*\|_2^2$$

$$\le \frac{\nu^2}{n} \|V^*\|_2^2 \sum_{i=1}^{n} \|H^i\|_2^2.$$

and so

$$\|(GD - C)\mathrm{Vec}(V^*)\|_2^2 \le \frac{\nu^2}{n} \|V^*\|_2^2 \sum_{i=1}^{n} \|H^i\|_2^2. \tag{32}$$

We now bound each $H^i$ using concentration inequalities. Define $A = \frac{1}{\sqrt{b}} X_i U_t$ and $B = \frac{1}{\sqrt{b}} X_i (U_t(U_t)^\top - I_d) U^*$. We denote the rows of $A$ and $B$ with $a_{i,j} = \frac{1}{\sqrt{b}}(U_t)^\top x_{i,j}$ and $b_{i,j} = \frac{1}{\sqrt{b}} U^* (U_t(U_t)^\top - I_d) x_{i,j}$, respectively. Note that, for $\mathcal{N}^k$ a Euclidean $\frac{1}{4}$-cover of the unit sphere in $k$ dimensions, we have

$$
\|B^\top A\|_2 \le 2 \max_{u,u' \in \mathcal{N}^k} u^\top B^\top A u'
$$

$$
= 2 \max_{u,u' \in \mathcal{N}^k} u^\top \left( \sum_{j=1}^{b} b_{i,j} a_{i,j}^\top \right) u'
$$

$$
= 2 \max_{u,u' \in \mathcal{N}^k} \sum_{j=1}^{b} \langle u, b_{i,j} \rangle \langle a_{i,j}, u' \rangle
$$

via Lemma 23. Now, fix some $(u, u') \in \mathcal{N}^k \times \mathcal{N}^k$. Then, $a_{i,j} = \frac{1}{\sqrt{b}}(U_t)^\top x_{i,j}$ and $b_{i,j} = \frac{1}{\sqrt{b}} U^* (U_t(U_t)^\top - I_d) x_{i,j}$. So, $\langle u, a_{i,j} \rangle$ is $\frac{5}{4\sqrt{b}}$-sub-Gaussian and $\langle b_{i,j}, u' \rangle$ is $\frac{5}{4\sqrt{b}}\mathrm{dist}(U_t, U^*)$-sub-Gaussian. This means their product is $\frac{25}{16b}\mathrm{dist}(U_t, U^*)$-sub-exponential.

Now, from Lemma 22, there exists $c' > 0$ such that

$$
\mathbb{P}\left[ \|H^i\|_2^2 \ge 2s \right] \le \mathbb{P}\left[ \max_{u,u' \in \mathcal{N}^k} \sum_{j=1}^{b} \langle u, b_{i,j} \rangle \langle a_{i,j}, u' \rangle \ge s \right]
$$

$$
\le 9^{2k} e^{-c'b \min\left( \frac{s^2}{2.5\mathrm{dist}^2(U_t, U^*)}, \frac{s}{1.6\mathrm{dist}(U_t, U^*)} \right)}.
$$

Let $\tau > 0$ satisfy $\frac{s}{1.6\mathrm{dist}(U_t, U^*)} = \max(\tau, \tau^2)$. Then

$$
\tau^2 = \min\left( \frac{s^2}{2.5\mathrm{dist}^2(U_t, U^*)}, \frac{s}{1.6\mathrm{dist}(U_t, U^*)} \right).
$$

Picking $\tau^2 = \frac{14k \log n}{c'b}$ and assuming that $b \ge 11k \log n$ ensures

$$
\mathbb{P}\left[ \|H^i\|_2 \ge \sqrt{\frac{35\mathrm{dist}^2(U_t, U^*)k \log n}{b}} \right] \le e^{-14k \log n} \tag{33}
$$

for any fixed $i \in [n]$. So

$$
\mathbb{P}\left[ \|(GD - C)\mathrm{Vec}(V^*)\|_2^2 \ge 35\nu^2 \|V^*\|_2^2 \mathrm{dist}^2(U_t, U^*) \frac{k \log n}{b} \right]
$$

$$
\le \mathbb{P}\left[ \frac{\nu^2}{n} \|V^*\|_2^2 \sum_{i=1}^{n} \|H^i\|_2^2 \ge 35\nu^2 \|V^*\|_2^2 \mathrm{dist}^2(U_t, U^*) \frac{k \log n}{b} \right]
$$

$$
\le \mathbb{P}\left[ \frac{\nu^2}{n} \sum_{i=1}^{n} \|H^i\|_2^2 \ge 35\nu^2 \mathrm{dist}^2(U_t, U^*) \frac{k \log n}{b} \right]
$$

$$
\le n\mathbb{P}\left[ \|H^1\|_2^2 \ge 35\mathrm{dist}^2(U_t, U^*) \frac{k \log n}{b} \right]
$$

$$
\le e^{-13k \log n}
$$

where the first inequality follows from (32) and the last inequality follows from (33). The rest of the proof follows from the fact that $F = [(G^{-1}((GD - C)\widetilde{v}^* + W))_1 \dots (G^{-1}((GD - C)\widetilde{v}^* + W))_k]$. That is, we bound $G^{-1}[W_1, \dots, W_k]$ via an application of Claim 1 along with Proposition 40 part (1). This gives a bound on the norm of $F = [G^{-1}((GD - C)\widetilde{v}^*)_1 \dots G^{-1}((GD - C)\widetilde{v}^*)_k] + G^{-1}[W_1, \dots, W_k]$ by the union bound. $\qquad\square$

Let $f_i$ be the $i$-th row of $F$ in row vector form. Also, recall that $G^i, C^i, D^i$ are the $k \times k$ matrices formed by taking the $i$-th diagonal element from each $G_{p,q}, C_{p,q}, D_{p,q}$ as in (31).

**Proposition 42.** *Suppose Assumption 31 holds. For all $t \in [T-1]$, we have that $v_{i,t+1}$ from Algorithm 1 satisfies*

$$\|v_{i,t+1}\|_2 \le \frac{5}{4}\Gamma$$

*with probability at least $1 - O(n^{-14})$ for all $i \in [n]$.*

*Proof.* By Lemma 39, we have (as row vectors) $v_{i,t+1} = v_i^*(U^*)^\top U_t - f_i$. This implies

$$\|v_{i,t+1}\|_2 \le \|v_i^*\|_2\|(U^*)^\top U_t\|_2 + \|f_i\|_2$$

and so $\|v_{i,t+1}\|_2 \le \Gamma + \|f_i\|_2$. Fix $i \in [n]$ and assume that $\nu\tau_k < 1$. This is not difficult to achieve when using Assumption 31. Recall that we defined $\tau_k = c_\tau\sqrt{\frac{35k\log n}{b}}$. Now, denoting the $i$-th row of a matrix with $(M)_{i,*}$

$$\|f_i\|_2 = \left\| G^{i^{-1}}(G^iC^i - D^i)(v_i^*)^\top + \left( \frac{1}{b}\sum_{i=1}^n\sum_{j=1}^b \zeta_{i,j}G^{i^{-1}}A_{i,j}U_t \right)^\top_{i,*} \right\|_2$$

$$\le \|G^{i^{-1}}\|_2\|(G^iC^i - D^i)(v_i^*)^\top\|_2 + \|G^{i^{-1}}\|_2 \left\| \left( \frac{1}{b}\sum_{j=1}^b \zeta_{i,j}x_{i,j}^\top U_t \right)^\top_{i,*} \right\|_2$$

$$\le \frac{\nu\tau_k}{1-\tau_k}\|v_i^*\|_2 \operatorname{dist}(U_t, U^*) + \sqrt{\frac{40R^2k\log n}{(1-\tau_k)^2 b}}$$

$$\le \frac{\nu\tau_k}{1-\tau_k}\Gamma + \sqrt{\frac{40R^2k\log n}{(1-\tau_k)^2 b}}$$

with probability at least $1 - 2e^{-14k\log(n)} - e^{-14\log(nb)}$ via the argument of Lemma 41 part (1), and the same arguments as Lemma 41 part (2) and Proposition 40 part (1) applied to the vectors $G^{i^{-1}}(G^iC^i - D^i)(v_i^*)^\top$ and $\left( \frac{1}{b}\sum_{j=1}^b \zeta_{i,j}x_{i,j}^\top U_t \right)_{i,*}$.

Now, assuming that $m \ge 4000c_\tau^2\frac{\max\{R^2,1\}\cdot\max\{\Gamma^2,1\}\gamma^4 k\log^2 n}{E_0^2\sigma_{\max,*}^2}T$ and $m \ge 4000\frac{R^2k}{\Gamma^2}T\log(nm)$, we have

$$\frac{\nu\tau_k}{1-\tau_k}\Gamma + \sqrt{\frac{40R^2k\log n}{(1-\tau_k)^2 b}} \le \frac{\Gamma}{4}.$$

Taking the union bound over all $i$ finishes the result. $\square$

**Lemma 43.** *Let $\tau_k' = 5\sqrt{13}\sqrt{\frac{\Gamma^4 d\log n}{nb}}$. Suppose Assumption 31 holds. Then, we have, for any iteration $t$, that*

$$\frac{1}{n}\left\| \left( \frac{1}{b}(\mathcal{A}')^\dagger\mathcal{A}'(V_{t+1}(U_t)^\top - V^*(U^*)^\top) - (V_{t+1}(U_t)^\top - V^*(U^*)^\top) \right)^\top V_{t+1} \right\|_2$$

$$\le \tau_k'\operatorname{dist}(U_t, U^*) + \frac{3R\Gamma^2\sqrt{15\cdot 24dk\log(n)\log(nb)}}{(nb)^{\frac{3}{4}}}$$

*with probability at least $1 - O(n^{-10})$.*

*Proof.* Take $W = (W_1, \ldots, W_k)^\top \in \mathbb{R}^{nk}$ where $W_p = \frac{2}{b}\sum_{i=1}^n\sum_{j=1}^b \zeta_{i,j}A_{i,j}u_{p,t}$. Recall that $F = [(G^{-1}((GD - C)\widetilde{v}^* + W))_1 \ldots (G^{-1}((GD - C)\widetilde{v}^* + W))_k] \in \mathbb{R}^{n\times k}$ by (30). Define $\widetilde{W} = [G^{-1}W_1 \ldots G^{-1}W_k]$. Let $Q_t$ be the matrix defined via rows $q_i$ where

$$q_i^\top = U_t(U_t)^\top U^*(v_i^*)^\top - U_t(f_i)^\top - U^*(v_i^*)^\top.$$

Finally, define by $\widetilde{Q}_t$ the matrix with rows $\widetilde{q}_{i,t} = q_{i,t} - \widetilde{W}_{i,*}(U_t)^\top$ where $\widetilde{W}_{i,*}$ is the $i$-th row of $\widetilde{W}$. Note that

$$\frac{1}{n}\left\|\left(\frac{1}{b}(\mathcal{A}')^\dagger\mathcal{A}'(V_{t+1}(U_t)^\top - V^*(U^*)^\top) - (V_{t+1}(U_t)^\top - V^*(U^*)^\top)\right)^\top V_{t+1}\right\|_2$$

$$\leq \frac{1}{n}\left\|\left(\frac{1}{b}(\mathcal{A}')^\dagger\mathcal{A}'(\widetilde{Q}_t) - \widetilde{Q}_t\right)^\top V_{t+1}\right\|_2 + \frac{1}{n}\left\|\left(\frac{1}{b}(\mathcal{A}')^\dagger\mathcal{A}'(\widetilde{W}_i(U_t)^\top) - \widetilde{W}_i(U_t)^\top\right)^\top V_{t+1}\right\|_2. \tag{34}$$

The lemma follows from bounding the right-hand terms individually.

Let $\tau_k' = 5\sqrt{13}\sqrt{\frac{\Gamma^4 d\log n}{nb}}$.

**Claim 1:** We have

$$\frac{1}{n}\left\|\left(\frac{1}{b}(\mathcal{A}')^\dagger\mathcal{A}'(\widetilde{Q}_t) - \widetilde{Q}_t\right)^\top V_{t+1}\right\|_2 \leq \tau_k'\mathrm{dist}(U_t, U^*)$$

with probability at least $1 - e^{-10d\log n} - 2e^{-13k\log n}$.

Note that

$$\|\widetilde{q}_i\|_2 \leq \|U_t(U_t)^\top U^*(v_i^*)^\top - U^*(v_i^*)^\top\|_2 + \|U_t(f_i)^\top - U_t(\widetilde{W}_i)^\top\|_2$$

$$\leq \Gamma\mathrm{dist}(U_t, U^*) + \|f_i - \widetilde{W}_i\|_2.$$

Now

$$\|f_i - \widetilde{W}_i\|_2 = \|{G^i}^{-1}\|_2\|G^iD^i - C^i\|_2\|(v_i^*)^\top\|_2$$

$$\leq \mathrm{dist}(U_t, U^*)\frac{\nu\Gamma\tau_k}{1 - \tau_k}$$

with probability at least $1 - 2e^{-13k\log n}$ via an argument identical to Lemma 41 part (2) without the label noise term. Choose $c_0 \geq 1000c^2$ in Assumption 31 to ensure $\nu\tau_k \leq \frac{1}{4}$. Then $\|f_i - \widetilde{W}_i\|_2 \leq \Gamma\mathrm{dist}(U_t, U^*)$ and hence $\|\widetilde{q}_i\|_2 \leq 2\Gamma\mathrm{dist}(U_t, U^*)$ with probability at least $1 - 2e^{-13k\log n}$.

By Proposition 42, we have $\|v_{i,t+1}\|_2 \leq \frac{5}{4}\Gamma$ with probability at least $1 - O(n^{-14})$ for each $i$. Observe

$$\frac{1}{nb}\left((\mathcal{A}')^\dagger\mathcal{A}'(\widetilde{Q}_t) - \widetilde{Q}_t\right)^\top V_{i,t+1} = \frac{1}{nb}\sum_{i=1}^n\sum_{j=1}^b\left(\langle x_{i,j}', \widetilde{q}_{i,t}\rangle x_{i,j}'(v_{i,t+1})^\top - \widetilde{q}_{i,t}(v_{i,t+1})^\top\right).$$

We first condition on the event

$$\mathcal{E} = \left\{\|\widetilde{q}_i\|_2 \leq 2\Gamma\mathrm{dist}(U_t, U^*) \text{ and } \|v_{i,t+1}\|_2 \leq \frac{5}{4}\Gamma \text{ for all } i \in [n]\right\}$$

which has probability at least $1 - O(n^{-10})$ by the union bound. Define the Euclidean $\frac{1}{4}$-covers of the $d$ and $k$-dimensional unit spheres $\mathcal{N}^d$ and $\mathcal{N}^k$, respectively. Via Lemma 23

$$\left\|\frac{1}{nb}\sum_{i=1}^n\sum_{j=1}^b\left(\langle x_{i,j}', \widetilde{q}_{i,t}\rangle x_{i,j}'(v_{i,t+1})^\top - \widetilde{q}_{i,t}(v_{i,t+1})^\top\right)\right\|_2$$

$$\leq \frac{4}{3}\max_{a\in\mathcal{N}^d, b\in\mathcal{N}^k}\frac{1}{nb}\sum_{i=1}^n\sum_{j=1}^b\left(\langle x_{i,j}', \widetilde{q}_{i,t}\rangle\langle a, x_{i,j}'\rangle\langle v_{i,t+1}, b\rangle - \langle a, \widetilde{q}_{i,t}\rangle\langle v_{i,t+1}, b\rangle\right).$$

The variable $\langle x_{i,j}', \widetilde{q}_{i,t}\rangle$ is $2\Gamma\mathrm{dist}(U_t, U^*)$-sub-Gaussian and $\langle a, x_{i,j}'\rangle$ is $\frac{5}{4}$-sub-Gaussian via Proposition 18. This means $\langle x_{i,j}', \widetilde{q}_{i,t}\rangle\langle a, x_{i,j}'\rangle$ is sub-exponential with norm $\frac{5}{2}\Gamma\mathrm{dist}(U_t, U^*)$. Then, the variable $\frac{1}{nb}\langle x_{i,j}', \widetilde{q}_{i,t}\rangle\langle a, x_{i,j}'\rangle\langle v_{i,t+1}, b\rangle$ is sub-exponential with norm

$$\frac{5\Gamma}{nb}\mathrm{dist}(U_t, U^*)\|v_i^*\|_2 \leq \frac{5\Gamma^2}{nb}\mathrm{dist}(U_t, U^*)$$

given that we have conditioned on $\mathcal{E}$. Note that the variables $\left( \langle x'_{i,j}, \widetilde{q}_{i,t} \rangle x'_{i,j}(v_{i,t+1})^\top - \widetilde{q}_{i,t}(v_{i,t+1})^\top \right)$ are centered. Furthermore, due to our conditioning we can concentrate these variables with respect to the randomness $x'_{i,j}$ since they are independent of $\widetilde{q}_{i,t}, v_{i,t+1}$. So, by Lemma 22 there exists a constant $c' > 0$ where

$$\mathbb{P}\left[ \left\| \frac{1}{nb} \sum_{i=1}^{n} \sum_{j=1}^{b} \left( \langle x'_{i,j}, \widetilde{q}_{i,t} \rangle x'_{i,j}(v_{i,t+1})^\top - \widetilde{q}_{i,t}(v_{i,t+1})^\top \right) \right\|_2 \geq 2s \, \Big| \, \mathcal{E} \right]$$
$$\leq 9^{d+k} \exp\left( -c'nb \min\left( \frac{s^2}{5^2\Gamma^4 \text{dist}(U_t, U^*)^2}, \frac{s}{5\Gamma^2 \text{dist}(U_t, U^*)} \right) \right). \tag{35}$$

From (35) we denote $\tau^2 = \min\left( \frac{s^2}{5^2\Gamma^4 \text{dist}(U_t, U^*)^2}, \frac{s}{5\Gamma^2 \text{dist}(U_t, U^*)} \right)$ and further set $\tau^2 = \frac{13(d+k)\log n}{c'nb}$. If $nb \geq \frac{13(d+k)\log n}{c'}$, then

$$\mathbb{P}\left[ \left\| \frac{1}{nb} \sum_{i=1}^{n} \sum_{j=1}^{b} \left( \langle x'_{i,j}, \widetilde{q}_{i,t} \rangle x'_{i,j}(v_{i,t+1})^\top - \widetilde{q}_{i,t}(v_{i,t+1})^\top \right) \right\|_2 \geq 5\sqrt{13}\text{dist}(U_t, U^*)\sqrt{\frac{\Gamma^4 d \log n}{nb}} \, \Big| \, \mathcal{E} \right]$$
$$\leq 9^{d+k} e^{-13(d+k)\log n} \leq e^{-10d\log n}.$$

Recall that $\mathbb{P}[\mathcal{E}] \geq 1 - O(n^{-10})$. Therefore, we have

$$\mathbb{P}\left[ \left\| \frac{1}{nb} \sum_{i=1}^{n} \sum_{j=1}^{b} \left( \langle x'_{i,j}, \widetilde{q}_{i,t} \rangle x'_{i,j}(v_{i,t+1})^\top - \widetilde{q}_{i,t}(v_{i,t+1})^\top \right) \right\|_2 \geq 5\sqrt{13}\text{dist}(U_t, U^*)\sqrt{\frac{\Gamma^4 d \log n}{nb}} \right]$$
$$\leq e^{-10d\log n} + O(n^{-10}) = O(n^{-10}). \tag{36}$$

This proves Claim 1.

**Claim 2:** We have

$$\frac{1}{n}\left\| \left( \frac{1}{b}(\mathcal{A}')^\dagger \mathcal{A}'(\widetilde{W}_i(U_t)^\top) - \widetilde{W}_i(U_t)^\top \right)^\top V_{t+1} \right\|_2 \leq \frac{3R\Gamma^2\sqrt{15 \cdot 24dk\log(n)\log(nb)}}{(nb)^{\frac{3}{4}}}$$

with probability at least $1 - O(n^{-10})$.

Observe

$$\frac{1}{nb}\left( (\mathcal{A}')^\dagger \mathcal{A}'(\widetilde{W}_i(U_t)^\top) - \widetilde{W}_i(U_t)^\top \right)^\top V_{t+1}$$
$$= \frac{1}{nb}\sum_{i=1}^{n}\sum_{j=1}^{b}\left( \langle x'_{i,j}, \widetilde{W}_i(U_t)^\top \rangle x'_{i,j}(v_{i,t+1})^\top - \widetilde{W}_i(U_t)^\top(v_{i,t+1})^\top \right).$$

Note that the random variables $\langle x'_{i,j}, \widetilde{W}_i(U_t)^\top \rangle x'_{i,j}(v_{i,t+1})^\top - \widetilde{W}_i(U_t)^\top(v_{i,t+1})^\top$ have mean zero since $x'_{i,j}$ and $\widetilde{W}_i(U_t)^\top$ are independent along with $\mathbb{E}\left[\widetilde{W}_i(U_t)^\top\right] = \vec{0}$. Using a covering argument identical to Claim 1, we have

$$\left\| \frac{1}{nb}\sum_{i=1}^{n}\sum_{j=1}^{b}\left( \langle x'_{i,j}, \widetilde{W}_i(U_t)^\top \rangle x'_{i,j}(v_{i,t+1})^\top - \widetilde{W}_i(U_t)^\top(v_{i,t+1})^\top \right) \right\|_2$$
$$\leq \frac{4}{3}\max_{a\in\mathcal{N}^d, b\in\mathcal{N}^k} \frac{1}{nb}\sum_{i=1}^{n}\sum_{j=1}^{b}\left( \langle x'_{i,j}, \widetilde{W}_i(U_t)^\top \rangle \langle a, x'_{i,j} \rangle \langle v_{i,t+1}, b \rangle - \langle a, \widetilde{W}_i(U_t)^\top \rangle \langle v_{i,t+1}, b \rangle \right). \tag{37}$$

Recall that $\widetilde{W}_i = \left( \frac{1}{nb}\sum_{i=1}^{n}\sum_{j=1}^{b} \zeta'_{i,j} x'_{i,j}(v_{i,t+1})^\top \right)_{i,*} \in \mathbb{R}^k$. Conditioning on $\|v_{i,t+1}\|_2 \leq \frac{5}{4}\Gamma$, we have $\|\widetilde{W}_i\|_2 \leq \frac{5R\Gamma\sqrt{24k\log(nb)}}{4\sqrt{nb}}$ with probability at least $1 - ke^{-12\log(nb)} \geq 1 - e^{-11\log(nb)}$ by the union bound on the components of $\widetilde{W}_i$. Multiplication by $U_t^\top$ on the right does not change this bound given orthonormality.

We thus condition on the new event

$$\mathcal{E} = \left\{ \|\widetilde{W}_i U_t^\top\|_2 \leq \frac{5R\Gamma\sqrt{24k\log(nb)}}{4\sqrt{nb}} \text{ and } \|v_{i,t+1}\|_2 \leq \frac{5}{4}\Gamma \text{ for all } i \in [n] \right\}$$

which has probability at least $1 - O(n^{-11})$ via our above work and Proposition 42. The variable $\langle x'_{i,j}, \widetilde{W}_i(U_t)^\top \rangle$ is $\left( \frac{5R\Gamma\sqrt{24k\log(nb)}}{4\sqrt{nb}} \right)$-sub-Gaussian via conditioning on $\mathcal{E}$. This means

$$\frac{1}{nb} \left( \langle x'_{i,j}, \widetilde{W}_i(U_t)^\top \rangle \langle a, x'_{i,j} \rangle \langle v_{i,t+1}, b \rangle - \langle a, \widetilde{W}_i(U_t)^\top \rangle \langle v_{i,t+1}, b \rangle \right)$$

is $\left( \frac{3R\Gamma^2\sqrt{24k\log(nb)}}{\sqrt{n^3 b^3}} \right)$-sub-exponential. Using an argument identical to how we showed (36), the inequality (37) implies

$$\frac{1}{n} \left\| \left( \frac{1}{b}(\mathcal{A}')^\dagger \mathcal{A}'(\widetilde{W}_i(U_t)^\top) - \widetilde{W}_i(U_t)^\top \right)^\top V_{t+1} \right\|_2 \leq \frac{3R\Gamma^2\sqrt{15 \cdot 24 dk \log(n)\log(nb)}}{(nb)^{\frac{3}{4}}} \tag{38}$$

with probability at least $1 - O(n^{-10})$. Combining (36) and (38) via the union bound finishes the second claim.

Combining (34) with Claims 1 and 2 via the union bound finishes the proof. $\square$

Recall the exact statements of Assumption 31 and 32. That is, there exist $c_0, c_1 > 1$ such that

$$m \geq c_0 \left( \frac{\max\{R^2, 1\} \cdot \max\{\Gamma^2, 1\}\gamma^4 k \log^2 n}{E_0^2 \sigma_{\max,*}^2} + \frac{R^2 k}{\Gamma^2} \log(nm) \right) T$$

and

$$n \geq c_1 \left( \frac{\max\{\Delta_{\epsilon,\delta}, 1\}(R+\Gamma)\Gamma d\sqrt{k}\log^2(nm)}{E_0^2 \lambda^2} + \frac{R^2\Gamma^2 d \log(nm)}{m} \right) T.$$

Lower bounds on the constants $c_0, c_1$ are used many times in the proof for the following lemma.

**Lemma** (Restatement of Lemma 33). *Let $E_0 = 1 - \text{dist}^2(U_0, U^*)$ and $\psi = \widetilde{O}\left((R+\Gamma)\Gamma\sqrt{dk}\right)$. Suppose Assumption 31 and 32 hold. Then, for any iteration $t$, we have that $P_{t+1}$ is invertible and*

$$\|P_{t+1}^{-1}\|_2 \leq \left( 1 - \frac{\eta\sigma_{\min,*}^2 E_0}{\sqrt{2\log n}} \right)^{-\frac{1}{2}}$$

*with probability at least $1 - O(n^{-10})$.*

*Proof.* Recall that $F = G^{-1}(GD - C)V^* + \frac{1}{b}G^{-1}\nabla_V\langle \mathcal{A}(V_{t+1}(U_t)^\top), \varsigma \rangle$ and let $\tau_k = c_\tau\sqrt{\frac{35k\log n}{b}}, \tau_k' = 15\sqrt{13}\sqrt{\frac{\Gamma^4 d \log n}{nb}}$ for some constant $c > 0$. Letting $c_0 \geq 2000c_\tau^2$ in Assumption 31, we have $\frac{1}{1-\tau_k} \leq \frac{4}{3}$ and noting $\|G^{-1}\|_2 \leq \frac{1}{1-\tau_k}$ by Lemma 41 part (1), we have

$$\|F\|_2 \leq \frac{4}{3}\nu\tau_k\|V^*\|_2 + \sqrt{\frac{47R^2 n \log(nb)}{b}} \tag{39}$$

with probability at least $1 - O(n^{-11})$ via the argument Lemma 41 part (2) for the spectral norm.

Recall that $Q_t = V_{t+1}(U_t)^\top - V^*(U^*)^\top$ and $V_t = V^*(U^*)^\top U_t - F$ by Lemma 39. Now, denote $H_t' = -\frac{1}{b}(\mathcal{A}')^\dagger \mathcal{A}'(Q_t)V_{t+1} + n\xi_t, H_t = -\frac{1}{b}(\mathcal{A}')^\dagger \mathcal{A}'(Q_t)$, and $W_t = \frac{\eta}{nb}\nabla_U\langle \mathcal{A}(V_{t+1}(U_t)^\top), \varsigma \rangle$. By Recursion 28, we have

$$P_{t+1}^\top P_{t+1} = \hat{U}_{t+1}^\top \hat{U}_{t+1}$$
$$= (U_t)^\top U_t - \frac{\eta}{n}((U_t)^\top H_t' + (H_t')^\top U_t) + ((U_t)^\top W_t + (W_t)^\top U_t) - \frac{\eta}{n}((W_t)^\top H_t' + {H_t'}^\top W_t)$$
$$+ \frac{\eta^2}{n^2}(H_t')^\top H_t' + (W_t)^\top W_t$$

By Weyl's inequality, the above implies

$$
\begin{aligned}
\sigma^2_{\min}(P_{t+1}) &\geq 1 - \frac{\eta}{n}\lambda_{\max}((U_t)^\top H_t V_{t+1} + (V_{t+1})^\top (H_t)^\top U_t) + \lambda_{\min}((U_t)^\top W_t + (W_t)^\top U_t) \\
&\quad - \frac{\eta}{n}\lambda_{\max}((W_t)^\top H_t V_{t+1} + (V_{t+1})^\top (H_t)^\top W_t) - \lambda_{\max}((U_t)^\top \xi_t + \xi_t^\top U_t) \\
&\quad - \lambda_{\max}((W_t)^\top \xi_t + \xi_t^\top W_t) + \frac{\eta^2}{n^2}\lambda_{\min}((H_t')^\top H_t') + \lambda_{\min}((W_t)^\top W_t) \\
&\geq 1 - \frac{\eta}{n}\lambda_{\max}((U_t)^\top H_t V_{t+1} + (V_{t+1})^\top (H_t)^\top U_t) + \lambda_{\min}((U_t)^\top W_t + (W_t)^\top U_t) \\
&\quad - \frac{\eta}{n}\lambda_{\max}((W_t)^\top H_t V_{t+1} + (V_{t+1})^\top (H_t)^\top W_t) - \lambda_{\max}((U_t)^\top \xi_t + \xi_t^\top U_t) \\
&\quad - \lambda_{\max}((W_t)^\top \xi_t + \xi_t^\top W_t).
\end{aligned}
$$

To complete this proof we must bound each of these terms individually. That is, we upper bound

$$
\frac{\eta}{n}\lambda_{\max}((U_t)^\top H_t V_{t+1} + (V_{t+1})^\top (H_t)^\top U_t) \tag{40a}
$$

$$
\lambda_{\max}((U_t)^\top \xi_t + \xi_t^\top U_t) \tag{40b}
$$

$$
\lambda_{\max}((W_t)^\top \xi_t + \xi_t^\top W_t) \tag{40c}
$$

$$
\frac{\eta}{n}\lambda_{\max}((W_t)^\top H_t V_{t+1} + (V_{t+1})^\top (H_t)^\top W_t) \tag{40d}
$$

and lower bound

$$
\lambda_{\min}((U_t)^\top W_t + (W_t)^\top U_t). \tag{41}
$$

These bounds follow from our previous propositions and lemmas.

Term (40a) has

$$
\begin{aligned}
&\frac{\eta}{n}\lambda_{\max}((U_t)^\top H_t V_{t+1} + (V_{t+1})^\top (H_t)^\top U_t) \\
&= \max_{\|a\|_2=1} \frac{\eta}{n} a^\top (U_t)^\top H_t V_{t+1} + (V_{t+1})^\top (H_t)^\top U_t a \\
&= \max_{\|a\|_2=1} \frac{2\eta}{n} a^\top (V_{t+1})^\top H_t U_t a \\
&\leq \max_{\|a\|_2=1} \frac{2\eta}{n} a^\top (V_{t+1})^\top \left(\frac{1}{b}(\mathcal{A}')^\dagger \mathcal{A}'(Q_t) - Q_t\right) U_t a + \max_{\|a\|_2=1} \frac{2\eta}{n} a^\top (V_{t+1})^\top Q_t U_t a.
\end{aligned}
$$

We have

$$
\max_{\|a\|_2=1} \frac{2\eta}{n} a^\top (V_{t+1})^\top \left(\frac{1}{b}(\mathcal{A}')^\dagger \mathcal{A}'(Q_t) - Q_t\right) U_t a \leq 2\eta\tau_k' + \frac{3R\Gamma^2\sqrt{15 \cdot 24dk \log(n)\log(nb)}}{(nb)^{\frac{3}{4}}}
$$

by Lemma 43. We are able to drop the second term on the right-hand side above later due to its very fast rate of decay in $n, b$. Now

$$
\begin{aligned}
\max_{\|a\|_2=1} \frac{2\eta}{n} a^\top (V_{t+1})^\top Q_t U_t a &= \max_{\|a\|_2=1} \frac{2\eta}{n} \left\langle Q_t, V_{t+1}aa^\top (U_t)^\top \right\rangle_F \\
&= \max_{\|a\|_2=1} \frac{2\eta}{n} \left\langle Q_t, V^*(U^*)^\top U_t aa^\top (U_t)^\top \right\rangle_F - \frac{2\eta}{n} \left\langle Q_t, Faa^\top (U_t)^\top \right\rangle_F.
\end{aligned}
$$

We bound the first term above via

$$
\frac{2\eta}{n}\left\langle Q_t, V^*(U^*)^\top U_t a a^\top (U_t)^\top \right\rangle_F
$$

$$
= \frac{2\eta}{n}\operatorname{tr}\left(\left(U_t(V_{t+1})^\top - U^*(V^*)^\top\right) V^*(U^*)^\top U_t a a^\top (U_t)^\top\right)
$$

$$
= \frac{2\eta}{n}\operatorname{tr}\left(\left(U_t(U_t)^\top U^*(V^*)^\top - U_t(F)^\top - U^*(V^*)^\top\right) V^*(U^*)^\top U_t a a^\top (U_t)^\top\right)
$$

$$
= \frac{2\eta}{n}\operatorname{tr}\left(\left(U_t(U_t)^\top - I\right) U^*(V^*)^\top V^*(U^*)^\top U_t a a^\top (U_t)^\top\right) - \frac{2\eta}{n}\operatorname{tr}\left(U_t(F)^\top V^*(U^*)^\top U_t a a^\top (U_t)^\top\right)
$$

$$
= -\frac{2\eta}{n}\operatorname{tr}\left((F)^\top V^*(U^*)^\top U_t a a^\top\right)
$$

$$
\le \frac{2\eta}{n}\left|\operatorname{tr}\left((F)^\top V^*(U^*)^\top U_t a a^\top\right)\right| \tag{42}
$$

$$
= \frac{2\eta}{n}\left|a^\top (F)^\top V^*(U^*)^\top U_t a\right|
$$

$$
\le \frac{2\eta}{n}\|a\|_2\|(F)^\top V^*(U^*)^\top U_t\|_2\|a\|_2
$$

$$
\le \frac{2\eta}{n}\|F\|_2\|V^*\|_2
$$

$$
\le \frac{8}{3}\eta\frac{\nu\tau_k}{1-\tau_k}\sigma_{\max,*}^2 + \sqrt{\frac{47\eta^2 R^2 \sigma_{\max,*}^2 \log(nb)}{b}}
$$

with probability at least $1 - O(n^{-11})$ by (39), where equalities 3 and 4 hold by the cycling property of the trace and $((U_t)_\perp U_t)^\top = 0$. Then

$$
-\frac{2\eta}{n}\left\langle Q_t, F a a^\top (U_t)^\top \right\rangle_F
$$

$$
= -\frac{2\eta}{n}\operatorname{tr}\left(\left(U_t(U_t)^\top U^*(V^*)^\top - U_t(F)^\top - U^*(V^*)^\top\right) F a a^\top (U_t)^\top\right)
$$

$$
= -\frac{2\eta}{n}\operatorname{tr}\left(\left(U_t(U_t)^\top - I\right) U^*(V^*)^\top F a a^\top (U_t)^\top\right) + \frac{2\eta}{n}\operatorname{tr}\left(F a a^\top (U_t)^\top U_t(F)^\top\right) \tag{43}
$$

$$
= \frac{2\eta}{n}a^\top (F)^\top F a
$$

$$
\le \frac{2\eta}{n}\|F\|_2^2
$$

$$
\le \frac{16}{3}\eta\frac{\nu^2\tau_k^2}{(1-\tau_k)^2}\sigma_{\max,*}^2 + \frac{154\eta^2 R^2 \log(nb)}{b}
$$

with probability at least $1 - O(n^{-11})$ where the third equality follows from the cycling property of the trace and $((U_t)_\perp U_t)^\top = 0$. Combining (42) and (43) gives us

$$
\max_{\|a\|_2 = 1}\frac{2\eta}{n}a^\top (V_{t+1})^\top Q_t U_t a \le 8\eta\frac{\nu\tau_k}{(1-\tau_k)^2}\sigma_{\max,*}^2 + \frac{154\eta^2 R^2 \log(nb)}{b}
$$

given that $\tau_k^2 \le \tau_k$ by selecting $c_0 \ge 35c_\tau^2$ in Assumption 31. Define $\bar\tau_k = \nu\tau_k + \frac{\tau_k'}{\sigma_{\max,*}^2}$. Letting $c_0, c_1 \ge 4000$, for (40a), we have

$$
\frac{\eta}{n}\lambda_{\max}((U_t)^\top H_t V_{t+1} + (V_{t+1})^\top (H_t)^\top U_t) \le 2\eta\tau_k' + 8\eta\frac{\nu\tau_k}{(1-\tau_k)^2}\sigma_{\max,*}^2 + \frac{154\eta^2 R^2 \log(nb)}{b}
$$

$$
+ \sqrt{\frac{47\eta^2 R^2 \sigma_{\max,*}^2 \log(nb)}{b}} + \frac{3R\Gamma^2\sqrt{15 \cdot 24dk\log(n)\log(nb)}}{(nb)^{\frac{3}{4}}}
$$

$$
\le 8\eta\frac{\bar\tau_k}{(1-\bar\tau_k)^2}\sigma_{\max,*}^2 + 4\sqrt{\frac{47\eta^2 R^2 \max\{\sigma_{\max,*}^2, 1\}\log(nb)}{b}}.
$$

Note that $(U_t)^\top \xi_t$ in (40b) is a $k \times k$ Gaussian matrix with independent columns. Now, by an equivalent statement as Corollary 27 for a $k \times k$ matrix

$$
\begin{aligned}
\lambda_{\max}((U_t)^\top \xi_t + \xi_t^\top U_t) &= \max_{a:\|a\|_2=1} a^\top (U_t)^\top \xi_t + \xi_t^\top U_t a \\
&= 2 \max_{a:\|a\|_2=1} a^\top (U_t)^\top \xi_t a \\
&\leq 2\|(U_t)^\top \xi_t\|_2 \\
&\leq 2\eta\hat{\sigma}(2\sqrt{k} + w)
\end{aligned}
$$

for $w > 0$ with probability at least $1 - 2e^{-\alpha^2}$ for some constant $c' > 0$. We select $\alpha = \sqrt{10k \log n}$, which means

$$
\lambda_{\max}((U_t)^\top \xi_t + \xi_t^\top U_t) \leq 4\eta\hat{\sigma}\sqrt{10k \log n}
$$

with probability at least $1 - O(n^{-10})$.

Now, for (40c)

$$
\begin{aligned}
\lambda_{\max}((W_t)^\top \xi_t + \xi_t^\top W_t) &= \max_{\|a\|_2=1} a^\top ((W_t)^\top \xi_t + \xi_t^\top W_t)a \\
&\leq 2 \max_{\|a\|_2=1} a^\top \xi_t^\top W_t a \\
&\leq 2\|\xi_t\|_2 \|W_t\|_2 \\
&\leq 2\eta\hat{\sigma}\sqrt{10d \log n}\left(\frac{4}{3}\sqrt{\frac{2 \cdot 15R^2\eta^2\Gamma^2 d \log n}{nb}}\right)
\end{aligned}
$$

with probability at least $1 - O(n^{-10})$ via Corollary 27 and Proposition 40 part (1). Now we bound the term (40d). Note that

$$
\begin{aligned}
\frac{\eta}{n}\lambda_{\max}((W_t)^\top H_t V_{t+1} + (V_{t+1})^\top (H_t)^\top W_t) &\leq \frac{2\eta}{n}\left\|\frac{1}{b}(\mathcal{A}')^\dagger \mathcal{A}'(Q_t)V_{t+1}\right\|_2 \|W_t\|_2 \\
&\leq \frac{2\eta}{n}\left(\left\|\left(\frac{1}{b}(\mathcal{A}')^\dagger \mathcal{A}'(Q_t) - Q_t\right)V_{t+1}\right\|_2 + \|Q_t V_{t+1}\|_2\right)\|W_t\|_2 \quad (44) \\
&\leq \left(3\eta\tau'_k \mathrm{dist}(U_t, U^*) + \frac{2\eta}{n}\|Q_t V_{t+1}\|_2\right)\|W_t\|_2
\end{aligned}
$$

by Lemma 43 and taking $c_0 \geq 4000c_\tau^2$ and $c_1 \geq 4000$ from Assumptions 31 and 32 so $2\eta\tau'_k \mathrm{dist}(U_t, U^*)$ is dominant over the noise term. Then

$$
\frac{\eta}{n}\lambda_{\max}((W_t)^\top H_t V_{t+1} + (V_{t+1})^\top (H_t)^\top W_t) \leq \left(2\eta\tau'_k \mathrm{dist}(U_t, U^*) + \frac{2\eta}{n}\|Q_t V_{t+1}\|_2\right)\|W_t\|_2. \quad (45)
$$

Note that $2\tau'_k \mathrm{dist}(U_t, U^*) \leq 1$ by taking $c_0 \geq 4000c_\tau^2$ and $c_1 \geq 4000$ from Assumptions 31 and 32 since $\tau'_k = 5\sqrt{13}\sqrt{\frac{\Gamma^4 d \log n}{nb}}$. Then, by (39), the orthonormality of $U_t, U^*$, and the definition of $Q_t$

$$
\begin{aligned}
\|Q_t V_{t+1}\|_2 &\leq \|U_t(U_t)^\top U^*(V^*)^\top - U_t(F)^\top - U^*(V^*)^\top\|_2 \|V^*(U^*)^\top U_t - F\|_2 \\
&\leq (2\|V^*\|_2 + \|F\|_2)(\|V^*\|_2 + \|F\|_2) \\
&\leq \left((2 + 2\nu\tau_k)\|V^*\|_2 + \sqrt{\frac{47R^2 n \log(nb)}{b}}\right)\left((1 + 2\nu\tau_k)\|V^*\|_2 + \sqrt{\frac{47R^2 n \log(nb)}{b}}\right) \\
&\leq (2 + 2\nu\tau_k)^2\|V^*\|_2^2 + \sqrt{\frac{47(2 + 2\nu\tau_k)^2 R^2 \|V^*\|_2^2 n \log(nb)}{b}} + \frac{47R^2 n \log(nb)}{b}.
\end{aligned}
$$

Then

$$
\frac{2\eta}{n}\|Q_t V_{t+1}\|_2 \leq 2\eta(2 + \nu\tau_k)^2\sigma_{\max,*}^2 + \frac{39}{5}\sqrt{\frac{47\eta^2 R^2 \sigma_{\max,*}^2 \log(nb)}{b}} + \frac{154\eta R^2 \log(nb)}{b} \quad (46)
$$

by selecting the constant $c_0$ in Assumption 31 to have $c_0 \geq 4000c_\tau^2$ such that $\nu\tau_k \leq \frac{1}{10}$. Recall that

$$\|W_t\|_2 \leq \frac{4}{3}\sqrt{\frac{2 \cdot 15\eta^2 R^2 \Gamma^2 d \log n}{nb}}$$

with probability at least $1 - O(n^{-10})$ by Proposition 40. So, combining (45) and (46)

$$\frac{\eta}{n}\lambda_{\max}((W_t)^\top H_t V_{t+1} + (V_{t+1})^\top (H_t)^\top W_t)$$

$$\leq \frac{4}{3}\sqrt{\frac{2 \cdot 15\eta^2 R^2 \Gamma^2 d \log n}{nb}} + \frac{4}{3}\sqrt{\frac{2 \cdot 15\eta^2 (1 + \nu\tau_k)^2 R^2 \Gamma^2 d\sigma_{\max,*}^2 \log(nb)}{nb}}$$

$$+ \frac{4}{3}\sqrt{\frac{2 \cdot 15\eta^2 R^2 \Gamma^2 d \log n}{nb}}\left(\frac{39}{5}\sqrt{\frac{47\eta^2 R^2 \sigma_{\max,*}^2 \log(nb)}{b}} + \frac{154\eta R^2 \log(nb)}{b}\right)$$

with probability at least $1 - O(n^{-11})$. Recall $\nu\tau_k \leq \frac{1}{10}$ since $c_0 \geq 4000c_\tau^2$. Moreover, for $c_1 \geq 4000$

$$\frac{4}{3}\sqrt{\frac{2 \cdot 15\eta^2 R^2 \Gamma^2 d \log(nb)}{nb}} + \frac{4}{3}\sqrt{\frac{2 \cdot 15\eta^2 (2 + 2\nu\tau_k)^2 R^2 \Gamma^2 d\sigma_{\max,*}^2 \log(nb)}{nb}}$$

$$+ \frac{4}{3}\sqrt{\frac{2 \cdot 15\eta^2 R^2 \Gamma^2 d \log n}{nb}}\left(\frac{39}{5}\sqrt{\frac{47\eta^2 R^2 \sigma_{\max,*}^2 \log(nb)}{b}} + \frac{154\eta R^2 \log(nb)}{b}\right)$$

$$\leq \frac{16}{3}\sqrt{\frac{2 \cdot 15\eta^2 R^2 \Gamma^2 d \max\{\sigma_{\max,*}^2, 1\}\log(nb)}{nb}}$$

by Assumptions 31 and 32.

Observe that for (41)

$$\lambda_{\min}((U_t)^\top W_t + (W_t)^\top U_t) \geq 2\lambda_{\min}((U_t)^\top W_t) \geq -2\big|\lambda_{\min}((U_t)^\top W_t)\big|$$

and

$$-2\big|\lambda_{\min}((U_t)^\top W_t)\big| \geq -2\sigma_{\min}((U_t)^\top W_t)) \geq -2\sigma_{\max}((U_t)^\top W_t)).$$

Note that concentration inequalities hold for the sum of uncorrelated sub-Gaussian random variables and $U_t^\top$ having orthonormal rows. Then, by a modified version of Lemma 40 part (2) for a $k \times k$-dimensional version of $W_t$

$$\sigma_{\max}((U_t)^\top W_t) = \max_{\|a\|_2=1} a^\top (U_t)^\top W_t a$$

$$\leq \|(U_t)^\top W_t\|_2$$

$$\leq \frac{4}{3}\sqrt{\frac{2 \cdot 15\eta^2 R^2 \Gamma^2 k \log(nb)}{nb}}$$

with probability at least $1 - O(n^{-10})$. Notice that our upper bound is independent of the data dimension $d$. Thus

$$\lambda_{\min}((U_t)^\top W_t + (W_t)^\top U_t) \leq \frac{8}{3}\sqrt{\frac{2 \cdot 15\eta^2 R^2 \Gamma^2 k \log(nb)}{nb}}.$$

Via the above work we have, simultaneously by the union bound, there exists a constant $\hat{c} > 0$ such that

$$\frac{\eta}{n}\lambda_{\max}((U_t)^\top H_t V_{t+1} + (V_{t+1})^\top (H_t)^\top U_t) \leq \hat{c}\eta\bar{\tau}_k\sigma_{\max,*}^2 + \hat{c}\sqrt{\frac{\eta^2 R^2 \max\{\sigma_{\max,*}^2, 1\}\log(nb)}{b}}$$

$$\lambda_{\max}((U_t)^\top \xi_t + \xi_t^\top U_t) \leq \hat{c}\eta\hat{\sigma}\sqrt{k \log n}$$

$$\lambda_{\max}((W_t)^\top \xi_t + \xi_t^\top W_t) \leq \hat{c}\eta\hat{\sigma}\sqrt{d \log n}\left(\sqrt{\frac{R^2 \eta^2 \Gamma^2 d \log n}{nb}}\right) \tag{47}$$

$$\frac{\eta}{n}\lambda_{\max}((W_t)^\top H_t V_{t+1} + (V_{t+1})^\top (H_t)^\top W_t) \leq \hat{c}\sqrt{\frac{\eta^2 R^2 \Gamma^2 d \max\{\sigma_{\max,*}^2, 1\}\log(nb)}{nb}}$$

$$\lambda_{\min}((U_t)^\top W_t + (W_t)^\top U_t) \leq \hat{c}\sqrt{\frac{\eta^2 R^2 \Gamma^2 k \log(nb)}{nb}}$$

with probability at least $1 - O(n^{-10})$. The value of $\hat{c}$ may change between lines. Its precise value does not matter and this constant is used to simplify our proof. By our choice of $\psi$, we have $\hat{\sigma} = O\left(\frac{(R+\Gamma)\Gamma\sqrt{Tdk}\log(nb)}{n\epsilon}\right)$. Putting together the bounds in (47) and selecting $c_0 \geq 4000c_\tau^2$ and $c_1 \geq 4000$ in Assumptions 31 and 32

$$
\begin{aligned}
\sigma_{\min}^2(P_{t+1}) \geq\ & 1 - \hat{c}\eta\bar{\tau}_k\sigma_{\max,*}^2 - \hat{c}\eta\hat{\sigma}\sqrt{k\log n} \\
& - \hat{c}\sqrt{\frac{\eta^2 R^2 \max\{\sigma_{\max,*}^2, 1\}\log(nb)}{b}} - \hat{c}\eta\hat{\sigma}\sqrt{d\log n}\left(\sqrt{\frac{R^2\eta^2\Gamma^2 d\log n}{nb}}\right) \\
& - \hat{c}\sqrt{\frac{\eta^2 R^2\Gamma^2 k\log(nb)}{nb}} - \hat{c}\sqrt{\frac{\eta^2 R^2\Gamma^2 d\max\{\sigma_{\max,*}^2, 1\}\log(nb)}{nb}} \\
\geq\ & 1 - \hat{c}\eta\bar{\tau}_k\sigma_{\max,*}^2 - 2\hat{c}\eta\hat{\sigma}\sqrt{d\log n} \\
& - 2\hat{c}\sqrt{\frac{\eta^2 R^2 \max\{\Gamma^2, 1\}\log(nb)}{b}} - \hat{c}\sqrt{\frac{\eta^2 R^2\Gamma^2 d\max\{\sigma_{\max,*}^2, 1\}\log(nb)}{nb}}
\end{aligned}
$$

with probability at least $1 - O(n^{-10})$. Via our choice of $\hat{\sigma}$, there is a constant $\hat{c} > 0$ such that

$$
\begin{aligned}
\sigma_{\min}^2(P_{t+1}) \geq\ & 1 - \hat{c}\eta\bar{\tau}_k\sigma_{\max,*}^2 - \frac{2\hat{c}\eta(R+\Gamma)\Gamma d\sqrt{Tk\log(1.25/\delta)\log^3(nb)}}{n\epsilon} \\
& - 2\hat{c}\sqrt{\frac{\eta^2 R^2 \max\{\Gamma^2, 1\}\log(nb)}{b}} - \hat{c}\sqrt{\frac{\eta^2 R^2\Gamma^2 d\max\{\sigma_{\max,*}^2, 1\}\log(nb)}{nb}}
\end{aligned}
$$

Recall again that $\tau_k = c_\tau\sqrt{\frac{35k\log n}{b}}$, $\tau_k' = 5\sqrt{13}\sqrt{\frac{\Gamma^4 d\log n}{nb}}$, and $\nu = \frac{\Gamma}{\sigma_{\max,*}} \geq 1$. Then

$$
\begin{aligned}
\eta\bar{\tau}_k\sigma_{\max,*}^2 &= \eta\nu\tau_k\sigma_{\max,*}^2 + \eta\tau_k' \\
&= c_\tau\eta\nu\sigma_{\max,*}^2\sqrt{\frac{35k\log n}{b}} + 5\sqrt{13}\eta\nu^2\sigma_{\max,*}^2\sqrt{\frac{d\log n}{nb}}.
\end{aligned}
$$

Recall the exact statements of Assumption 31 and 32. That is, there exist $c_0, c_1 > 1$ such that

$$
m \geq c_0\left(\frac{\max\{R^2, 1\}\cdot\max\{\Gamma^2, 1\}\gamma^4 k\log^2 n}{E_0^2\sigma_{\max,*}^2} + \frac{R^2 k}{\Gamma^2}\log(nm)\right)T
$$

and

$$
n \geq c_1\left(\frac{\max\{\Delta_{\epsilon,\delta}, 1\}(R+\Gamma)\Gamma d\sqrt{k}\log^2(nm)}{E_0^2\lambda^2} + \frac{R^2\Gamma^2 d\log(nm)}{m}\right)T.
$$

The problem parameters that lower bound $m, n$ now come into use during the following steps. Furthermore, recall that by Assumption 4 we know some $\lambda > 0$ such that $\lambda \leq \sigma_{\min,*}$. Then, there exists a constant $\hat{c} > 0$ such that

$$
\hat{c}\eta\bar{\tau}_k\sigma_{\max,*}^2 \leq \frac{\hat{c}\eta\sigma_{\min,*}^2 E_0}{\sqrt{c_0\log n}} + \frac{\hat{c}\eta\lambda^2 E_0^2}{\sqrt{c_0 c_1\gamma^2 k^{\frac{3}{2}}\log^2(nm)\log n}}.
$$

and thus

$$
\hat{c}\eta\bar{\tau}_k\sigma_{\max,*}^2 \leq \frac{\hat{c}\eta\sigma_{\min,*}^2 E_0}{\sqrt{c_0\log n}} + \frac{\hat{c}\eta\sigma_{\min,*}^2 E_0^2}{\sqrt{c_0 c_1\gamma^2 k^{\frac{3}{2}}\log^2(nm)\log n}}.
\tag{48}
$$

since $\lambda \leq \sigma_{\min,*}$. Further, this same constant $\hat{c}$ satisfies

$$\frac{2\hat{c}\eta(R+\Gamma)\Gamma d\sqrt{Tk\log(1.25/\delta)\log^3(nb)}}{n\epsilon} + \hat{c}\sqrt{\frac{\eta^2 R^2 \max\{\Gamma^2,1\}\log(nb)}{b}}$$
$$+ \hat{c}\sqrt{\frac{\eta^2 R^2 \Gamma^2 d \max\{\sigma_{\max,*}^2,1\}\log(nb)}{nb}}$$
$$\leq \frac{2\hat{c}\eta\lambda^2 E_0^2}{c_1\sqrt{\log(nm)}} + \hat{c}\eta\sigma_{\min,*}^2 E_0 \sqrt{\frac{1}{c_0 \max\{\Gamma^2,1\}k\log n}}$$
$$+ \hat{c}\eta\lambda\sigma_{\min,*}E_0^2 \sqrt{\frac{\max\{\sigma_{\max,*}^2,1\}}{c_0 c_1 \max\{\Gamma^2,1\}k^{\frac{3}{2}}\log^2(n)\log(nm)}}$$

and thus

$$\frac{2\hat{c}\eta(R+\Gamma)\Gamma d\sqrt{Tk\log(1.25/\delta)\log^3(nb)}}{n\epsilon} + \hat{c}\sqrt{\frac{\eta^2 R^2 \max\{\Gamma^2,1\}\log(nb)}{b}}$$
$$+ \hat{c}\sqrt{\frac{\eta^2 R^2 \Gamma^2 d \max\{\sigma_{\max,*}^2,1\}\log(nb)}{nb}}$$
$$\leq \frac{2\hat{c}\eta\sigma_{\min,*}^2 E_0^2}{c_1\sqrt{\log(nm)}} + \hat{c}\eta\sigma_{\min,*}^2 E_0 \sqrt{\frac{1}{c_0 \max\{\Gamma^2,1\}k\log n}} \tag{49}$$
$$+ \hat{c}\eta\sigma_{\min,*}^2 E_0^2 \sqrt{\frac{\max\{\sigma_{\max,*}^2,1\}}{c_0 c_1 \max\{\Gamma^2,1\}k^{\frac{3}{2}}\log^2(n)\log(nm)}}$$

since $\lambda \leq \sigma_{\min,*}$. Note that $E_0^2 \leq E_0$ since $E_0 \in (0,1)$. Then, combining $c_0, c_1 \geq \max\{10\sqrt{2}\hat{c}, 10\sqrt{2}\hat{c}^2, 4000c_\tau^2, 4000\}$ with (48) and (49) gives us

$$\sigma_{\min}^2(P_{t+1}) \geq 1 - \frac{\eta\sigma_{\min,*}^2 E_0}{\sqrt{2\log n}} \tag{50}$$

with probability at least $1 - O(n^{-10})$. □

## B.4 Private FedRep experiments

In this subsection we describe the synthetic data experiments we designed to compare our Private FedRep (Algorithm 1) to the Private Alternating Minimization Meta-Algorithm (Priv-AltMin) of (Jain et al., 2021). Our comparison is described in Figure 2 as a graph of population mean square error (MSE) over choice of privacy parameter $\epsilon > 0$. Note that we also experimented with non-private AltMin, which performs similarly to the non-private FedRep in Figure 2.

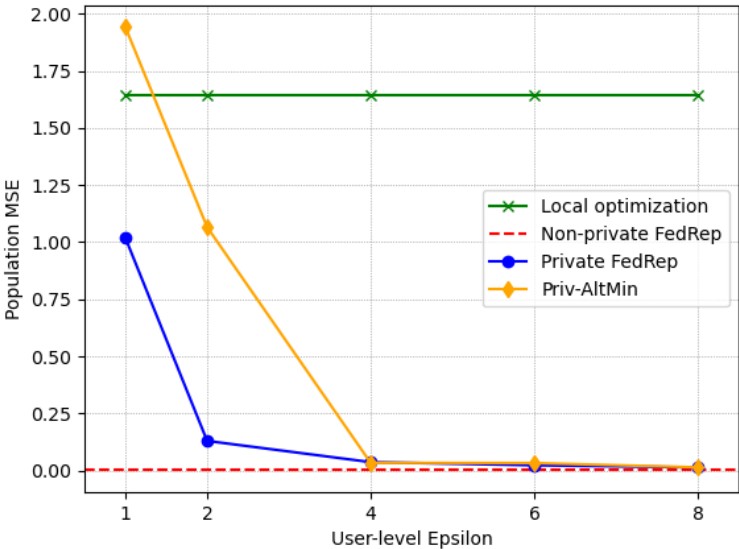

Figure 2: Graph of population MSE over choice of privacy parameter $\epsilon \in [1, 8]$. Local optimization is done via non-private gradient descent on each user's data separately and their population MSEs averaging over $n$ users.

The features $x \in \mathbb{R}^d$ of our synthetic data are sampled from a standard normal Gaussian distribution. We select optimal parameters $U^*, v_1^*, \ldots, v_n^*$ by sampling $U$ from a $d \times k$-dimensional Gaussian distribution then generating an orthonormal matrix $U^*$ via $(U^*, P) = \mathrm{QR}(U)$ and sampling $v_i^* \sim \mathcal{N}(0, I_k)$ for all $i \in [n]$. Labels are generated as in Assumption 6 and we choose Gaussian noise with standard deviation $R = 0.01$. Further, both Private FedRep and Priv-AltMin are initialized using an implementation of Algorithm 2.

Our problem is instantiated with $d = 50$, $k = 2$, $m = 10$, and $n = 20,000$. For FedRep we prune our hyperparameters, deciding on $T = 5$ and learning rate $\eta = 2.5$ with clipping parameter $\psi = 10$. Similarly, Priv-AltMin with iterations optimized for $T = 5$ and clipping parameter $10^{-4}$. The Gaussian mechanism variance for both algorithms is calculated using the privacy parameter $\Delta_{\epsilon,\delta} = \frac{\sqrt{16 \log(1.25/\delta)}}{\epsilon}$ with $\delta = 10^{-6}$.

# C  Missing Proofs for Section 4

We first restate our algorithm along with some initial definitions and results.

Define $\mathcal{B}_F = \{U \in \mathbb{R}^{d \times k} : \|U\|_F \leq \sqrt{2k}\}$. Assume for simplicity that $m$ is even. We also partitioned $S_i = S_i^0 \cup S_i^1$ where $S_i^0 = \{z_{1,j}, \ldots, z_{\frac{m}{2},j}\}$ and $S_i^1 = \{z_{\frac{m}{2}+1,j}, \ldots, z_{m,j}\}$ for each $i \in [n]$. Further, we denote $S^t = (S_1^t, \ldots, S_n^t)$ where $t \in \{0, 1\}$. Suppose that $S = (S_1, \ldots, S_n) \subset \mathcal{Z}^{nm(d+1)}$ is a sequence of $n$ datasets with $m$ samples each. Let $S_M = ((S_1)_M, \ldots (S_n)_M)$ where $(S_i)_M = \{(Mx_{i,j}, y_{i,j}) : j \in [m]\}$ for all $i \in [n]$

---

**Algorithm 3** Private Representation Learning for Personalized Classification

---

**Require:** dataset sequences $S^0$ and $S^1$ of equal size, score function $f(U', \cdot) = -\min_{V \in \mathcal{V}} \widehat{L}_\rho(U', V; \cdot)$ over matrices
$\quad U' \in \mathbb{R}^{k' \times k}$, privacy parameter $\epsilon > 0$, target dimension $k' = O\left(\frac{r^2 \Gamma^2 \log(nm/\delta)}{\rho^2}\right)$,

1: Sample $M \in \mathbb{R}^{k' \times d}$ with entries drawn i.i.d uniformly from $\left\{\pm\frac{1}{\sqrt{k'}}\right\}$
2: Let $S_M = ((S_1)_M, \ldots, (S_n)_M)$ where $(S_i)_M = \{(Mx, y) : (x, y) \in S_i^0\}$ for $i \in [n]$
3: Let $\mathcal{N}^\gamma$ be a Frobenius norm $\gamma$-cover of $\mathcal{B}_F$
4: Run the exponential mechanism over $\mathcal{N}^\gamma$, privacy parameter $\epsilon$, sensitivity $\frac{1}{n}$, and score function $f(U', S_M)$, to select
$\quad \widetilde{U} \in \mathcal{N}^\gamma$
5: Let $U^{\text{priv}} \leftarrow M^\top \widetilde{U}$
6: Each user $i \in [n]$ independently computes $v_i^{\text{priv}} \leftarrow \arg\min_{\|v\|_2 \leq \Gamma} \widehat{L}(U^{\text{priv}}, v, S_i^1)$
7: **Return:** $U^{\text{priv}}, V^{\text{priv}} = [v_1^{\text{priv}}, \ldots v_n^{\text{priv}}]^\top$

---

**Definition** (Restatement of Definition 13). Let $(U, v) \in \mathbb{R}^{d \times k} \times \mathbb{R}^k$ and $(x, y) \in \mathbb{R}^d \times \{-1, 1\}$ any data point. We define the **margin loss** as

$$\ell_\rho(U, v, z) = \mathbb{1}\left[y\langle x, Uv\rangle \leq \rho\right]$$

and denote the **0-1 loss**

$$\ell(U, v, z) = \ell_0(U, v, z) = \mathbb{1}\left[y\langle x, Uv\rangle \leq 0\right].$$

**Definition** (Restatement of Definition 14). Let $G \subset \mathbb{R}^d$ be any set of $t$ vectors. Fix $\tau, \beta \in (0, 1)$. We call the random matrix $M \in \mathbb{R}^{k' \times d}$ a $(t, \tau, \beta)$-**Johnson-Lindenstrauss (JL) transform** if for any $u, u' \in G$

$$|\langle Mu, Mu'\rangle - \langle u, u'\rangle| \leq \tau \|u\|_2 \|u'\|_2$$

with probability at least $1 - \beta$ over $M$.

**Proposition 44** ((Woodruff et al., 2014)). *Let $G \subset \mathbb{R}^q$ be any set of $t$ vectors. Fix $\tau, \beta \in (0, 1)$. For $M \in \mathbb{R}^{k \times q}$ a $(t, \tau, \beta)$-JL transform for any $u \in G$*

$$(1 - \tau)\|u\|_2^2 \leq \|Mu\|_2^2 \leq (1 + \tau)\|u\|_2^2$$

*with probability at least $1 - \beta$ over $M$, which holds simultaneously with Definition 14.*

**Lemma** (Restatement of Lemma 15). *Let $\tau, \beta \in (0, 1)$. Take $G \subset \mathbb{R}^d$ to be any set of $t$ vectors. Setting $k' = O\left(\frac{\log\left(\frac{t}{\beta}\right)}{\tau^2}\right)$*

*for a $k' \times d$ matrix $M$ with entries drawn uniformly and independently from $\left\{\pm\frac{1}{\sqrt{k'}}\right\}$ implies that $M$ is a $(t, \tau, \beta)$-JL transform.*

**Lemma 45** (Lemma D.1 (Bassily et al., 2022)). *Fix $\beta \in (0, 1)$. Suppose $U \in \mathbb{R}^{d \times k}$ and that $x_{i,j} \in \mathbb{R}^d$, $v_i \in \mathbb{R}^k$ for all $(i, j) \in [n] \times [m]$. Let $M \in \mathbb{R}^{k' \times d}$ be a $((n+1)m, \gamma, \beta/2)$-JL transform in the sense of Proposition 44. Then, there exists a constant $c \geq 1$ such that, with probability at least $1 - \beta$ over the randomness of $M$, we have*

$$\|Mx_{i,j}\|_2^2 \leq \left(1 + c\sqrt{\frac{\log(nm/\beta)}{k'}}\right)\|x_{i,j}\|_2^2 \tag{1}$$

$$\|MU\|_F^2 \leq \left(1 + c\sqrt{\frac{\log(nm/\beta)}{k'}}\right)\|U\|_F^2 \tag{2}$$

$$\left| v_i^\top U^\top M^\top M x_{i,j} - v_i^\top U^\top x_{i,j} \right| \le c \| U v_i \|_2 \| x_{i,j} \|_2 \sqrt{\frac{\log\left(nm/\beta\right)}{k'}}. \tag{3}$$

*for all $(i,j) \in [n] \times [m]$ simultaneously.*

Recall that $\mathcal{U}$ is the space of orthonormal $d \times k$ matrices and $\mathcal{V}$ the set of $n \times k$ matrices with columns whose Euclidean norms are bounded by $\Gamma > 0$. Throughout the following results we assume that $\gamma \le c\sqrt{\frac{\log(nm/\beta)}{k'}}$ for some constant $c \ge 1$ and a target dimension $k'$ for a JL transform. Furthermore, whenever we define a JL transform $M$, we assume that it preserves the norm of data points $x_{i,j}$ for all $(i,j) \in [n] \times [m]$ and some fixed matrix in $U \in \mathcal{U}$. Let $\mathcal{N}^\gamma$ be a Frobenius $\gamma$-cover of the $\sqrt{2k}$-radius Frobenius ball $\mathcal{B}_F$. By cover we mean that $\mathcal{N}^\gamma$ contains the center points of Frobenius balls of $\gamma$-radius whose union contain $\mathcal{B}_F$. Note as well that $\mathcal{N}^\gamma \subset \mathcal{B}_F$.

**Proposition 46.** *Let $M \in \mathbb{R}^{k' \times d}$ be a $((n+1)m, \gamma, \beta/2)$-JL transform in the sense of Proposition 44. Assume $c^2 \log\left(nm/\beta\right) \le k'$ and that $x_{i,j}$ is a sequence of $r$-bounded feature vectors for each $(i,j) \in [n] \times [m]$. Suppose $\mathcal{N}^\gamma$ is a Frobenius norm $\gamma$-cover of the $k' \times k$-dimensional $\sqrt{2k}$ radius ball. Moreover, let $U \in \mathcal{U}$ and $\widetilde{U} \in \mathcal{N}^\gamma$ where $\|\widetilde{U} - MU\|_F \le \gamma$. Then, there exists a constant $c \ge 1$ where, with probability at least $1 - \beta$ over the randomness of $M$, we have*

$$\left| v_i^\top \widetilde{U}^\top M x_{i,j} - v_i^\top U^\top x_{i,j} \right| \le (\sqrt{2}+1) c r \Gamma \sqrt{\frac{\log\left(nm/\beta\right)}{k'}}$$

*for any $V = [v_1, \ldots, v_n] \in \mathcal{V}$ and all $(i,j) \in [n] \times [m]$ simultaneously.*

*Proof.* Note that since $U$ has orthonormal columns, we have $\|U v_i\|_2 = \|v_i\|_2 \le \Gamma$ for all $j$. So

$$\left| v_i^\top U^\top M^\top M x_{i,j} - v_i^\top U^\top x_{i,j} \right| \le c r \Gamma \sqrt{\frac{\log\left(nm/\beta\right)}{k'}} \tag{47}$$

with probability at least $1 - \frac{1}{2}\beta$ by part (3) of Lemma 45. Recall that $\gamma \le c\sqrt{\frac{\log(nm/\beta)}{k'}}$ for some constant $c \ge 1$. Assume $c^2 r^2 \Gamma^2 \log\left(nm/\beta\right) \le k'$, which implies $\gamma \le 1$. Let $\mathcal{B}_F$ be the $k' \times k$-dimensional Frobenius ball of radius $\sqrt{2k}$. We define $\mathcal{N}^\gamma$ to be a Frobenius norm $\gamma$-cover of $\mathcal{B}_F$. Note $\|MU\|_F \le \sqrt{2k}$ with probability at least $1 - \frac{1}{2}\beta$ by part (2) of Lemma 45. That is, with probability at least $1 - \frac{1}{2}\beta$, there exists $\widetilde{U} \in \mathcal{N}^\gamma$ such that $\|\widetilde{U} - MU\| \le \gamma$. Let $\widetilde{U} \in \mathcal{N}^\gamma$ be within $\gamma$ Frobenius-distance of $MU$ conditioned on the event $\|MU\|_F \le \sqrt{2k}$.

Now, there exists a constant $c \ge 1$ such that

$$\begin{aligned}
\left| v_i^\top \widetilde{U}^\top M x_{i,j} - v_i^\top U^\top M^\top M x_{i,j} \right| &\le \|v_i\|_2 \left\| \widetilde{U}^\top M x_{i,j} - U^\top M^\top M x_{i,j} \right\|_2 \\
&\le \|v_i\|_2 \|\widetilde{U} - MU\|_F \|M x_{i,j}\|_2 \\
&\le \sqrt{2} r \Gamma \gamma \\
&\le \sqrt{2} c r \Gamma \sqrt{\frac{\log\left(nm/\beta\right)}{k'}}
\end{aligned}$$

where the third inequality holds by part (1) of Lemma 45 along with our choice of $\widetilde{U}$ and the fourth inequality by our choice of $\gamma$. Hence

$$\left| v_i^\top \widetilde{U}^\top M x_{i,j} - v_i^\top U^\top M^\top M x_{i,j} \right| \le \sqrt{2} c r \Gamma \sqrt{\frac{\log\left(nm/\beta\right)}{k'}} \tag{48}$$

with probability at least $1 - \frac{1}{2}\beta$. Combining (47) and (48) with the union bound and triangle inequality completes the proof. $\qquad \square$

**Lemma** (Restatement of Lemma 16). *Fix $\epsilon, \rho > 0, \beta \in (0,1)$. Algorithm 3 is $(\epsilon, 0)$-user-level DP. Sample $S \sim \mathcal{D}^m$. Then, Algorithm 3 returns $U^{\mathrm{priv}}$ from input $S$ such that*

$$\min_{V \in \mathcal{V}} \widehat{L}(U^{\mathrm{priv}}, V; S^0) \le \min_{(U,V) \in \mathcal{U} \times \mathcal{V}} \widehat{L}_\rho(U, V; S^0) + \widetilde{O}\left( \frac{r^2 \Gamma^2 k}{\epsilon \rho^2 n} \right)$$

*with probability at least $1 - \beta$ over the randomness of $S$ and the internal randomness of the algorithm.*

*Proof.* Let $M \in \mathbb{R}^{k' \times d}$ be a $((n+1)m, \gamma, \beta/4)$-JL transform in the sense of Proposition 44. Further, let $\mathcal{B}_F \subseteq \mathbb{R}^{k' \times k}$ be the Frobenius ball of radius $\sqrt{2k}$. Recall that $\gamma \le c\sqrt{\frac{\log(nm/\beta)}{k'}}$ for a constant $c \ge 1$. Assume $c^2 \log(nm/\beta) \le k'$, which implies $\gamma \le 1$.

Define $V_{U'} \in \arg\min_{V \in \mathcal{V}} \widehat{L}_\rho(U', V; S^0)$ for each $U' \in \mathbb{R}^{d \times k}$ and fix $U \in \arg\min_{U' \in \mathcal{U}} \widehat{L}_\rho(U', V_{U'}; S^0)$. Denote the columns of $V_U$ as $v_1, \dots, v_n$. Let $\mathcal{N}^\gamma$ be a Frobenius norm $\gamma$-cover over $\mathcal{B}_F$. We have $\|MU\|_F \le \sqrt{2k}$ with probability at least $1 - \frac{1}{4}\beta$ by part (2) of Lemma 45. Choose $\gamma = \frac{a\rho}{(\sqrt{2}+1)cr\Gamma}$ for some constant $a \in (0, 1)$ along with $\hat{U} \in \mathcal{N}^\gamma$ within Frobenius distance $\gamma$ of $MU$ conditioned on the event $\|MU\|_F \le \sqrt{2k}$. Then, by Proposition 46, for all $(i, j) \in [n] \times [m]$ simultaneously, we have

$$\left| v_i^\top \hat{U}^\top M x_{i,j} - v_i^\top U^\top x_{i,j} \right| \le a\rho$$

with probability at least $1 - \frac{1}{2}\beta$. Assuming $1 - a\rho \ge 0$, the above implies, with probability at least $1 - \frac{1}{2}\beta$, that

$$\widehat{L}(M^\top \hat{U}, V_{M^\top \hat{U}}; S^0) \le \widehat{L}(M^\top \hat{U}, V_U; S^0) \le \widehat{L}_\rho(U, V_U; S^0) \tag{49}$$

where the left-hand inequality holds by $V_{M^\top \hat{U}}$ being a minimizer for $\widehat{L}(M^\top \hat{U}, \cdot\ S^0)$.

Let $\bar{U} \in \arg\min_{U' \in \mathcal{B}_F} \widehat{L}(M^\top U', V_{M^\top U'}; S^0)$. Then, by definition of $\bar{U}$ and the fact that $\mathcal{N}^\gamma \subset \mathcal{B}_F$, we have

$$\widehat{L}(M^\top \bar{U}, V_{M^\top \bar{U}}; S^0) \le \widehat{L}(M^\top \hat{U}, V_{M^\top \hat{U}}; S^0). \tag{50}$$

Let $(S_i^0)_M = \{(Mx_{i,j}, y_{i,j}) : j \in [m/2]\}$ for all $i \in [n]$ and $S_M = ((S_i^0)_M)_{i \in [n]}$. This definition and the characterization of our losses in Definition 13 imply $\widehat{L}(\bar{U}, V_{M^\top \bar{U}}; S_M^0) = \widehat{L}(M^\top \bar{U}, V_{M^\top \bar{U}}; S^0)$. Via the exponential mechanism over $\mathcal{N}^\gamma$ given score function $-\widehat{L}(U', V_{M^\top U'}; S_M^0)$ for $U' \in \mathcal{N}^\gamma$ and the usual empirical loss guarantees for the exponential mechanism, we obtain $U^{\mathrm{priv}} \in \mathbb{R}^{d \times k}$ where $U^{\mathrm{priv}} = M^\top \widetilde{U}$ for some $\widetilde{U} \in \mathcal{N}^\gamma$ such that

$$\widehat{L}(U^{\mathrm{priv}}, V_{U^{\mathrm{priv}}}; S^0) = \widehat{L}(\widetilde{U}, V_{M^\top \widetilde{U}}; S_M^0) \le \widehat{L}(\bar{U}, V_{M^\top \bar{U}}; S_M^0) + O\left(\frac{\log|\mathcal{N}^\gamma|}{\epsilon n}\right)$$

with probability at least $1 - \frac{1}{2}\beta$. This gives us

$$\widehat{L}(U^{\mathrm{priv}}, V_{U^{\mathrm{priv}}}; S^0) \le \widehat{L}(M^\top \bar{U}, V_{M^\top \bar{U}}; S^0) + O\left(\frac{r^2 \Gamma^2 k \log\left(\frac{r\Gamma\sqrt{k}}{\rho}\right) \log((nm/\beta))}{\epsilon \rho^2 n}\right) \tag{51}$$

since $|\mathcal{N}^\gamma| = O\left(\left(\frac{\sqrt{k}}{\gamma}\right)^{k'k}\right)$ and $k' = O\left(\frac{r^2 \Gamma^2 \log((nm/\beta))}{\rho^2}\right)$ by our choice of $\gamma$. Combining (49), (50), and (51) via the union bound along with recalling our definition $V_{U'} \in \arg\min_{V \in \mathcal{V}} \widehat{L}_\rho(U', V; S^0)$ finishes the proof. $\qquad\square$

**Theorem** (Restatement of Theorem 17). *Fix $\epsilon, \rho > 0, \beta \in (0, 1)$. Algorithm 3 is $(\epsilon, 0)$-user-level DP in the billboard model. Sample user data $S \sim \mathcal{D}^m$. Then, Algorithm 3 returns $U^{\mathrm{priv}}, V^{\mathrm{priv}}$ from input $S$ such that*

$$L(U^{\mathrm{priv}}, V^{\mathrm{priv}}; \mathcal{D}) \le \min_{(U,V) \in \mathcal{U} \times \mathcal{V}} \widehat{L}_\rho(U, V; S^0) + \widetilde{O}\left(\frac{r^2 \Gamma^2 k}{n\epsilon\rho^2} + \sqrt{\frac{r^2 \Gamma^2}{m\rho^2}}\right)$$

*with probability at least $1 - \beta$ over the randomness of $S$ and the internal randomness of the algorithm.*

*Proof.* Let $V^{\mathrm{priv}} \in \mathcal{V}$ be the matrix with columns $v_i^{\mathrm{priv}} \in \arg\min_{\|v\| \le \Gamma} \widehat{L}(U^{\mathrm{priv}}, v; S_i^1)$ and $\hat{V} \in \mathcal{V}$ with columns $\hat{v}_i \in \arg\min_{\|v\| \le \Gamma} \widehat{L}(U^{\mathrm{priv}}, v; S_i^0)$ for each $i \in [n]$. We first note that

$$
\begin{aligned}
&L(U^{\mathrm{priv}}, V^{\mathrm{priv}}; \mathcal{D}) - \min_{(U,V) \in \mathcal{U} \times \mathcal{V}} \widehat{L}_\rho(U, V; S^0) \\
&= \left(L(U^{\mathrm{priv}}, V^{\mathrm{priv}}; \mathcal{D}) - L(U^{\mathrm{priv}}, \hat{V}; \mathcal{D})\right) + \left(L(U^{\mathrm{priv}}, \hat{V}; \mathcal{D}) - \widehat{L}(U^{\mathrm{priv}}, \hat{V}, S^0)\right) \\
&\quad + \left(\widehat{L}(U^{\mathrm{priv}}, \hat{V}, S^0) - \min_{(U,V) \in \mathcal{U} \times \mathcal{V}} \widehat{L}_\rho(U, V; S^0)\right)
\end{aligned}
\tag{52}
$$

Let $M \in \mathbb{R}^{k' \times d}$ be a $((n+1)m, \gamma, \beta/6)$-JL transform in the sense of Proposition 44. Let $\gamma = c\sqrt{\frac{\log(nm/\beta)}{k'}}$ for a constant $c > 0$ and $\gamma = O\left(\frac{\rho}{r\Gamma}\right)$. Assume $c^2 \log(nm/\beta) \leq k'$, which implies $\gamma \leq 1$. Denote by $\mathcal{B}_F$ the $k' \times k$ Frobenius ball of radius $\sqrt{2k}$. Let $\mathcal{N}^\gamma$ be a $\gamma$-cover of $\mathcal{B}_F$. Using Lemma 16, we have

$$\widehat{L}(U^{\mathrm{priv}}, \hat{V}; S^0) - \min_{(U,V) \in \mathcal{U} \times \mathcal{V}} \widehat{L}_\rho(U, V; S^0) \leq \tilde{O}\left(\frac{r^2 \Gamma^2 k}{\epsilon \rho^2 n}\right) \tag{53}$$

with probability at least $1 - \frac{1}{3}\beta$ over the randomness of $M$.

**Claim 1:** We have

$$L(U^{\mathrm{priv}}, V^{\mathrm{priv}}; \mathcal{D}) - L(U^{\mathrm{priv}}, \hat{V}; \mathcal{D}) \leq \tilde{O}\left(\sqrt{\frac{k'}{m}}\right)$$

with probability at least $1 - \frac{1}{3}\beta$.

By the definition of $V^{\mathrm{priv}}$, we have

$$\widehat{L}(U^{\mathrm{priv}}, V^{\mathrm{priv}}; S^1) \leq \widehat{L}(U^{\mathrm{priv}}, \hat{V}; S^1). \tag{54}$$

Our proof strategy is to obtain generalization error bounds for the parameters $(U^{\mathrm{priv}}, V^{\mathrm{priv}})$ and $(U^{\mathrm{priv}}, \hat{V})$ with respect to the loss $\widehat{L}(\,\cdot\,; S^1)$, which we leverage to prove the claim. We first analyze the generalization properties with respect to $\ell, \mathcal{D}$ of our 2-layer linear networks induced by the matrix product $Uv$ for any $U \in \mathcal{B}_F$ and $v \in \mathbb{R}^{k'}$ with $\|v\|_2 \leq \Gamma$. We denote the family of 2-layer linear networks induced by the matrix product $Uv$ as $\mathcal{L}$ and define $\mathcal{B}_\Gamma \subseteq \mathbb{R}^{k'}$ to be the centered $\sqrt{2k}\Gamma$-radius Euclidean ball. Let $\mathcal{H}_0$ be the space of binary classifiers induced by taking the sign of the functionals $\langle \,\cdot\,, Uv \rangle$ with $Uv \in \mathcal{L}$. Similarly, define $\mathcal{H}$ to be the space of binary linear classifiers induced by functions $\langle \,\cdot\,, w \rangle$ for all $w \in \mathcal{B}_\Gamma$.

Since $\mathcal{L} \subseteq \mathcal{B}_\Gamma$ each functional $\langle \,\cdot\,, Uv \rangle$ must also be contained in the space of functionals $\langle \,\cdot\,, w \rangle$ with $w \in \mathcal{B}_\Gamma$. This naturally implies $\mathcal{H}_0 \subseteq \mathcal{H}$. Hence, the VC dimension of $\mathcal{H}_0$ is no larger than the VC dimension of $\mathcal{H}$, i.e. at most $k' + 1$. Recall $\mathcal{N}^\gamma \subseteq \mathcal{B}_F$ and that, by the description of Algorithm 3, there is a particular $\widetilde{U} \in \mathcal{N}^\gamma$ such that $U^{\mathrm{priv}} = M^\top \widetilde{U}$. Then, we have $\widetilde{U} \in \mathcal{B}_F$ and thus the binary classifier induced by $\widetilde{U}v$ is in $\mathcal{H}_0$ for any $v$ with $\|v\|_2 \leq \Gamma$.

To garner a generalization guarantee, we use the fact that the VC dimension of $\mathcal{H}_0$ is $O(k')$. Recall that $(S_i)_M = \{(Mx_{i,j}, y_{i,j}) : (x_{i,j}, y_{i,j}) \in S_i\}$ for each $i \in [n]$. Denote $S_M = ((S_i)_M)_{i \in [n]}$ and define the sequence of distributions $\mathcal{D}_M = ((\mathcal{D}_1)_M, \ldots, (\mathcal{D}_n)_M)$ where for, each $i \in [n]$, we have that $x' \sim (\mathcal{D}_i)_M$ has the same distribution as $Mx$ with $x \sim \mathcal{D}_i$. Thus, we obtain, from the VC dimension generalization error bounds on $\mathcal{H}$ that, for each $i \in [n]$, the following

$$\left| L(\widetilde{U}, v_i^{\mathrm{priv}}; (\mathcal{D}_i)_M) - \widehat{L}(\widetilde{U}, v_i^{\mathrm{priv}}; (S_i^1)_M) \right| \leq \tilde{O}\left(\sqrt{\frac{k'}{m}}\right)$$

and

$$\left| L(\widetilde{U}, \hat{v}_i; (\mathcal{D}_i)_M) - \widehat{L}(\widetilde{U}, \hat{v}_i; (S_i^1)_M) \right| \leq \tilde{O}\left(\sqrt{\frac{k'}{m}}\right)$$

with probability at least $1 - \frac{1}{6n}\beta$. Then, by the union bound and taking the arithmetic mean over the $n$ users

$$\left| L(\widetilde{U}, V^{\mathrm{priv}}; \mathcal{D}_M) - \widehat{L}(\widetilde{U}, V^{\mathrm{priv}}; S_M^1) \right| \leq \tilde{O}\left(\sqrt{\frac{k'}{m}}\right)$$

and

$$\left| L(\widetilde{U}, \hat{V}; \mathcal{D}_M) - \widehat{L}(\widetilde{U}, \hat{V}; S_M^1) \right| \leq \tilde{O}\left(\sqrt{\frac{k'}{m}}\right)$$

with probability at least $1 - \frac{1}{3}\beta$. Via the inner product that characterizes our losses in Definition 13, we have $L(\widetilde{U}, \hat{V}; \mathcal{D}_M) = L(U^{\mathrm{priv}}, \hat{V}; \mathcal{D})$ and $L(\widetilde{U}, V^{\mathrm{priv}}; \mathcal{D}_M) = L(U^{\mathrm{priv}}, V^{\mathrm{priv}}; \mathcal{D})$. Using the above inequalities and (54), we obtain

$$L(U^{\mathrm{priv}}, V^{\mathrm{priv}}; \mathcal{D}) - L(U^{\mathrm{priv}}, \hat{V}; \mathcal{D}) \leq \tilde{O}\left(\sqrt{\frac{k'}{m}}\right)$$

with probability at least $1 - \frac{1}{3}\beta$, which proves Claim 1.

Another application of the VC bound and the union bound gives us, with probability at least $1 - \frac{1}{3}\beta$, that

$$L(U^{\mathrm{priv}}, \hat{V}; \mathcal{D}) - \widehat{L}(U^{\mathrm{priv}}, \hat{V}; S^0) \leq \widetilde{O}\left(\sqrt{\frac{k'}{m}}\right). \tag{55}$$

Combining (53), Claim 1, and (55) via the union bound, and recalling that $k' = O\left(\frac{r^2 \Gamma^2 \log(nm/\beta)}{\rho^2}\right)$ by our choice of $\gamma$, finishes the proof. $\qquad\square$

