# OpenReview forum: "Private Model Personalization Revisited"
_ICML.cc/2025/Conference — ICML 2025 poster_

### Official Review · Reviewer_hcg6 · 2025-03-04

**Overall Recommendation:** 3

**Summary:**

The paper addresses the problem of model personalization under user-level differential privacy (DP) in the federated learning setting. The authors propose a novel private federated learning algorithm based on the FedRep framework, which learns a shared low-dimensional embedding and user-specific local models while ensuring DP. The main contributions include: (1) an efficient private federated algorithm that handles noisy labels and sub-Gaussian data distributions, (2) improved privacy-utility trade-offs compared to prior work, (3) a private initialization algorithm for the shared embedding, and (4) dimension-independent risk guarantees for binary classification using the Johnson-Lindenstrauss transform. The paper claims to improve the privacy error term by a factor of $\tilde{O}(dk)$ in certain parameter regimes and extends the applicability to broader data distributions compared to prior work.

## update after rebuttal
The authors have addressed two of my concerns, and I am generally satisfied with the rebuttal, so I have raised my score. While I raised my score, I remain somewhat unenthusiastic about the paper overall.

**Claims And Evidence:**

The main claims are supported by proofs.

**Essential References Not Discussed:**

I have not read any paper about model personalization.

**Experimental Designs Or Analyses:**

The empirical evaluation includes a simple privacy-utility trade-off analysis in terms of MSE.

**Methods And Evaluation Criteria:**

The proposed methodology is well-explained and aligns with intuitive expectations. The evaluation criterion used is MSE, which is appropriate for the vector estimation problem.

**Other Comments Or Suggestions:**

Please ensure the correct usage of \cite, \citep, and \citet. Note that the formatting of citations may vary depending on the LaTeX template used.

**Other Strengths And Weaknesses:**

I feel the practical significance of the work is rather limited. The current low-dimensional embedding assumption seems applicable only to limited scenarios. Moreover, the methodology relies heavily on the low-dimensional embedding assumption and does not appear to extend to neural network-based methods. Could the authors provide some discussion on this point?

Additionally, the contribution does not appear significant enough to me. The algorithm builds largely on FedRep, and the theoretical contribution seems to be an extension of Jain et al., 2021.

**Questions For Authors:**

The paper is written in a highly technical manner, presenting theoretical assumptions and results. I suggest the authors provide more high-level discussions, particularly regarding the following points. I would be happy to raise my score if these questions are addressed well:

- Intrinsic Difference Between Classification and Regression: What is the fundamental difference between classification and regression? If they are both treated as supervised problems, are there conclusions that hold for one but not the other? The authors treat these two problems separately in two sections, but I am somewhat confused about their differences beyond the technical aspects (e.g., different scores, target spaces). Could the authors comment on this?

- Intrinsic Difference Between Gaussian and Sub-Gaussian Assumptions: What is the fundamental difference between Gaussian and sub-Gaussian distribution assumptions? In previous literature (Jain et al., 2021), is the Gaussian distribution necessary due to its unique properties (e.g., rotation invariance, equivalence between independence and uncorrelatedness), or is it merely about tail probabilities? If the latter, I do not see a significant improvement. The paper emphasizes the improvement from Gaussian to sub-Gaussian assumptions as a main contribution multiple times, but there seems to be no detailed discussion on this point.

**Relation To Broader Scientific Literature:**

The paper extends previous work (Jain et al., 2021). I am not familiar with the task of model personalization, so I do not have a strong opinion on this point.

**Theoretical Claims:**

The proofs are reviewed but not rigorously verified.

---

> ### Author Rebuttal · Authors · 2025-03-30
>
> We thank the reviewer for their detailed comments. We respectfully disagree with several conclusions and clarify key contributions and motivations below.
>
> **On the Practical Significance of the Low-Dimensional Embedding Assumption**:\
> The *low-dimensional* shared representation assumption is widely adopted in both theoretical and practical work on model personalization and federated learning, including Collins et al. (ICML 2021), Jain et al. (NeurIPS 2021), Tripuraneni et al. (ICML 2021), and Duchi et al. (NeurIPS 2022). It reflects practical scenarios (e.g., in multitask/meta-learning) and enables rigorous analysis and tractable algorithms.
>
> The *linear* shared representation setting captures a **fundamental learning problem** that has attracted significant recent interest in the literature, including the works referenced above. Many influential works explore it to understand key statistical and computational challenges in learning from heterogeneous data. Despite its apparent simplicity, this setting remains **analytically challenging and far from fully understood**, as even recent works require structural assumptions (e.g., Gaussianity, identity covariance, boundedness) to obtain formal guarantees. These challenges are magnified under differential privacy, which introduces additional technical hurdles such as sensitivity control and dealing with noise accumulation while ensuring good convergence behavior. While neural-network-based personalization has shown empirical success, it is generally **infeasible to derive formal risk guarantees** in such settings due to their non-convexity. As a result, most works rely on empirical evaluation. Our work, by contrast, focuses on providing **rigorous theoretical insights** in a meaningful and analyzable setup—paving the way for future extensions to more complex models.
>
> **On the Role of FedRep and Comparison to Prior Work**:\
> While we build on FedRep (Collins et al., 2021), our contributions go significantly beyond:
> - We extend FedRep to satisfy **user-level differential privacy** in a federated setting, while also handling **noisy labels**—an important practical case not previously addressed. The addition of noisy labels requires substantial changes to the utility analysis.
> - Compared with the original FedRep algorithm, we make several modifications that are crucial to our analysis. For instance, we modify the algorithm to use **disjoint batches** for updating $U$ and $V$, unlike FedRep. This crucial change allows tighter control of gradient norms, reduces the noise required for DP, and leads to **improved convergence rates** and sharper utility bounds.
> - Our **utility bound** improves over the **centralized approach of Jain et al. (2021)** by a factor of $\tilde{O}(dk)$ in key regimes.
> - Unlike Jain et al. (2021), our algorithm is **fully federated**, avoiding the need for centralization or exact minimization—making it more scalable and practical.
> These results constitute both conceptual and analytical innovations and establish state-of-the-art privacy-utility trade-offs in this setting.
>
> **On Regression vs. Classification**:\
> Classification with margin loss requires a different analytical and algorithmic approach from regression with quadratic loss. The regression analysis benefits from the structure of the quadratic loss, which enables deriving risk bounds through optimization and concentration arguments that are tied to the sub-Gaussianity and covariance structure of data. In contrast, margin-based losses lack these convenient analytical properties, requiring distinct analysis strategy and algorithmic approach. In our work, we develop a tailored approach for the classification setting—leveraging dimensionality reduction and margin-based analysis—to obtain the **first dimension-independent risk bound under user-level DP** in this framework.
>
> **On the Significance of Extending to Sub-Gaussian Distributions**:\
> The extension from Gaussian to sub-Gaussian data is **not a minor technical change**. Gaussianity permits powerful simplifications due to properties such as rotational invariance and moment equivalence. For example, (Jain et al. 2021) explicitly leverages rotational invariance (spherical symmetry) for their initialization analysis. Further, the proof of their main bound hinges upon several results from (Thekumparampil et al., 2021) that rely on the said properties for the case of independent, standard Gaussian features. In contrast, sub-Gaussian distributions are much more general (e.g., bounded or light-tailed features) and lack such symmetries.
>
> Our extension beyond Gaussian data required a **new analysis** that leverages concentration inequalities (e.g., Vershynin, 2018) to handle sub-Gaussian data with heterogeneous, non-identical feature distributions, significantly broadening the applicability of our private personalization algorithm.
>
> **On Citation Formatting**: We will ensure citation formatting consistency.

---

> > ### Comment · Reviewer_hcg6 · 2025-04-02
> >
> > I appreciate the authors' rebuttal, which addresses part of my concern regarding the extension from Gaussian to sub-Gaussian distributions. Regarding the regression vs. classification point, it would also be helpful if the authors could clarify which theoretical results do not apply without the quadratic loss—this would make my verification easier.
> >
> > While I will raise my score, I remain somewhat unenthusiastic about the paper overall.

---

> > > ### Author Response · Authors · 2025-04-02
> > >
> > > Thanks to the reviewer for the prompt response and for the positive update. We are happy to clarify the role of the quadratic loss in our theoretical analysis.
> > >
> > > The use of the quadratic loss is essential to several key lemmas in our work, particularly those that underpin the convergence analysis of our private FedRep algorithm.
> > >
> > > Most notably,  Lemma 27 (Appendix B.1) is critical to establishing a recursive relationship between $\text{dist}(U_{t+1}, U^*)$ and $\text{dist}(U_t, U^*)$, which is used in proving Lemma 9, one of our main results. In Lemma 27, we show that
> > > $$\text{dist}(U_{t+1}, U^*) \leq \alpha_t \text{dist}(U_t, U^*) + \text{error term}$$
> > > where $\alpha_t = \left\lVert P_{t+1}^{-1} \right\rVert_2   \left\lVert I_k-\eta \Sigma_{V_t}  \right\rVert_2 $ is a value between 0 and 1.
> > >
> > > The derivation of this recursive inequality explicitly relies on the closed-form expression for the gradient of the quadratic loss in the linear regression setting. In particular, lines 786 - 796 in the appendix use the structure of the quadratic loss to simplify and control this recursion. It is unclear whether a similar recursive relationship could be derived for other loss functions lacking this structure.
> > >
> > > Furthermore, in Lemma 37 (Appendix B.3), we provide a closed-form expression for $V_{t+1}$, the matrix where row $i$ corresponds to the local vector  $v_{t+1}^i$ obtained by minimizing a quadratic loss. This closed-form expression plays a critical role in other key results, such as Proposition 40 (used in the privacy analysis) and Lemma 33, which contributes to the final convergence bounds. For general loss functions, such a closed-form solution may not exist, and our arguments in those sections may not extend directly.
> > >
> > > We also note that prior works, such as (Collins et al., Jain et al. 2021 and Thekumparampil et al., 2021), similarly rely on the analytical tractability afforded by the quadratic loss in their theoretical analyses.

---

### Official Review · Reviewer_qvLQ · 2025-03-14

**Overall Recommendation:** 3

**Summary:**

The authors present a novel technique to unlock differently private personalised models in the shared representation framework in a federated setting.

**Claims And Evidence:**

The claims are sufficient and well supported.

**Essential References Not Discussed:**

Nothing of note.

**Experimental Designs Or Analyses:**

The design of experiments seems to be sufficient for the domain of the paper (albeit the actual experiments are few, as mentioned previously).

**Methods And Evaluation Criteria:**

While the paper is focused on theory the experiments are rather light and only compare with a base method using only synthetic datasets. It would be great to have an expanded experiment section applying it to real-world datasets and for additional methods to raise convition about its applicability.

**Other Comments Or Suggestions:**

Nothing of note.

**Other Strengths And Weaknesses:**

It is highly appreciated that the authors shared the code and hope, if their work gets published, to be attached as an artefact to this submission to enhance dissemination.

**Questions For Authors:**

I would like to ask the authors how they can ensure that the model remains private after iterations, surely assuming they use composition under differential privacy to preserve the DP under aggregation only weakens over time as results are "revealed" and propagated.

It would be great to expand on this and address this concern.w

**Relation To Broader Scientific Literature:**

The authors address a very important problem, especially in the case of federated computing. Unlocking the ability to have user personalised models is not only novel but really important for a variery of applications.

**Theoretical Claims:**

The theoretical claims appear to be sound.

---

> ### Author Rebuttal · Authors · 2025-03-30
>
> We thank the reviewer for their positive feedback.
>
> **On the scope of the experimental evaluation**: We would like to emphasize that the primary contribution of our work is theoretical, and the experimental section is designed to validate our theoretical findings and demonstrate the concrete advantages of our algorithm. Our experiments follow the same setup as (Jain et al., 2021), allowing for a direct and meaningful comparison that highlights the improved privacy-utility trade-off achieved by our method. While our focus is not on empirical benchmarking across datasets, we believe the current results effectively showcase the key strengths of our approach. Extending the empirical evaluation to additional datasets and broader personalization settings is a valuable direction for future work and will further complement the strong theoretical foundation established in this paper.
>
> **On the privacy guarantee after composition**: Indeed, we take into account the privacy loss over the iterations of the algorithm. We leverage the adaptive composition guarantee of zero-concentrated differential privacy (zCDP) to bound the overall privacy loss of our algorithm and ensure it satisfies $(\epsilon, \delta)$-DP (see the proof of Theorem 8, Appendix B.1). Specifically, we show that the privacy cost for each iteration is $O\left(\frac{\epsilon^2}{T\log(1/\delta)}\right)$-zCDP. By applying composition over $T$ iterations, we obtain an overall privacy cost of $O(\epsilon^2/\log(1/\delta))$-zCDP, which implies an $(\epsilon, \delta)$-DP guarantee. Also, note that, after $U^\text{priv}$ is computed and sent to the users, each user $i$ will compute their own final local vectors $v_i^\text{priv}$ independently via this $U^\text{priv}$ matrix and a reserved set of fresh local data. The vector $v_i^\text{priv}$ will be kept by user $i$ (for each $i$) and never shared; thus, computing $v_i^\text{priv}$ will not lead to any additional privacy loss.

---

### Official Review · Reviewer_CiH8 · 2025-03-14

**Overall Recommendation:** 4

**Summary:**

The authors study model personalization under user-level differential privacy in a shared representation framework for the federated learning setting. Specifically, there are $n$ users, and the data for user $i\in[n]$ is generated using $y = x^Tw^\star_i + \zeta$ where $\zeta$ is sub-gaussian noise and $w_i^\star = U^\star v_i^\star$ for some shared representation $U^\star\in \mathbb{R}^{d\times k}$ where $k<<d$. This setting was popularized by works such as [this one](https://arxiv.org/pdf/2002.11684) studying representation learning for meta-learning and other applications. This representation learning model has been previously shown to be useful for understanding the effectiveness of [Federated averaging](https://arxiv.org/pdf/2205.13692) and proposing a new algorithm called [FedRep for personalized federated learning](https://arxiv.org/pdf/2102.07078).

In this paper, the authors propose a federated algorithm that extends the FedRep method to ensure user-level differential privacy while including a private initialization step. Their algorithm accommodates sub-Gaussian feature distributions, noisy labels, and heterogeneous user data. In comparison to the [most closely related work](https://proceedings.neurips.cc/paper/2021/hash/f8580959e35cb0934479bb007fb241c2-Abstract.html) on private model personalization this paper relaxes the gaussian data assumption and allows federated training (i.e., without exchanging raw data and only sharing privatized model updates). The authors also improve the utility guarantee provided by this prior work in a reasonable low data regime. Overall, the theory in the paper and one small toy experiment show that the new approach improves the balance between privacy and utility for linear regression and matches or exceeds earlier methods’ performance.

The paper also addresses the binary classification setting with a margin-based loss. It uses a Johnson-Lindenstrauss transform to reduce dimensionality and obtain a margin-based risk bound that does not depend on the original feature dimension.

**Claims And Evidence:**

Yes, this is a technically solid paper.

**Essential References Not Discussed:**

No references I can think of.

**Experimental Designs Or Analyses:**

Yes.

**Methods And Evaluation Criteria:**

Yes, this is a theoretical paper, and the central guarantees in Theorems 10 and 16 make sense.

**Other Comments Or Suggestions:**

1. $u$ was not defined while defining the clipping function in lines 193-194. I suppose it is the unit Frobinus norm version of the Matrix?

2. In terms of presentation, it would also be good to state the utility guarantee for a new user using the algorithm's learned shared representation. This can be stated in a corollary after Theorem 10.

**Other Strengths And Weaknesses:**

The paper is very well written, with explicit assumptions and theoretical results. It relaxes assumptions required by prior work by allowing sub-Gaussian feature distributions and improving over the existing centralized results in reasonable regimes.

If I were to nitpick, the paper could benefit from additional empirical evaluations that measure performance on more diverse datasets and compare implementation complexities with other approaches. The paper focuses primarily on the strongly convex quadratic loss, so researchers who work with other loss functions might find the scope somewhat narrow. Having said that, the authors deliver a technically solid paper, and I am inclined to accept it.

**Questions For Authors:**

1. Have the authors thought about co-variate heterogeneity? We often see a combination of covariate and label shifts in a distributed setting. It would be nice to see how these different sources of heterogeneity affect the final personalization guarantee.

**Relation To Broader Scientific Literature:**

See the summary.

**Theoretical Claims:**

I checked some proofs just to understand what is happening in the paper, but not every proof.

---

> ### Author Rebuttal · Authors · 2025-03-30
>
> We thank the reviewer for their thoughtful and constructive comments and for their appreciation of our work.
>
> **On the scope of the experimental evaluation**: We would like to emphasize that the primary contribution of our work is theoretical, and the experimental section is designed to validate our theoretical findings and demonstrate the concrete advantages of our algorithm. Our experiments follow the same setup as (Jain et al., 2021), allowing for a direct and meaningful comparison that highlights the improved privacy-utility trade-off achieved by our method. While our focus is not on empirical benchmarking across datasets, we believe the current results effectively showcase the key strengths of our approach. Extending the empirical evaluation to additional datasets and broader personalization settings is a valuable direction for future work and will further complement the strong theoretical foundation established in this paper.
>
> On the other comments of the reviewer:
> 1. **Definition of $u$**: The reviewer is correct about the definition of $u$. We will correct the oversight and define the clipping function more clearly. Indeed, $u$ represents the normalized direction of the gradient; namely, it is the matrix $M$ normalized by its Frobenius norm, $M/\left\lVert M \right\rVert_F$.
> 2. **Utility guarantee for new users**: We agree that a utility corollary for new users using the learned shared representation is valuable. Indeed, the bound on the distance to $U^*$ (given by Lemma 9) can be used to bound the excess risk of new users. We will add this result as a corollary of Theorem 10 as suggested by the reviewer.
> 3. **Co-variate heterogeneity**: Our analysis does allow for a certain degree of co-variate heterogeneity among users. In particular, our results hold when user distributions differ as long as they are sub-Gaussian and share a common covariance structure. Note that the prior work of (Jain et al., 2021) lacks co-variate heterogeneity altogether as it requires all the data features to be i.i.d. standard Gaussian. We agree that extending the framework to handle more general co-variate and label shifts is an important direction for future work.

---

### Decision · Program_Chairs · 2025-05-01

**Decision:**

Accept (poster)

**Comment:**

The paper studies model personalization with user-level differential privacy. In this problem, each user wants to learn a different parameter in R^d. These parameters share an unknown embedding that reduces the dimensions from d to k i.e. each parameter vector is the product of a k-dimensional vector and the embedding. The goal is to learn the embedding and the parameter vectors for all users. The paper gives a private algorithm based on previous work in the non-private setting. The new algorithm works for all subgaussian data, generalizing previous work for only gaussian data. Additionally, the privacy error is reduced in certain settings. All reviewers are positive about the paper. The technical content is significant though the reviewers also note that the result only applies to the somewhat limited setting of a linear model with quadratic loss functions (this is common to a lot of work in the area).